# Elevated atmospheric $CO_2$ concentration and vegetation structural changes contributed to GPP increase more than climate and forest cover changes in subtropical forests of China

Tao Chen[1, 2, *], Félicien Meunier[2], Marc Peaucelle[3], Guoping Tang[1, *], Ye Yuan[4], Hans Verbeeck[2]

[1.] Carbon-Water Observation and Research Station in Karst Regions of Northern Guangdong, School of Geography and Planning, Sun Yat-Sen University, Guangzhou 510006, China

[2.] CAVElab – Computational and Applied Vegetation Ecology, Department of Environment, Ghent University, Ghent 9000, Belgium

[3.] INRAE, Université de Bordeaux, UMR 1391 ISPA, 33140 Villenave-d'Ornon, France

[4.] State Key Laboratory of Desert and Oasis Ecology, Xinjiang Institute of Ecology and Geography, Chinese Academy of Sciences, Urumqi 830011, China

*Corresponding to: Tao Chen (chent265@mail2.sysu.edu.cn); Guoping Tang (tanggp3@mail.sysu.edu.cn)

**Abstract:** The subtropical forests of China play a pivotal role in the global carbon cycle and in regulating the global climate. Quantifying the individual and combined effects of forest cover change (FCC), vegetation structural change (e.g., leaf area index (LAI)), $CO_2$ fertilisation, and climate change (CC) on the annual gross primary productivity (GPP) dynamics of different subtropical forest types are essential for mitigating carbon emissions and predicting future climate changes, but these impacts remain unclear. In this study, we used a processed-based model to comprehensively investigate the impacts of these factors on GPP variations with a series of model experiments in China's subtropical forests from 2001 to 2018. Simulated GPP showed a significant increasing trend (20.67 gC/m$^2$/year, $p < 0.001$) under the interaction effects of FCC, LAI change, rising $CO_2$ and CC. The $CO_2$ fertilisation (6.84 gC/m$^2$/year, $p < 0.001$) and LAI change (3.79 gC/m$^2$/year, $p = 0.004$) were the two dominant drivers of total subtropical forest GPP increase, followed by the effects of FCC (0.52 gC/m$^2$/year, $p < 0.001$) and CC (0.92 gC/m$^2$/year, $p = 0.080$). We observed different responses to drivers depending on forest types. The evergreen broadleaved forests showed the maximum carbon sequestration rate due to the positive effects of all drivers. Both the FCC (0.19 gC/m$^2$/year, $p < 0.05$) and CC (1.22 gC/m$^2$/year, $p < 0.05$) significantly decreased evergreen needleleaved forest GPP, while their negative effects were almost offset by the positive impact of LAI changes. Our results indicated that LAI outweighed FCC in promoting GPP, which is an essential driver that needs to be accounted for in studies, and ecological and management programs. Overall, our study offers a novel perspective on different drivers of subtropical forest GPP changes and provides valuable information for policy makers to better manage subtropical forests to mitigate climate change risks.

**Keywords:** Subtropical forests, Gross primary production (GPP), Vegetation structure change, Climate change, BEPS process-based model

**Abbreviations:** BEPS, the Boreal Ecosystem Productivity Simulator; GPP, Gross primary productivity; FCC, Forest cover change; LAI, Leaf area index; CC, Climate change; $CO_2$, Carbon dioxide; EBF, Evergreen broadleaved forest; ENF, Evergreen needle-leaved forest; DBF, Deciduous broadleaved forest; MXF, Mixed forest; QYZ, Qianyanzhou station; DHS, Dinghushan station; ALS, Ailaoshan station; $V_{cmax}$, the maximum carboxylation rate; NEP, Net ecosystem productivity; ER, Ecosystem respiration.

## 1. Introduction

Terrestrial ecosystems can capture carbon dioxide ($CO_2$) from the atmosphere through photosynthesis, which is regarded as a potential solution for slowing down the increase in global $CO_2$ concentrations (Keenan et al., 2016) and mitigating global warming (Fang et al., 2018; Shevliakova et al., 2013). Forest ecosystems, which cover about 30% of the global land area (Forzieri et al., 2022), are one of the main terrestrial carbon sinks (Mathias and Trugman, 2022; Pan et al., 2011) through photosynthesis (Beer et al., 2010). China's forest ecosystems, with an area of approximately $1.95 \times 10^6$ $km^2$ (Li et al., 2014), are mainly distributed in the subtropical regions, which are an important component of the global forest ecosystem and crucial to the global and regional climate system (Fang et al., 2010; Yu et al., 2014). However, China is still one of the world's top emitters of greenhouse gases that directly contribute to global warming (Friedlingstein et al., 2022; Yu et al., 2014). GPP is an important indicator reflecting ecosystem carbon sequestration capacity, which drives terrestrial carbon sequestration and partially offsets anthropogenic $CO_2$ emissions. Therefore, precise quantification of China's subtropical forest GPP and understanding of its driving mechanisms are of great importance for scientists and policy makers to mitigate climate change and carbon emissions with the carbon sink potential of Chinese subtropical forests (Fang et al., 2010; Yu et al., 2014).

Several key national ecological restoration programmes have been implemented in China to reverse land and environmental degradation (Lu et al., 2018). As a result, China's natural and planted forest area increased by $2.3 \times 10^7$ ha and $2.6 \times 10^7$ ha during the past two decades, respectively (Chen et al., 2021b). Remote sensing observations have also identified hotspots of forest gains and greening in southern China resulting from these programs' implementation (Chen et al., 2019a; Tong et al., 2018). However, the subtropical regions are the most developed in China and have a very high density with more than 10% (approximately 0.82 billion) of the world population. Intense land cover/use changes have become prominent in this region due to rapid industrialisation and urbanisation, leading to serious changes to forest ecosystems (e.g., LAI and GPP) (Chen et al., 2019b; Tong et al., 2018; Zhang et al., 2014). Previous studies reported that LAI was an important biotic driver of carbon sink increase in China's forest ecosystems (Chen et al., 2019a; Chen et al., 2019b). Moreover, LAI is a critical parameter for depicting vegetation canopy structure, which can influence some important photosynthetic parameters (e.g., quantum yield ($\alpha$), diurnal ecosystem respiration rate ($R_d$), etc.), and in particular, it can determine the amount of photosynthetically active sunlight that is absorbed by vegetation and thus influence the photosynthetic assimilation rate (Piao et al., 2020). In addition, LAI can influence the annual productivity of vegetation by determining the length of the growing season (i.e., phenology). Meanwhile, the annual mean atmospheric $CO_2$ concentration in China has reached new highs due to large anthropogenic emissions (e.g., 407 ppm in 2017) (CMA, 2018). Elevated $CO_2$ concentrations may enrich intercellular $CO_2$ content in leaves and thus enhance photosynthetic rates and plant productivity (i.e.,

GPP) at the ecosystem scale, which is known as the $CO_2$ fertilisation effect (Piao et al., 2020). $CO_2$ fertilisation was also identified as the pivotal driver for enhancing carbon sink in terrestrial ecosystems, and some studies even reported that the southern region of China was more affected by the $CO_2$ fertilisation effect than other Chinese regions (Chen et al., 2019b; Zhu et al., 2016).

In addition to these drivers, annual mean temperature in the Chinese subtropical monsoon region has increased by more than 1.0 °C over the past 30 years (Fang et al., 2018), which was higher than the global surface temperature increase (Sun et al., 2019) and also influenced the China's forest carbon uptake (Gao et al., 2017; Yuan et al., 2016). Recently, several studies investigated the roles of climate factors in regulating forest GPP changes at the site or global scales (Barman et al., 2014; Ma et al., 2015),

as well as in some regions of China (Ma et al., 2019; Yao et al., 2018b). For instance, previous studies showed that temperature was the major factor influencing GPP variations in the Yangtze River Basin of southern China (Nie et al., 2023), as well as in other southern parts of China (Ma et al., 2019). Generally, a proper increasing temperature can promote enzyme activity and $CO_2$ fixation (Siddik et al., 2019; Moore, et al., 2021). However, when the temperature increases exceed the optimal temperature, the

activity of enzymes in plants will decrease, thereby affecting their photosynthesis rates and carbon sequestration. Climate warming can also increase the vapour pressure deficit (VPD), leading to more drought stress on plants (Yuan et al., 2019). When atmospheric moisture is insufficient, plants tend to inhibit photosynthesis by reducing stomatal conductance, thereby significantly reducing GPP (Yuan et al., 2019; Grossiord et al., 2020). Furthermore, Li et al., (2022) highlighted that precipitation dominated

the interannual changes in forest GPP in Southwest China, while vegetation productivity response to precipitation variations shows large spatial heterogeneity (Camberlin et al., 2007), which largely depends on topographic attributes, vegetation types, and even soil texture. However, a previous study also indicated that the GPP changes were more affected by solar radiation than by precipitation and temperature in humid regions of China (Chen et al., 2021a). Therefore, the dominant factors affecting

forest GPP varied depending on regions and time scales, and thus these studies in identifying the drivers of changes in GPP resulted in divergent conclusions. Moreover, some of recent studies mainly considered different forests as a single forest type and attempted to untangle the individual and combined impact of different factors on forest GPP changes (Chen et al., 2021a; Zhang et al., 2022). However, the relative contributions of these factors to China's subtropical forest GPP variations of specific forest types were

still not clear.

    Over the past decades, different methods have been used to estimate vegetation GPP. Process-based models, especially in combination with remote sensing data (Chen et al., 2019b; Liu et al., 1997), are by far one of the most important tools for studying different forests by explicitly representing processes and their interactions with the environment and for disentangling the drivers of GPP variations

over multiple spatiotemporal scales. The Boreal Ecosystem Productivity Simulator (BEPS) was developed based on the FOREST-BGC model (Running and Coughlan, 1988), which is a process-based diagnostic model and has the advantage of incorporating remote sensing data (e.g., LAI and land cover type) to represent the solid biophysical processes. Recently, the BEPS model has been widely used to simulate carbon fluxes at the regional and global scales (Chen et al., 2019b; Chen et al., 2012; Liu et al.,

1997; Luo et al., 2019; Wang et al., 2021a). Although, it has been well evaluated and validated in China

(Feng et al., 2007; Liu et al., 2018; Peng et al., 2021; Wang et al., 2018), it has not been used to unravel the drivers of different forest changes.

Therefore, in this study, we focus on the subtropical forest ecosystems of China. The BEPS model is used to simulate the GPP of different subtropical forests. The specific objective of this study is to (1) test the performance of the BEPS model in simulating the GPP of China's subtropical forest ecosystems, (2) quantify the spatiotemporal trends in the GPP of different forest types across the subtropics, and (3) disentangle the relative effects of forest cover change, climate change, LAI change, and $CO_2$ fertilisation on different forest GPP variations in the study area. The results of this study can provide forest managers with a basic reference on the carbon sequestration potential of different Chinese subtropical forests. Moreover, investigating the dynamics of GPP and their dominant driving factors in the study area is crucial for decision-makers to adjust and optimise forest management policies promptly, so as to ensure that forests can provide the best ecological services for humans.

## 2.   Materials and methods

### 2.1 Study area description

In this study, we focused on China's subtropical forests which account for approximately 64% (~$1.25 \times 10^6$ km$^2$) of the total forested area in China. The boundary of the subtropical region was derived from the Resource and Environment Science and Data Center of China (He et al., 2021a; He et al., 2019). It covers a latitudinal range of 21.33–33.91 °N and a longitudinal range of 91.39–122.49 °E and has a typical subtropical monsoon climate. The mean annual temperature was between 10.8 °C and 22.9 °C normally increasing from the northwest toward the southeast. The mean annual precipitation ranged from 800 mm in the north to more than 2,000 mm in the south, with 80% of precipitation concentrated in the growing season. The main forest types in the subtropical region of China include the evergreen broad-leaved forest (EBF), evergreen needle-leaved forest (ENF), deciduous broad-leaved forest (DBF) and mixed forest (MXF) (Fig.1). There are three operating flux towers in the study area: Qianyanzhou (QYZ), Dinghushan (DHS), and Ailaoshan (ALS). A more detailed description of these flux tower sites can be found in Table S1.

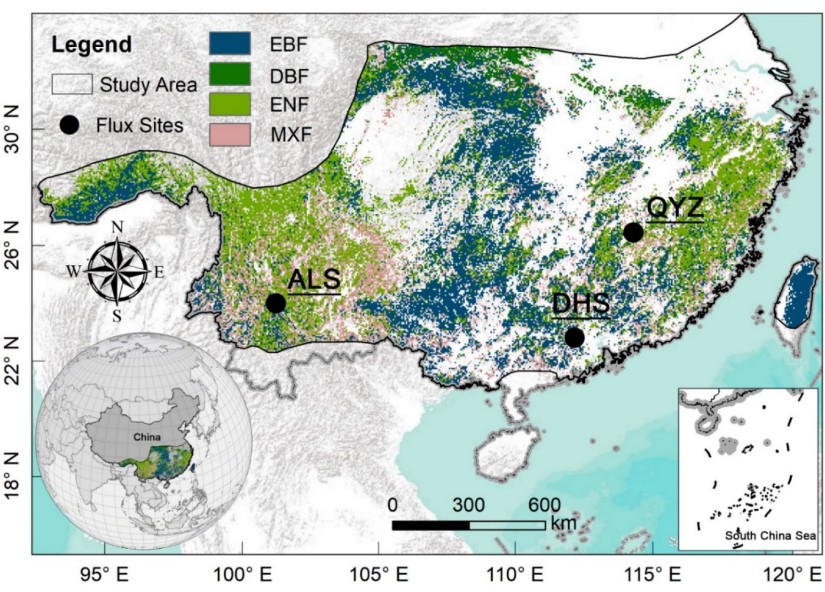

**Figure 1.** Location of the study area and 3 flux sites. The forest cover map (2018) shown here was derived from the European Space Agency land cover data (ESA CCI-LC). The forest types of ALS and DHS are EBF, and the forest type of QYZ is ENF.

## 2.2 Model description

In this study, we used the BEPS model to simulate the subtropical forest GPP and net ecosystem productivity (NEP) with a resolution of 0.05°. The BEPS is a process-based model driven by the remotely sensed leaf area index (LAI), land cover types, soil, and meteorological data. Recently, it was used to simulate the terrestrial ecosystem carbon and water fluxes over different regions, such as the globe (Chen et al., 2019b; Chen et al., 2012), North America (Sprintsin et al., 2012; Xie et al., 2018), Europe (Wang et al., 2003), and East Asia (Matsushita and Tamura, 2002), as well as the whole or southern China (Liu et al., 2018; Liu et al., 2014; Peng et al., 2021). A more detailed description of the original BEPS can be found in Text S1 of the Supplementary section and previous studies (Chen et al., 2019b; Chen et al., 1999; Ju et al., 2006; Liu et al., 1999; Liu et al., 1997). In BEPS, daily GPP (gC m$^{-2}$day$^{-1}$) is calculated as (Chen et al., 1999):

$$GPP = GPP_{sun}LAI_{sun} + GPP_{shade}LAI_{shade} \qquad (1)$$

where GPP$_{sun}$ (g C m$^{-2}$day$^{-1}$) and GPP$_{shade}$ (g C m$^{-2}$day$^{-1}$) denote the GPP per unit area of sunlit and shaded leaves; LAI$_{sun}$ (m$^2$ m$^{-2}$) and LAI$_{shade}$ (m$^2$ m$^{-2}$) respectively represent the LAI of sunlit and shaded leaves. LAI$_{sun}$ and LAI$_{shade}$ depend on the mean solar zenith angle (θ, unitless):

$$LAI_{sun} = 2cos\theta \times (1 - exp(-0.5\Omega LAI/cos\theta)) \qquad (2)$$

$$LAI_{shade} = LAI - LAI_{sun} \qquad (3)$$

where LAI is the total canopy leaf area index (m$^2$ m$^{-2}$) and Ω is the clumping index (unitless).

In the BEPS model, maximum carboxylation rate V$_{cmax}$ (μmol m$^{-2}$ s$^{-1}$) is one of the most important and sensitive parameters for influencing the photosynthetic rate of plants and estimating carbon fluxes (Croft et al., 2017; Luo et al., 2019). $V_{cmax}$ mainly depends on V$_{cmax25}$ and air temperature (T$_a$, °C) in BEPS(see Text S1 of supplementary section (Eq. S4)). Generally, V$_{cmax25}$ is a commonly defined constant among different plant functional types (PFTs) in the model. However, V$_{cmax25}$ actually has large spatial variations (Table S2) due to changes in species composition, soil properties and climate within the same PFT, even observations showed a 2-3 fold variation in V$_{cmax25}$ for the same PFT (Chen et al., 2022b). As a result, using a PFT with fixed V$_{cmax25}$ in the model may distort the spatial distribution of the GPP simulation (Chen et al., 2022b). Therefore, in this study, we introduced a spatial variation of V$_{cmax25}$ derived from remote sensing data to replace the constant V$_{cmax25}$ in the original BEPS model. The other parameters, including the clumping index, maximum stomatal conductance, specific leaf area, respiration coefficient for leaves, stems, coarse and fine roots, Q10 for leaves, stems, and roots, etc., used in the BEPS model for each plant functional type can be found in Liu et al. (2018), which were specially parameterised for simulating the carbon fluxes of terrestrial ecosystems in China based on flux tower observations (Liu et al., 2013a; Liu et al., 2016; Liu et al., 2013b) and the published literature (Feng et al., 2007; Liu et al., 2015; Zhang et al., 2012).

### 2.3  Data and processing

**(1) Flux tower data**

To evaluate the models' performance, we acquired daily eddy covariance (EC)-derived GPP and NEP (net ecosystem productivity) from three flux tower sites over the study area (Fig. 1), which was available from the ChinaFLUX network (Yu et al., 2006). ChinaFLUX has undergone strict data quality control, including coordinate rotation, Webb-Pearman-Lenuing (WPL) correction and nighttime flux correction. For instance, the nighttime $CO_2$ flux correction mainly included removing outliers when there was precipitation, $CO_2$ concentration exceeded the instrument's measurement range, and there were fewer than 15,000 valid samples. Additionally, the u* threshold was also used to judge low flux values. For the QYZ and ALS stations, when the threshold of u* was below 0.2 m s$^{-1}$, the flux data was considered unreliable and was removed. However, the threshold of u* =0.05 m s$^{-1}$ was used for DHS station, and when u* was below 0.05 m s$^{-1}$, the flux data was rejected and removed. The NEE was also partitioned into gross ecosystem productivity (GEP) and ER with the method of Reichstein et al. (2005).

**(2) Remote sensing data**

**LAI**. The Global Land Surface Satellite (GLASS) LAI product from 2001 to 2018 was obtained from the University of Maryland. It was generated using the general regression neural networks (GRNNs) with a spatiotemporal resolution of 0.05° and 8-day (Xiao et al., 2016). The daily LAI at 0.05° resolution was obtained by linear interpolation of the 8-day GLASS LAI, which was used to drive the BEPS model (Wang et al., 2022). The GLASS LAI was used in this study because of its higher accuracy in China's forests compared to other satellite LAI products (Liu et al., 2018; Xie et al., 2019). For example, Liu et al. (2018) estimated the accuracy of different satellite-derived LAI products for the simulation of carbon and water fluxes in China's forests based on the BEPS model, and proved that GLASS LAI showed higher accuracy in simulating forest GPP than other LAI products (e.g., FSGOM LAI and MODIS LAI). These conclusions also have been reported consistently in other studies (Chen et al., 2021a; Jiang et al., 2017; Xie et al., 2019). Therefore, it was reasonable to use GLASS LAI as an input to model subtropical forest GPP in this study.

**Satellite-derived $V_{cmax25}$ products**. We obtained the spatial variation of satellite-derived $V_{cmax25}$ products from the National Ecosystem Science Data Center, National Science & Technology Infrastructure of China from 2001 to 2018, with a spatiotemporal resolution of 500 m and 8-day. We used the annual mean $V_{cmax25}$ for each pixel that varied from 2001 to 2018, and it was further resampled to 0.05°×0.05° for driving the model. $V_{cmax25}$ product was produced by satellite-derived leaf chlorophyll content (LCC) (Xu et al., 2022) and a semi-mechanistic model (Lu et al., 2022). It has been shown that this can effectively reduce the uncertainty in the simulations of the BEPS model (Lu et al., 2022; Lu et al., 2020; Wang et al., 2020b). More mechanisms for deriving $V_{cmax25}$ from remote sensing data are available in Lu et al. (2022), Luo et al. (2018), and Xu et al. (2022).

**Published GPP products**. To better estimate the model performance of the BEPS model, we also used five global GPP products generated by different methods to compare with our simulated GPP, which were further aggregated into 0.05°×0.05° for comparison. The five published GPP products include (a)

MODIS GPP (MOD17A2H Version 6) (Running et al., 2015), (b) EC-LUE GPP generated by a revised light use efficiency model (Zheng et al., 2020), (c) NIRv GPP produced by near-infrared reflectance ($NIR_V$) and machine learning (Wang et al., 2021b), (d) VPM GPP produced by the Vegetation Photosynthesis Model (VPM) (Zhang et al., 2017), and (e) another published BEPS GPP product (hereinafter referred to as $BEPS_g$ GPP), which was also generated by the BEPS model but with independently driven data and globally calibrated parameters (Chen et al., 2019b; He et al., 2021b). See Table S3 for more details on the five GPP products.

**(3) Climate data**

We obtained daily meteorological data including the temperature, precipitation, relative humidity, and downward solar radiation from the Climate Meteorological Forcing Dataset (CMFD) (He et al., 2020), and used them to drive the BEPS model. The CMFD is a high spatial (about 0.1°) and temporal (e.g., hourly and daily) resolution reanalysis product and covers the period of 1979-2018, which has been evaluated against the in-situ meteorological data (He et al., 2020) and was widely used in previous studies (Huang et al., 2021; Wang et al., 2020a; Yang et al., 2017a). To ensure consistency with the resolution of other driving data, the CMFD was also resampled to 0.05° based on the bilinear interpolation method.

**(4) Land cover data**

The annual land cover data sets from the European Space Agency (ESA) were used for simulations (ESA, 2017). ESA CCI land cover data has a resolution of 300 metres, spanning the 1992-present period. The overall global accuracy of CCI land cover data is 75.4%, with higher accuracy for forests (ESA, 2017). In this study, the original CCI land cover data were first aggregated into 0.05°×0.05° by using the CCI LC user tool. Considering the CCI land cover data comprised of 37 original vegetation classes, we referred to Tagesson et al., (2020) to reclassify the CCI land cover data into nine classes, including the evergreen broadleaved forest (EBF), evergreen needleaved forest (ENF), deciduous broadleaved forest (DBF), mixed forest (MF), cropland (CRO), grassland (GRA), shrubland (SHR), urban (URB), and barren land (BAR).

**(5) Soil and atmospheric $CO_2$ data**

The available water capacity (AWC) data with a spatial resolution of 0.05° were extracted from the re-gridded Harmonized World Soil Database (RHWSD) v1.2 (FAO, 2012; Wieder et al., 2014) and were used to drive the model in this study. We also obtained the annual mean atmospheric $CO_2$ concentration data (2001-2018) from the Hawaiian Mauna Loa observatory.

**2.4 Experiment design**

To understand the individual and combined effects of forest cover change, LAI change, $CO_2$ fertilisation, and climate change on annual subtropical forest GPP variations from 2001to 2018, we designed five groups of simulations in this study (Table 1). First, in scenario $S_{baseline}$, the model was run based on all the dynamic inputs during 2001-2018, including the dynamic land cover, LAI, $CO_2$, and all climate variables. In scenario $S_1$, we fixed the land cover in 2001 and allowed all other driven data to vary from 2001 to 2018. In scenario $S_2$, we conducted four different simulations to investigate how the

key climatic factors ($S_{2.1}$: precipitation; $S_{2.2}$: temperature; $S_{2.3}$: solar radiation) and all forms of climate change ($S_{2.4}$) influenced the subtropical forest GPP. We individually fixed the precipitation, temperature, solar radiation, and all climatic factors to the year 2001, while allowing all other factors (i.e., land cover, LAI, and $CO_2$) to change over time. In scenario $S_3$, the LAI was fixed at the level of 2001, and other factors were changed over time. In scenario $S_4$, we fixed $CO_2$ concentration (371.31 ppm) to 2001, with other drivers being dynamic. Finally, the differences between $S_{baseline}$ and the different scenarios were calculated for estimating the effect of different drivers on subtropical forest GPP changes.

**Table 1** Design of the scenarios for unravelling the effect of forest cover change, LAI change, $CO_2$ fertilisation, and climate change on subtropical forest GPP variations.

| Scenarios | | Land cover | LAI | Climate | Atmospheric $CO_2$ | Purpose |
|---|---|---|---|---|---|---|
| $S_{baseline}$ | | Dynamic | Dynamic | Dynamic | Dynamic | Estimating actual dynamics of GPP |
| $S_1$ | | Fixed in 2001 | Dynamic | Dynamic | Dynamic | Estimating the effect of forest cover change on GPP |
| $S_2$ | $S_{2.1}$ | Dynamic | Dynamic | Fixed in 2001 | Dynamic | Estimating the effect of precipitation on GPP |
| | $S_{2.2}$ | Dynamic | Dynamic | Fixed in 2001 | Dynamic | Estimating the effect of temperature on GPP |
| | $S_{2.3}$ | Dynamic | Dynamic | Fixed in 2001 | Dynamic | Estimating the effect of radiation on GPP |
| | $S_{2.4}$ | Dynamic | Dynamic | Fixed in 2001 | Dynamic | Estimating the effect of climate change on GPP |
| $S_3$ | | Dynamic | Fixed in 2001 | Dynamic | Dynamic | Estimating the effect of LAI on GPP |
| $S_4$ | | Dynamic | Dynamic | Dynamic | Fixed in 2001 | Estimating the effect of $CO_2$ fertilisation on GPP |

### 2.5 Statistical analysis

Three statistical metrics were used to assess the performance of the BEPS model in the simulation of GPP and NEP. They were the coefficient of determination ($R^2$), the root mean square error (RMSE), and the mean bias error (MBE).

The average values of 3×3 pixels centred around the flux sites (provided that these grid pixels had the same land cover type) were used to validate the predicted GPP and NEP (Peng et al., 2021; Wang et al., 2022). In addition, linear regression analysis was used to detect the long-term trend of the differences between the real and control experiments, which were considered as the impact of the controlled variables on the GPP changes.

Moreover, spatial correlation was adopted in this study to compare the spatial consistency of our simulated GPP with other GPP products. The spatial correlation was calculated pixel by pixel at the annual scale. First, two GPP time series for a certain pixel were obtained in the same period, and then the correlation between the two GPPs was calculated. Eventually, the spatial distribution of the correlation coefficients could be achieved.

## 3. Results

### 3.1 Model performance

We first compared the simulated daily GPP with the flux-site GPP (Fig. 2). The overall accuracy of GPP simulated by the BEPS model agreed well with measurements from the three flux sites (ALS: $R^2$ = 0.58, RMSE = 1.57 gC m$^{-2}$ day$^{-1}$ and MBE = 0.03 gC m$^{-2}$ day$^{-1}$; DHS: $R^2$ = 0.44, RMSE = 1.17 gC m$^{-2}$ day$^{-1}$ and MBE = 0.25 gC m$^{-2}$ day$^{-1}$; QYZ: $R^2$ = 0.77, RMSE = 1.36 gC m$^{-2}$ day$^{-1}$ and MBE = 0.05 gC m$^{-2}$ day$^{-1}$) (Fig. 2a-c). The BEPS model also showed good performance in simulating daily GPP each year (Table S4, Fig. S1-S3). For example, the $R^2$ ranged between 0.50 and 0.72 for ALS (2009-2013), between 0.43 and 0.65 for DHS (2003-2010) and between 0.70 and 0.85 for QYZ (2003-2010). Simulated GPP also captured both the absolute values and the inter-annual variability of observed annual GPP in the three flux sites (Fig. 2d-f). Compared with the annual measured GPP, the overall accuracy ($R^2$) of GPP simulated by the BEPS model was 0.89 (ALS), 0.53 (DHS) and 0.73 (QYZ), respectively (Fig. 2d-f). We further found that the BEPS model agreed reasonably well with measured daily NEP (Table S5, Fig. S4-S6). The overall accuracy ($R^2$) of simulated daily NEP was 0.25 (ALS), 0.35 (DHS) and 0.42 (QYZ), respectively (Table S5). In this study, we used the NEP for testing the model performance, because NEP (i.e., -NEE (net ecosystem exchange)) is a direct measurement of carbon fluxes between the atmosphere and ecosystems. Therefore, we not only used the observed GPP from the flux sites to validate our model, but also NEP. The validation of model performance based on measured NEP was relatively lower than that of GPP. One cause is that the simulation of NEP in the model was influenced not only by the accuracy of simulated GPP, but also by the accuracy of simulated heterotrophic respiration ($R_h$) and autotrophic respiration ($R_a$). We also compared the modelled annual NEP with the annual terrestrial sink from Global Carbon Budget (GCB) 2023 (outputs from multi process-based models) (Friedlingstein et al., 2023). The modelled NEP and GCB showed good consistency over the 2001-2018 period in terms of the trends (0.09 Pg C/year vs. 0.07 Pg C/year) (Fig. S7a) and Pearson's coefficient ($R^2$ = 0.46, p < 0.05) (Fig. S7b). Furthermore, we simulated the annual net primary productivity (NPP) based on our model, and obtained 33 measured subtropical forest NPP values from the published literature to validate the simulated NPP (Table S6 and Fig. S8). The results also confirmed that our model performed well in simulating NPP ($R^2$ = 0.62, $p$ < 0.001) (Fig. S8).

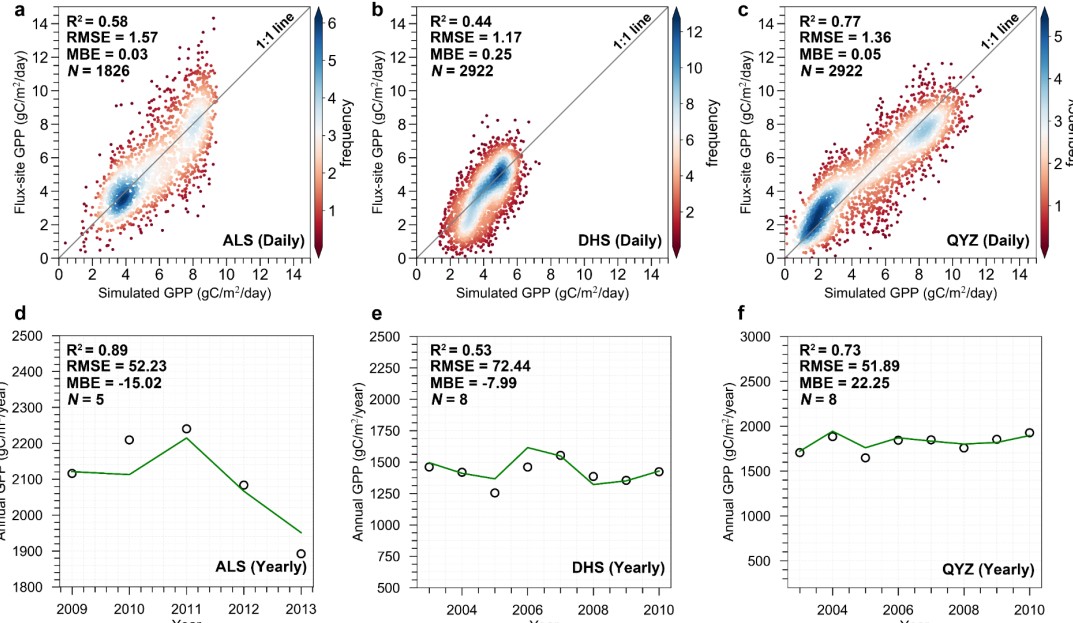

**Figure 2.** Comparison of simulated GPP with measured GPP from three flux tower stations at daily (a-c) and annual (d-f) scales. The green lines and dark circles represent the simulated GPP and observed GPP, respectively. There may be relatively low-quality issues with observed flux data from DHS, which may affect our validation results. For example, as reported by Wang et al., (2006), the low observed

values of $CO_2$ flux are mainly caused by a $CO_2$ leak during the nighttime at the DHS station. In addition, the effect of topography also led to generally low fluxes in the southerly direction at DHS site (Li et al., 2021).

At the regional level, our BEPS model captured the spatial gradient in GPP well when compared with the other GPP products (Fig. S9). The mean $R^2$ values between our simulated GPP and NIRv GPP,

EC-LUE GPP, MODIS GPP, $BEPS_g$ GPP, and VPM GPP were 0.52, 0.67, 0.41, 0.54 and 0.41, respectively (Fig. s10f). The simulated GPP was specially consistent with the spatial pattern of the EC-LUE GPP (Fig. S10). In nearly 67% and 34% of forest areas, the $R^2$ was higher than 0.6 and 0.8, respectively. Additionally, we compared the multi-year mean of annual total GPP in our study with that of other GPP products throughout the entire forest and different forest types (Fig. S11). The multi-year

mean of annual total GPP for the entire forest area in our study was $2.23 \pm 0.14$ PgC year$^{-1}$, close to the magnitudes of the three GPP products (i.e., $BEPS_g$ GPP product: $2.54 \pm 0.16$ PgC year$^{-1}$; MODIS GPP: $2.10 \pm 0.07$ PgC year$^{-1}$; VPM GPP: $2.05 \pm 0.10$ PgC year$^{-1}$) and the mean of the five GPP products ($2.07 \pm 0.11$ PgC year$^{-1}$), respectively (Fig. S11). Meanwhile, for the entire and different forests, the annual GPP of this study and other GPP products also showed a similar increasing trend (Fig. S11f-j).

For example, the trend of DBF and MXF in this study was close to the VPM GPP and the EC-LUE GPP (Fig. S11h, Fig. S11j). Although our simulated GPP is slightly higher for the entirety of subtropical forests, EBF and ENF as compared to other GPP products, it is very close to other GPP products for specific forest types such as DBF and MXF (Fig. S11). Similarly, these commonly used GPP products also have large differences when compared to each other (Fig. S11). These results indicate there is still a

large discrepancy in modelling GPP to date, due to many differences in model structure, parameterisation,

and driving data. For example, the MODIS GPP, EC-LUE GPP and VPM GPP were simulated by different light use efficiency (LUE) models. However, most current LUE-based models do not completely integrate some key environmental regulations into vegetation productivity, such as the effect of atmospheric $CO_2$ concentration, which may result in underestimation. In this study, GPP was simulated by a process-based model (i.e., BEPS) that considered the $CO_2$ fertilisation effect, which may lead to a higher GPP compared to other GPP products.

### 3.2 Spatiotemporal variations of the subtropical forest GPP

Based on the scenario $S_{baseline}$ (Table 1), the simulated forest GPP showed a significant increasing trend (20.67 gC/m²/year, $p$ = 0.000) during 2001-2018 over the entirety of subtropical forests due to the interactive effect of different drivers (Fig. 3a). Among the four forest types, EBF showed the largest significant increasing trend (28.24 gC/m²/year, $p$ = 0.000), followed by DBF (20.68 gC/m²/year, $p$ = 0.000), MXF (16.12 gC/m²/year, $p$ = 0.000) and ENF (15.20 gC/m²/year, $p$ = 0.000). Spatially, 90.4% of forested land in the study area showed an increasing trend in GPP, while 9.6% of forested land exhibited a decreasing trend in GPP (Fig. 3b). The areas with significantly increased and decreased GPP accounted for 70.1% and 2.6% of the entire subtropical forest area, respectively (Fig. 3b).

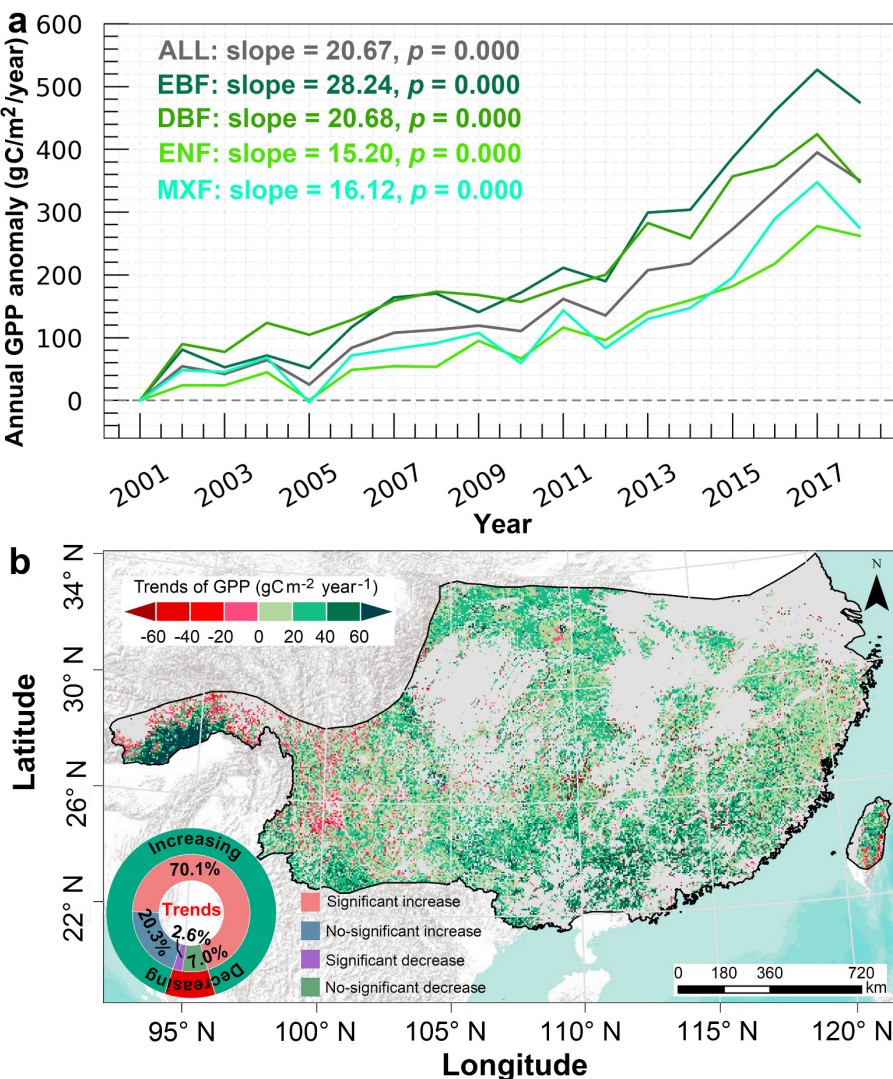

**Figure 3.** (a) Temporal variations of the annual subtropical forest GPP anomaly during 2001-2018 The

annual GPP anomaly is calculated relative to the base year of 2001; (b) Spatial distribution of the annual trends in actual GPP. Light grey within the study area indicates non-forested areas.

### 3.3 Disentangling the effects of driving factors on subtropical forest GPP changes

### 3.3.1 Impacts of different driving factors on subtropical forest GPP changes

We investigated the area of gains or losses for different subtropical forest types between 2001 and 2018 using the ESA CCI land cover data (Fig. S12). We found that FCC increased the entire subtropical forest GPP at a rate of 0.52 gC/m$^2$/year (p = 0.000) (Fig. 4a), and the increase was mainly driven by EBF GPP (0.39 gC/m$^2$/year, p = 0.011) and MXF GPP (1.14 gC/m$^2$/year, p = 0.000). However, FCC had a negative effect on the DBF GPP and ENF GPP variations at the rate of -0.06 gC/m$^2$/year (p = 0.632) and -0.19 gC/m$^2$/year (p = 0.002), respectively. Spatially, 92.2% of the total GPP was relatively stable, and only 7.8% of GPP exhibited an increase or decrease under the effect of FCC (Fig. 4b). Among it, 3.9% of the GPP increased significantly and the increases were mainly located in the southwest and northern regions (e.g., the south slope of the Qinling mountains, the southwest karst region), while 2.6% of GPP was significantly reduced in the eastern regions where the ENF is dominated (Fig. 4b).

The annual total precipitation and annual mean temperature over the entire forest region and different forest areas showed an increasing trend, while the annual total radiation displayed a decreasing trend for the entire forest region and different forest areas (Fig. S13). The individual effects of precipitation, temperature, and solar radiation on subtropical forest GPP were first investigated in Fig. S14, and their combined effects on GPP changes were shown in Fig. 4c-d. The results showed that climate change increased GPP across the entire forest area (0.92 gC/m$^2$/year, $p$ = 0.080), with a significant increase in the GPP of EBF (3.83 gC/m$^2$/year, $p$ = 0.000) and DBF (2.49 gC/m$^2$/year, $p$ = 0.003), while climate change decreased the GPP of ENF (-1.22 gC/m$^2$/year, $p$ = 0.016) and MXF (-1.23 gC/m$^2$/year, $p$ = 0.075) (Fig. 4c). Spatially, 10.3% and 19.1% of the study area exhibited a significant upward trend and downward trend (Fig. 4d), respectively, due to the effect of climate change.

The LAI of entire and different forests showed significant upward trends during the study period (Fig. S15). The simulations showed that LAI exerted a significant positive effect of 3.79 gC/m$^2$/year ($p$ = 0.004) in the entire forest region (Fig. 4e), confirming the positive role of LAI in subtropical forest GPP variations. There was significant spatial heterogeneity in the effect of LAI on GPP changes (Fig. 4f). A significant ($p < 0.05$) positive effect of LAI on GPP was observed over 29.9% of the study area, mainly located in the south and north (Fig. 4f). The areas with a significant decreasing trend ($p < 0.05$) accounted for 6.0% and are mainly distributed in the western and central parts of the study area (Fig. 4f). There are more positive changes in GPP due to the effect of LAI that heavily offsets the negative changes in GPP, ultimately making LAI the second dominant factor in GPP increases throughout China's subtropical forests.

The annual mean $CO_2$ concentration increased from 371.3 ppm to 408.7 ppm during 2001-2018 (Fig. S16), which led to a significant increase of all subtropical forest GPP at the rate of 6.84 gC/m$^2$/year ($p$ = 0.000) (Fig. 4g). The significantly positive effects of $CO_2$ fertilisation on EBF GPP (6.91 gC/m$^2$/year, $p$ = 0.000) and ENF GPP (7.02 gC/m$^2$/year, $p$ = 0.000) was higher than that of DBF GPP (5.93 gC/m$^2$/year, $p$ = 0.000) and MXF GPP (6.66 gC/m$^2$/year, $p$ = 0.000). $CO_2$ fertilisation showed significant positive effects on GPP in almost all of China's subtropical forests (nearly accounting for 99.48% of the total

forest area) (Fig. 4h), suggesting the high sensitivity of forests in this area to elevated $CO_2$ concentrations.

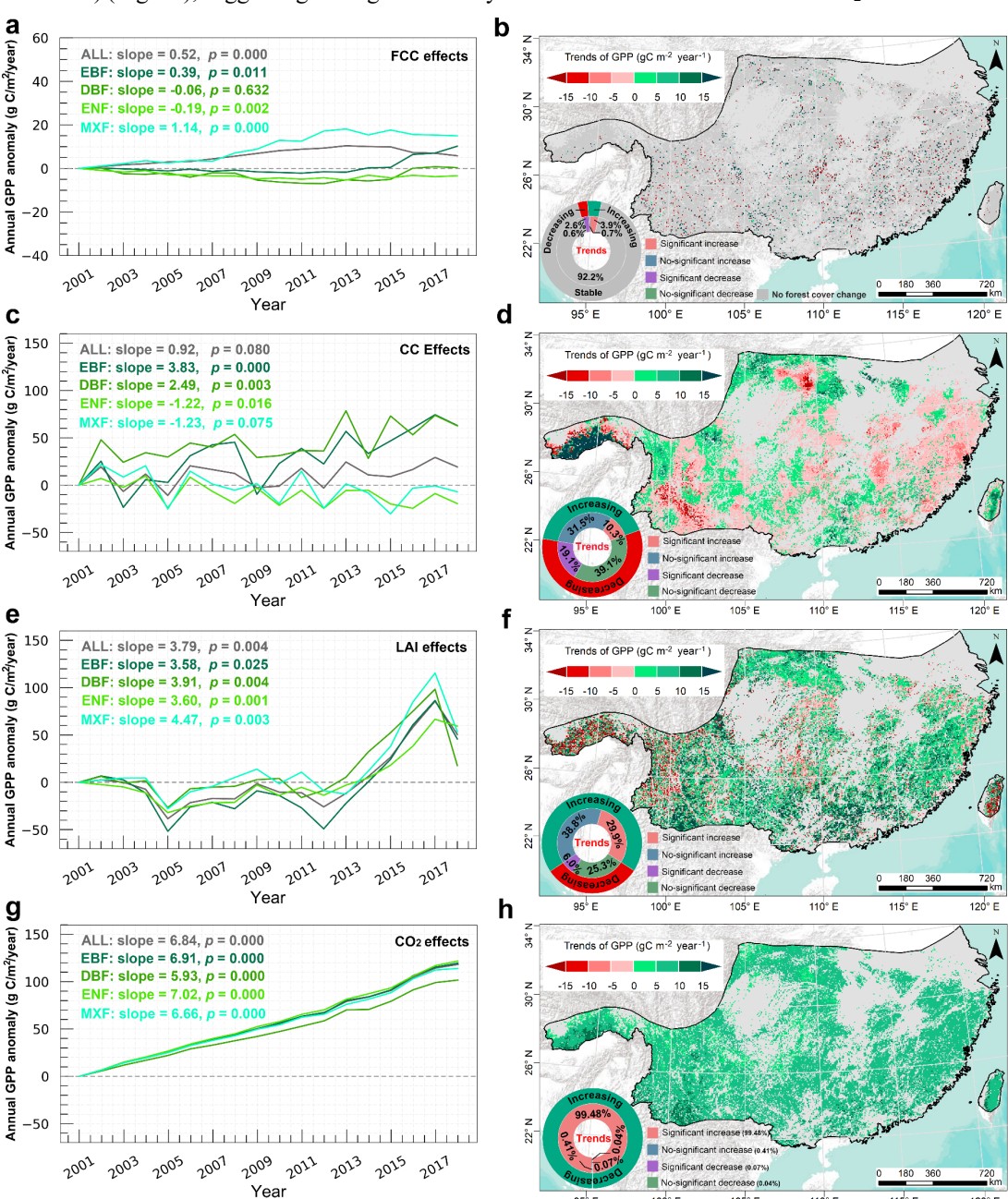

**Figure 4.** Temporal variation of the effects of FCC (a), CC (c), LAI (e), and rising $CO_2$ concentration (g) on annual subtropical forest GPP trends. Spatial distribution of the impacts of FCC (b), CC (d), LAI (f), and rising $CO_2$ concentration (h) on subtropical forest GPP. Light grey in the study area indicates non-forested areas.

### 3.3.2 Comparison of the effects among FCC, CC, LAI, and $CO_2$ fertilisation and the dominant drivers

We compared how different drivers contribute to annual trends in GPP of different subtropical forests (Fig. 5). For all forests together, the enhanced $CO_2$ concentration made the largest contribution to the overall GPP enhancement, followed by LAI, CC and FCC (Fig. 5a). In addition to the $CO_2$ fertilisation effect, LAI was another dominant contributor to subtropical forest GPP increase across the

entire and different forest types (Fig. 5), especially the positive effect of LAI almost counteracts the negative effect of forest cover change and climate change on ENF GPP. The forest cover change mainly contributed to MXF GPP increase (Fig. 5e), but also resulted in an ENF GPP decrease (Fig. 5d). Climate change increased the GPP (EBF and DBF) of broad-leaved forests (Fig. 5b and 5c), but it decreased ENF GPP and MXF GPP (Fig. 5d and 5e). Overall, the GPP of EBF in the subtropical regions of China experienced the largest annual growth rate (Fig. 5b) when compared with other forest types, and changes in GPP responses to different drivers depend on forest types.

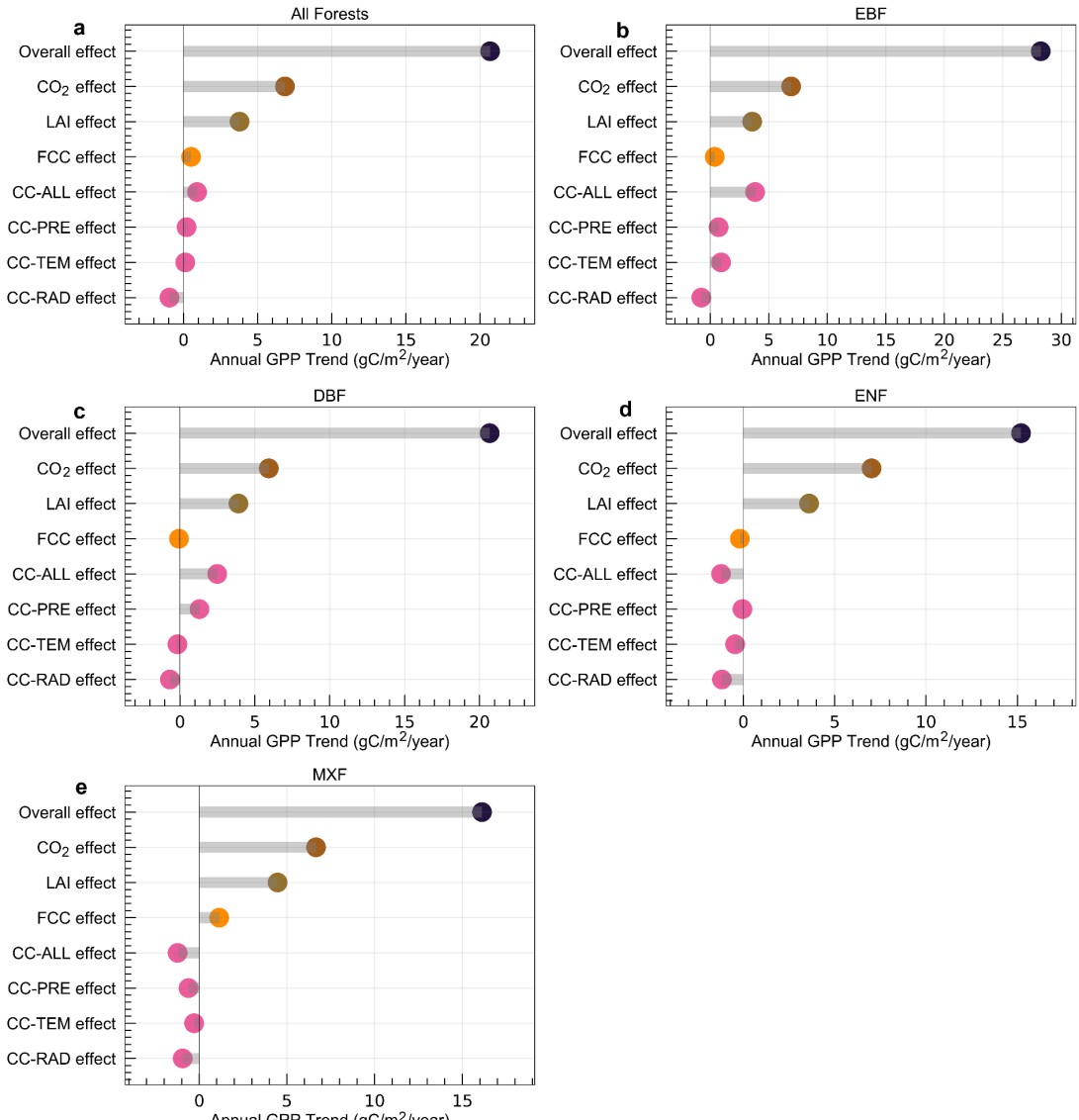

**Figure 5.** Comparison of different trend drivers in GPP for entire (a) and different forests (b-e). The overall effect denotes the combined effect of all driving factors; the LAI effect indicates the impact of LAI change on subtropical forest GPP. FCC effect indicates the effect of forest cover change on GPP; CC-ALL, CC-PRE, CC-TEM, and CC-RAD represent the impacts of all climatic factors including, precipitation, temperature, and solar radiation on subtropical forest GPP variations.

We also investigated the spatial distribution of the effects of dominant factors on subtropical forest GPP changes at each grid cell level (Fig. 6). The results showed that the dominant factors affecting the

subtropical forest GPP varied greatly in space (Fig. 6). The $CO_2$ fertilisation (41.7%) and LAI change (35.7%) were the two dominant factors of subtropical forest GPP changes in most regions (Fig. 6). However, CC (8.9%) was the dominant factor driving subtropical forest GPP to increase in the western and northern mountainous areas, and FCC (4.6%) was the dominant driver of subtropical forest GPP decrease in the east.

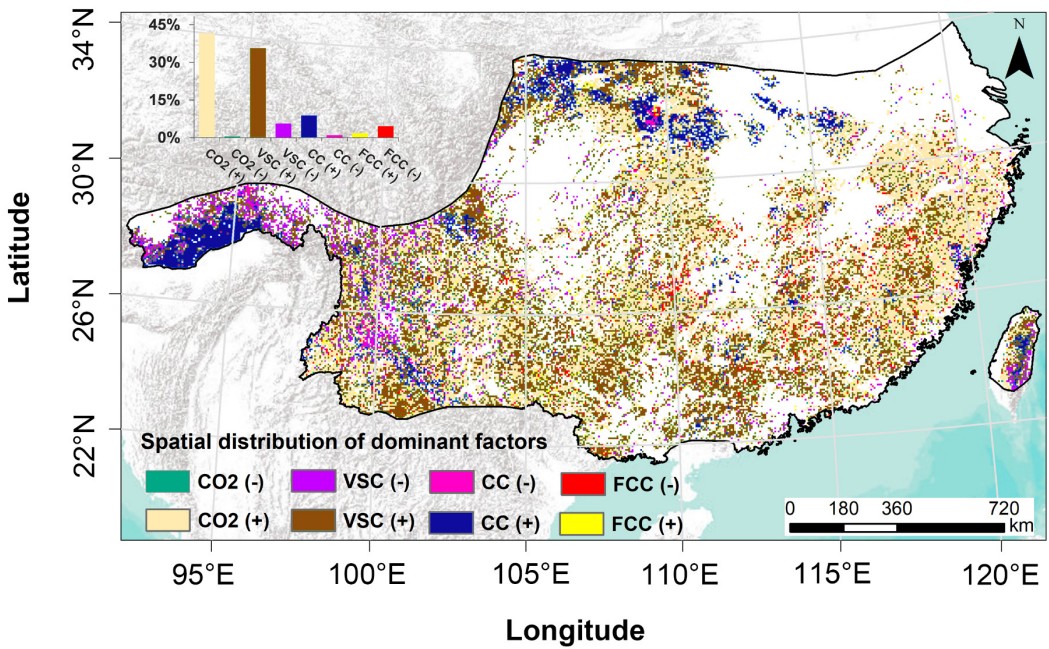

**Figure 6.** Spatial distribution of the effects of dominant factors on subtropical forest GPP changes. The symbols (+) and (−) denote the positive and negative effects of these factors on GPP trends, respectively.

## 4. Discussion

### 4.1 The effects of the FCC, CC, LAI, and $CO_2$ fertilisation on subtropical forest GPP variation

Overall, the GPP in both the entire forest region and different forest types displayed an increasing trend over the past two decades (Fig. 3), which is in line with many previous findings (Chen et al., 2021b; He et al., 2019; Li et al., 2022; Tong et al., 2018). However, there were obvious differences among these factors that contribute to the increase in GPP among subtropical forests. The $CO_2$ fertilisation effect (6.84 gC/m²/year, $p < 0.001$) and LAI change (3.79 gC/m²/year, $p = 0.004$) were the two dominant drivers of total subtropical forest GPP increase, followed by the effect of FCC (0.52 gC/m²/year, $p < 0.001$) and CC (0.92 gC/m²/year, $p = 0.080$). We also calculated the contributions of different factors to the total GPP of the study area, and also found that the $CO_2$ fertilisation effect (8.23 TgC/year, p<0.001) and LAI (4.55 TgC/year, p=0.005) contributed more to the increase in the total GPP of subtropical forests than that of FCC (1.35 TgC/year, p<0.001) and CC (1.11 TgC/year, p=0.08).

### 4.1.1 The effect of FCC on subtropical forest GPP

In the past two decades, the Chinese government has made an enormous investment to implement some key ecological restoration programs to improve the forest areas, such as the Grain for Green Program (GGP, initiated in 2000) and the Yangtze and Pearl River Basin Shelterbelt programs (Viña et al., 2016; Zhang et al., 2022). Nationwide field samplings confirmed the increase of vegetation cover

and carbon sink via these ecological projects since the end of the 20[th] century (Lu et al., 2018). Especially, the forest restoration hotspots were observed on the south slope of the Qinling Mountains (Chen et al., 2021b) and the southwest karst region (Tong et al., 2018) of China. In similar regions, we also observed the positive effect of FCC on GPP (Fig. 4a-4b). This is due to the increase in the total area of EBF and MXF (Fig. 4a), which is mainly converted from cropland (Table S7). For example, our statistics showed that, before conversion, the regional average GPP of cropland in 2001 was 1732.40 g C/m$^2$/year, whereas after the cropland was converted to EBF, the regional average GPP was 1935.69 g C/m$^2$/year in 2018, an increase of 203.29 g C/m$^2$/year.

Previous studies (Chen et al., 2021a; Chen et al., 2021b; Zhang et al., 2022) usually considered different forests in China as a single forest type, which may ignore the different effects of a specific forest type on forest GPP variations. In this study, we identified the positive effect (0.52 gC/m$^2$/ year) of FCC on GPP for all subtropical forest types together. However, contradictions with previous results were also found. The total area of ENF was reduced obviously during the study period in eastern and southern regions, and most of it was converted to MXF (19,040 km$^2$) and cropland (13,100 km$^2$) (Table S7). Here, we further counted the changes in GPP caused by conversion between ENF and MXF and cropland, and found that the decrease in ENF GPP of 268.65 g C/m$^2$/year due to the conversion between ENF and cropland (i.e., ENF = 2120.51 g C/m$^2$/year in 2001; ENF = 1851.86 g C/m$^2$/year in 2018) was greater than the increase in the ENF GPP of 141.24 g C/m$^2$/year due to the conversion between ENF and MXF (i.e., ENF = 1436.65 g C/m$^2$/year in 2001; ENF = 1577.89 g C/m$^2$/year in 2018), ultimately resulting in a slight decrease in ENF GPP (Fig. 4a). This side effect may be overlooked if different forest types are not considered. For example, the reduction in ENF GPP (-0.19 gC/m$^2$/year) mainly located in the eastern and southern regions was offset by the GPP of EBF and MXF (total: 1.53 gC/m$^2$/year) in most regions (Fig. 4a-b). Therefore, under the influence of FCC, the entire subtropical forest GPP showed an increasing trend (0.52 gC/m$^2$/year) (Fig. 4a).

### 4.1.2 The effect of CC on subtropical forest GPP

Under the combined effect of all climatic factors, an overall increase (0.92 gC/m$^2$/year) in subtropical forest GPP was observed in the study area (Fig. 4c). However, different climatic factors play different roles in regulating subtropical forest GPP changes (Fig. S14). Precipitation increased the whole subtropical forest GPP (0.21 gC/m$^2$/year) (Fig. S14a), especially in the northern and western mountains (Fig. S14b). This is because the slight increase in precipitation in these areas, without exceeding a certain threshold, can increase soil water availability and alleviate drought stress on forest growth, thereby facilitating forest photosynthesis and enhancing GPP (He et al., 2019; Li et al., 2022). Temperature is another complex driver of forest GPP variation. Many studies suggested that an increase in temperature can benefit vegetation productivity (Myneni, et al., 1997; Nemani, et al., 2003; Song et al., 2022), or could reduce vegetation productivity due to increased VPD as a result of a high temperature increase (Yuan et al., 2019; Lopez et al., 2021). Our findings also proved that the effect of temperature on subtropical forest GPP varied spatially (Fig. S14d). Most of the region (59.7%) experienced a decline in subtropical forest GPP due to the effects of climate warming, while 40.3% of subtropical forest GPP mainly located in the western mountains displayed a significant upward trend (Fig. S14d). This is because

the increase in temperature in mountainous areas can extend the growing season and enhance photosynthesis (Nemani et al., 2003; Piao et al., 2005; Zhang et al., 2014), thereby improving subtropical forest GPP. The magnitude of GPP increase in these small areas is significantly higher than in other regions, because temperature, precipitation, and radiation all contribute to an increase in GPP in these areas (Fig. S14). Although the area of GPP reduction due to temperature is relatively large, the magnitude of the impact is relatively small, resulting in smaller areas with higher magnitude offsetting the larger areas of GPP decrease. On the contrary, solar radiation in this study showed a downward trend (Fig. S13e-f). As a direct limiting factor of vegetation growth, the reduction of solar radiation can directly affect forest photosynthesis, thus reducing subtropical forest GPP. As expected, solar radiation in this study declined GPP over 67.2% of the total area (Fig. S14f), which may be associated with the recent increase in air pollution in China (Chen et al., 2021a; Zhang et al., 2014). The combined effects of these climatic factors caused a positive effect (0.92 $gC/m^2$/year) on the entirety of studied subtropical forest GPP, while different forest types showed different responses to climate change (Fig. 4c). For example, climate change has a positive effect on the GPP of EBF, but a negative effect on the GPP of ENF. The main reason is that ENF is predominantly located in the eastern and western parts of the subtropics (Fig. 1). In these areas, individual climatic factors (e.g., temperature, precipitation, and solar radiation) or their interactions caused the GPP of ENF to decrease (Fig. 4c-4d). In particularly, solar radiation declined significantly in the eastern region, which led to a decrease in GPP. EBF is primarily distributed in the central and western regions (Fig. 1) where climate change mainly contributes to the increase of EBF GPP (Fig. 4c-4d). Overall, future measures to combat climate change should consider different forest types and their geographical locations.

## 4.1.3 The effect of LAI on subtropical forest GPP

As the most important proxy of growth and structural change in vegetation (Chen et al., 2019b; Chen et al., 2021a), LAI can significantly influence the carbon cycle. Since the 2000s, some key forest protection programs, including the Natural Forest Protection Project (NFPP, initiated in 1998), have been carried out in the subtropical region of China (Chen et al., 2020). Due to forest protection and reasonable forest use and management with the support of ecological engineering, forest natural growth has increased the LAI (Chen et al., 2020) and further contributed to the GPP increase in China (Tong et al., 2018). A recent study showed that land-use management in China, especially forest management, has contributed significantly to earth greening, accounting for 25% of the increase in global LAI (Chen et al., 2019a). Chen et al. (2019b) estimated the effect of structural change in vegetation using LAI on the global terrestrial carbon sink since the 1980s, and confirmed that LAI significantly improved global carbon uptake. Especially, the LAI also promoted the forest carbon sink in China's subtropical region, but the contribution of different changes in forest LAI to GPP was not revealed. Evidence from our study demonstrated that LAI is the second dominant contributor (3.79 $gC/m^2$/year) to the increase in GPP of the entirety of subtropical forests (Fig. 4e), and also identified the MXF as the main contributor to the positive effect of LAI on GPP changes. Recently, although some studies have demonstrated the positive effects of LAI on forest carbon sequestration in China (Chen et al., 2019b; Chen et al., 2020; Zhang et al., 2022), these studies did not isolate the independent effects of LAI on different forest GPP. Currently, some ecological projects in China are aimed at protecting forests, others are aimed at increasing forest

area. It has been long debated how different ecological projects impact ecosystem services in carbon sequestration (Chen et al., 2020; Yin and Yin, 2010; Yu et al., 2011). In this study, we designed an experiment to examine the individual impact of LAI (i.e., mainly reflecting forest structure change) on subtropical forest GPP changes. The results showed that forest LAI change more than forest cover change positively impacted GPP increases in the study area (Fig. 4, Fig. 5), indicating that forest protection projects in the subtropical region of China may have greater carbon uptake potential. Consistent with our study period (2001–2018), Chen et al. (2021b) also reported an increase in vegetation carbon sequestration in China based on the two indicators of GPP and NPP, especially with an accelerated increase in carbon sequestration potential after 2010. They showed that GPP and NPP in China increased obviously at the rate of 49.1–53.1 $TgC/yr^2$ and 22.4–24.9 $TgC/yr^2$, respectively. The significant increase of subtropical forest GPP and NPP was highly attributed to human activities (e.g., ecological restoration projects) in southern and eastern China. Human-induced NPP gains can especially offset climate-induced NPP losses in southern China.

### 4.1.4 The effect of $CO_2$ fertilisation on subtropical forest GPP

Our results also suggested that $CO_2$ fertilisation was the major contributor to overall forest GPP increase in China's subtropical region (6.84 $gC/m^2/year$) (Fig. 4g and Fig. 5). Elevated $CO_2$ concentrations can enrich intercellular $CO_2$ and stimulate vegetation photosynthetic rates, thereby enhancing vegetation productivity. Recent studies suggested that the $CO_2$ fertilisation effect was the main driver in promoting global or regional vegetation productivity (Chen et al., 2022a; Chen et al., 2019b; Schimel et al., 2015; Xie et al., 2020). This was also confirmed by the results of free-air $CO_2$ enrichment (FACE) experiments (Norby et al., 2010) and a previous study using terrestrial biosphere models, remote sensing-based methods, ecological optimality theory, and an emergent constraint based on global carbon budget estimates (Keenan et al., 2023). Due to the inherent differences in the driving factors, it should be noted that the contribution of the $CO_2$ fertilisation effect to subtropical forest GPP changes mostly originates from the long-term trend of $CO_2$. However, the trend of climatic factors during the study period is not significant (Figure S9). The temporal attribution of climate to GPP is mainly due to its variability.

Moreover, how much the net terrestrial carbon uptake increases in response to rising in atmospheric $CO_2$ is not just dependent on GPP but also on the processes like respiration, mortality, longevity, etc. For example, the increase in forest GPP due to $CO_2$ fertilisation leads to increased tree growth, and the final decomposition of the increased plant matter improves litter and soil organic matter pools, thereby enhancing heterotrophic respiration (Rh) (Quetin et al., 2023). Therefore, the $CO_2$ fertilisation effect can be counteracted by respiration. To date, there is no consensus on the response of photosynthesis and respiration to long-term increases in $CO_2$, due to the magnitude of such an impact and associated mechanisms still remaining uncertain (Sun et al., 2023). While several studies found the simultaneous reduction of respiration at elevated $CO_2$ (Sun et al., 2023.; Hamilton et al., 2001). The opposite conclusion has also been reported (Chen, Y et al., 2022; Crous et al., 2012). Additionally, the effect of elevated atmospheric $CO_2$ on GPP is also related to tree mortality. For example, elevated atmospheric $CO_2$ concentrations can lead to faster tree growth and decreasing the carbon turnover time. Consequently, the acceleration of the tree's life cycle and death will reduce carbon sequestration (Needham et al., 2020). Besides, the $CO_2$ fertilisation effect on forest carbon sinks can be limited by

longevity. For example, Jiang et al., (2020) examined the responses of mature forests to atmospheric $CO_2$

enrichment. They found that elevated $CO_2$ led to a 12% increase in carbon uptake through GPP, but the carbon sequestration had not increased, and most of the carbon was returned to the atmosphere through respiration (Jiang et al., 2020). Currently, the forests in China are characterized by relatively young stand age (< 40 years old) due to a large number of new plantations, and thus China's forest carbon sequestration potential may continue to increase in the near future due to rising $CO_2$ concentration (Yao

et al., 2018a). However, as the trend of increasing atmospheric $CO_2$ concentration may slow down, the carbon sink potential of China's forests may be further reduced in the future due to the weakening of the $CO_2$ fertilisation effect.

### 4.2 Model and Uncertainties

In the BEPS model, LAI is separated into two parts including the LAI of sunlit and shaded leaves,

which are adopted to calculate photosynthesis at the leaf level based on the FvCB photosynthesis model (Farquhar et al., 1980), and further use those results to compute GPP at the canopy level by adding the photosynthesis rates of sunlit and shaded leaves. Moreover, the Ball-Berry equation (Ball et al., 1987) was used in the model to calculate the stomatal conductance of sunlit and shaded leaves, which influence intercellular $CO_2$, photosynthesis rates, and evapotranspiration (ET). Therefore, LAI directly determined

the allocation of light and water availability and influenced the gross photosynthesis rate of sunlit and shaded leaves. The accuracy of LAI may impact its contribution to GPP variations through these processes. Atmospheric $CO_2$ concentrations affect intercellular $CO_2$ through stomatal conductance, which, together with temperature and maximum carboxylation rate ($V_{cmax}$), determine the Rubisco-limited ($A_c$) and RuBP-limited ($A_j$) gross photosynthesis rate in the model. Over the past few decades,

$CO_2$ concentrations continuously have increased and reached the current level of over 400 ppm. Elevated atmospheric $CO_2$ concentrations can increase photosynthesis by accelerating the rate of carboxylation, thereby influencing the GPP changes. Additionally, solar radiation variability would directly influence the potential electron transport rate and thus regulate the RuBP-limited ($A_j$) gross photosynthesis rate. Temperature in the model directly impacts the $V_{cmax}$ and the $CO_2$ compensation point without dark

respiration ($\Gamma$), thereby determining the gross photosynthesis rate. Temperature positively affects the $V_{cmax}$ when it is below optimal. However, when temperature exceeds its optimum, $V_{cmax}$ will not continue to increase with the temperature. Therefore, temperature changes in the model may have a positive or negative impact on GPP.

It should be noted that changes in LAI could be influenced by both climatic factors and elevated

atmospheric $CO_2$ concentrations (Chen et al., 2019; Chen et al., 2021a; Sun et al., 2022). Previous studies reported that elevated atmospheric $CO_2$ concentrations were the dominant driver of global LAI increase, and there are also regional differences in the impact mechanism of climate factors on LAI changes (Zhu et al., 2016; Zhu et al., 2017), thereby influencing GPP dynamics. Moreover, the interactions between these driving factors can further influence LAI, and even the interactive impacts of these factors on LAI

may offset each other. For instance, rising $CO_2$ concentrations and solar radiation can affect temperature and VPD (Chen et al., 2021a). High VPD leads plants to close their stomata, resulting in lower intercellular $CO_2$ concentrations in the leaves, which reduces the rate of photosynthesis (Yuan et al.,

2019). Additionally, changes in LAI can feed back to the climate through biogeochemical and biogeophysical processes (Li et al., 2023). There is a bidirectional interaction between vegetation and the atmosphere, and the relationship between vegetation dynamics and driving factors is complex. The current methods used in this study cannot elucidate the complex interactions of climate factors and elevated $CO_2$ concentrations on LAI changes, which may bring some uncertainties to our results.

We first used the $V_{cmax25}$ product retrieved from remote sensing data (i.e., leaf chlorophyll content) to replace the constant value of the $V_{cmax25}$ in the model. Wang et al. (2019), Luo et al. (2018) and Croft et al. (2017) indicated that the use of remotely sensed leaf chlorophyll content to invert $V_{cmax25}$ can improve the accuracy of GPP simulation in evergreen conifer forests and a temperate deciduous forest. Our results suggested that the BEPS model with spatial varying $V_{cmax25}$ values can also reach a reasonable simulation of subtropical forest GPP over spatiotemporal scales (Fig. 2, Fig. S1-S6). Incorporating the spatial variation of the $V_{cmax25}$ inverted by remotely sensed data into the process-based model does not require its pre-calibration (Chen et al., 2022b), thus it has great potential to be applied to areas with few flux sites, such as China's subtropical forest area. However, the $V_{cmax25}$ retrieved from remote sensing data is still in the early developing stage (Chen et al., 2022b; Luo et al., 2019). For example, the $V_{cmax25}$ product used in this study was mainly generated by MODIS surface reflectance, and thus the data quality of the surface reflectance may cause uncertainty in the $V_{cmax25}$ product. The uncertainties in MODIS reflectance datasets can arise from sensor calibration issues, cloud contamination, atmospheric correction errors, etc. Changes in the reflectance could also result in large changes in the modelled chlorophyll values, thereby affecting the $V_{cmax25}$ product. Additionally, the $V_{cmax25}$ was produced by a semi-mechanistic model (Friend., 1995), and the key parameter $K_{cat}^{25}$ (i.e., the Rubisco turnover rate at 25 ˚C) in the model would bring uncertainties in modelling $V_{cmax25}$, because current ground-based data are still rarely used for calibration of this parameter and validation of the $V_{cmax25}$ products (Lu et al., 2022; Chen et al., 2022b).

## 5. Conclusions

In this study, the BEPS model was used to simulate the subtropical forest GPP, and the performance of the BEPS model in simulating subtropical forest GPP was examined. A significant increasing trend (20.67 gC/m$^2$/year, $p < 0.001$) was detected in the subtropical forest GPP over the past two decades, indicating that there is sustained increase in the carbon sink potential of subtropical forests under the background of global change, with evergreen broad-leaved forests (EBF) being the biggest contributor (28.24 gC/m$^2$/year, $p < 0.001$) to total GPP enhancement of the entire subtropical forests. We designed different groups of simulations to examine the individual and combined impacts of forest cover change (FCC), climate change (CC), leaf area index (LAI), and $CO_2$ fertilisation on inter-annual trends in subtropical forest GPP. There are obvious differences in drivers of different subtropical forest GPP variations.

Although the $CO_2$ fertilisation effect was the largest contributor to the overall subtropical forest GPP increase, LAI was another most important and not negligible contributor to subtropical forest GPP growth in China. FCC contributed to the mixed forest (MXF) GPP (1.14 gC/m$^2$/year, $p < 0.001$) and EBF GPP (0.39 gC/m$^2$/year, $p < 0.001$) increase, but induced the evergreen needle-leaved forest (ENF) GPP

to decrease (-0.19 gC/m$^2$/year, $p < 0.001$). CC increased the EBF and deciduous broadleaved forest (DBF) GPP, but decreased ENF and MXF GPP. Especially, the EBF and DBF GPP in this region are very sensitive ($p < 0.05$) to CC. Therefore, we emphasised that the mitigation of climate change and carbon emissions through forests should consider their different types. Furthermore, our results highlighted that the LAI effect, which was greater than that of FCC, was the most important driver of subtropical forest GPP enhancement, suggesting that forest use and management have a more significant positive impact on GPP increase than forest cover change in the study area. It may be attributed to the implementation of China's forest protection and restoration programs.

**Acknowledgments**

This work is jointly supported by the National Natural Science Foundation of China (Grant No. 42171025), the Fonds Wetenschappelijk Onderzoek (FWO Grant n° G018319N), and the program of the China Scholarships Council (Grant No. 202106380124).

**Data Availability statement**

We obtained the flux tower data from the ChinaFLUX network (http://www.chinaflux.org/), the GLASS LAI from the University of Maryland (http://www.glass.umd.edu/Contact.html), the V$_{cmax25}$ products from the National Ecosystem Science Data Center, National Science & Technology Infrastructure of China (http://www.nesdc.org.cn), the meteorological datasets from the National Tibetan Plateau Third Pole Environment Data Center (https://data.tpdc.ac.cn/en/), the annual land use/cover datasets and the CCI LC user tool from the European Space Agency (ESA) (http://maps.elie.ucl.ac.be/CCI/viewer/), the soil data from the FAO (https://doi.org/10.3334/ORNLDAAC/1247), and the atmospheric $CO_2$ data from the National Oceanic and Atmospheric Administration's Earth System Research Laboratories (https://gml.noaa.gov/obop/mlo/).

**Author contributions**

Conceptualization, methodology, data analysis, writing— original draft, writing—review and editing: TC; conceptualization, methodology, writing— original draft, writing—review and editing: FM. Model, writing— original draft, writing—review and editing: MP. Conceptualization, funding acquisition, project administration, writing—review and editing: GT. Visualization, writing—review and editing: YY. Conceptualization, data analysis, funding acquisition, project administration, writing— original draft, writing—review and editing: HV. All authors have read and agreed to the published version of the manuscript.

**Supplement**

The supplement related to this article is available online.

**Competing interests**

The authors declare that they have no known competing financial interests or personal relationships that could have appeared to influence the work reported in this paper.

**Disclaimer**

Publisher's note: Copernicus Publications remains neutral with regard to jurisdictional claims in published maps and institutional affiliations

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
