# Peer review of "Elevated atmospheric CO2 concentration and vegetation structural changes contributed to GPP increase more than climate and forest cover changes in subtropical forests of China"

_Biogeosciences, 2023_

## Author Comment (AC1)

**RC2**: 'Comment on bg-2023-140', Anonymous Referee #2, 30 Oct 2023

Dear Reviewer and Editor,

We would like to thank the reviewer for his interest in our study, and for the feedback provided. We appreciate these constructive and specific comments, which will help improve the quality of the manuscript. We have carefully inspected all reviewer comments. Below, you will find our responses to the comments (responses in blue).  Please find the response to each comment below.

We hope that you will find the result satisfying.

Sincerely,

Tao Chen, Félicien Meunier, Marc Peaucelle, Guoping Tang, Ye Yuan, Hans Verbeeck

Review of "Elevated atmospheric $CO_2$ and vegetation structural changes contributed to GPP increase more than climate and forest cover changes in subtropical forests of China" by Chen et al.

The manuscript by Chen et al. investigates drivers of subtropical forest GPP trends in China using a process-based model that runs to provide causal attribution. The study concludes that the primary drivers of GPP change are the $CO_2$ fertilization effect and increased LAI. While the study conducts comprehensive model experiments and maintains a well-organized structure, it lacks a convincing theoretical framework for designing the experiments and conducting the analysis, which is essential for consideration in publication. Additionally, the manuscript requires careful revision for the English language and logical syntax. Please refer to my comments for further details.

**Response:** Thank you very much for your valuable and thoughtful comments and suggestions. Below we go through point-by-point our answers to the comments, and our responses are in blue. Moreover, we have also streamlined the results, figures and text as suggested. Additionally, we have carefully checked and improved the English writing in the revised manuscript.

General comments:

1. Introduction: In the second paragraph, several relevant drivers are listed, followed by the research question "the relative contributions of these factors…not clear" in the next paragraph. It does not adequately explain to the reader why these factors are crucial to GPP or provide mechanistic expectations. For instance, in Line 60, rather than stating the increased temperature "has also influenced the

forest carbon uptake", it would be beneficial to summarize the specific mechanisms and reasons behind this influence. Is the influence positive or negative? Some clarifications would be helpful.

**Response:** Thank you very much for the valuable suggestions. To make the possible mechanisms behind the GPP changes clearer, we have added the following sentences to the revised text.

"Previous studies showed that temperature is the major factor influencing GPP variations in the Yangtze River Basin located in southern China (Nie et al., 2023), as well as other southern parts of China (Ma et al., 2019). Li et al., (2022) highlighted that precipitation dominated the interannual changes in the GPP of forest ecosystems in Southwest China, while the GPP changes were more affected by solar radiation than by precipitation and temperature in humid region of China (Chen et al., 2021a). The changes in GPP in response to different climatic factors can be both positive and negative across different regions and periods. For example, properly increasing temperature can promote enzyme activity and $CO_2$ fixation (Siddik et al., 2019; Moore, et al., 2021). However, when the temperature increases exceed the optimal temperature, the activity of enzymes in plants will decrease, thereby affecting the photosynthesis rate and carbon sequestration. Additionally, climate warming could increase the vapor pressure deficit (VPD), leading to more drought stress on plants (Yuan et al., 2019). Generally, when atmospheric moisture is insufficient, plants tend to inhibit photosynthesis by reducing stomatal conductance, thereby significantly reducing GPP (Yuan et al., 2019; Grossiord et al., 2020). The vegetation productivity response to the precipitation variations shows large spatial heterogeneity (Camberlin et al., 2007), which largely depends on topographic attributes, vegetation types, and even soil texture."

References:

Nie, C., et al., 2023. The Spatio-Temporal Variations of GPP and Its Climatic Driving Factors in the Yangtze River Basin during 2000–2018. Forests, 14(9):1898.

Li, Y., et al., 2022. Interannual variations in GPP in forest ecosystems in Southwest China and regional differences in the climatic contributions. Ecological Informatics, 69: 101591.

Ma et al., 2019. Trends and controls of terrestrial gross primary productivity of China during 2000–2016. Environmental Research Letters, 14, 8.

Chen, S. et al., 2021a. Vegetation structural change and $CO_2$ fertilization more than offset gross primary production decline caused by reduced solar radiation in China. Agricultural and Forest Meteorology, 296: 108207.

Siddik, M.A., et al., 2019. Responses of indica rice yield and quality to extreme high and low temperatures during the reproductive period. European Journal of Agronomy, 106, 30-38.

Moore, C.E., et al., 2021. The effect of increasing temperature on crop photosynthesis: from enzymes to ecosystems. Journal of Experimental Botany, 72 (8), 2822-2844.

Yuan, W. P., et al., 2019. Increased atmospheric vapor pressure deficit reduces global vegetation growth. Science Advances, 5, eaax1396.

Grossiord, C., et al., 2020. Plant responses to rising vapor pressure deficit. New Phytologist, 226(6), 1550–1566.

Camberlin, P., et al., 20007. Determinants of the interannual relationships between remote sensed photosynthetic activity and rainfall in tropical Africa. Remote sensing of environment, 106, 199–216.

2. Experiment design: I have two main concerns concerning the experiment design in Table 1. A) When assessing the effect of climate variables on GPP, one of the climate variables (e.g., precipitation) is fixed as the value in 2001 in the forcing for the S2 scenario. As I understand it, that means in the S2 scenario there is no climatological cycle at all. The difference in GPP between S2 and the control run should include the effect of both the long-term trend and short-term variabilities of climate. This means, by design, the trend of GPP driven by climate is overshadowed by the shorter-term variabilities (Figure 6). However, when designing the $CO_2$ and LAI scenarios, the difference of $CO_2$ or LAI forcings are less variable (Figure S10, S11), thus a "clear" trend of GPP can be observed in both Figure 7 (a) and 8 (a). There is no surprise when the authors find that $CO_2$ and LAI are the most prominent drivers, when they are comparing the effect of "trend" (e.g., $CO_2$) and "trend + variabilities" (e.g., precipitation). One may need to test to which extent the way of prescribing climate forcings influences the conclusion, e.g., by removing the trend of climate variables but keeping variabilities. B) Is the GLASS LAI also sensitive to climate change and increasing $CO_2$? With an increased carbon uptake due to increasing $CO_2$, more carbon can be allocated to leaf growth. I wonder if the authors have some thoughts about the causal link when discussing the effect of LAI on GPP.

**Response:** Thank you very much for the comment and suggestion. As suggested, we used the mean of each climate variable, including the precipitation, temperature, and solar radiation rather than the initial (2001) value for the different variables to redo the simulation. Here, taking the precipitation as an example, we compared the simulated results based on the mean value of the precipitation over the study periods (see below right figure) with the simulations in the first year (2001) (see below left figure), and found that there are no significant differences between them. Although there are relative differences in the magnitude of the slopes (i.e., compared the present simulations with previous simulations) under the effect of precipitation changes on GPP, they are not significant and show a similar effect,

suggesting that the effect of precipitation on changes in GPP in different forests is less influenced by trends. As shown in Fig. S9, our results also indicated that the annual variations of the climate variables have insignificant trends from 2001 to 2018. From Fig. 6 and Fig. S9, the results also suggested that the year 2001 was not an extreme year for any of those variables. Therefore, the initial year (2001) used in this study may be reasonable. The same experiment designs were also adopted in previous studies (Chen et al., 2021a; Sun et al., 2022). Overall, considering that there are no obvious trends in these climate variables and the low effects of these variables when compared to the $CO_2$ and LAI, whatever the experimental design it wouldn't change our findings.

[Figure]

For question B, we acknowledge that LAI may be affected by climatic factors and $CO_2$ fertilization. We added the following discussion to the revised manuscript as suggested.

Revised text to:

"Changes in LAI could be influenced by climatic factors and elevated atmospheric $CO_2$ concentration (Chen et al., 2019; Chen et al., 2021a; Sun et al., 2022). For example, previous studies reported that the elevated atmospheric $CO_2$ concentration was the dominant driver of global LAI increase, and there are regional differences in the impact mechanism of climate factors on LAI changes (Zhu et al., 2016; Zhu et al., 2017), thereby influencing the GPP dynamics. Moreover, the interactions between these driving factors can also influence the LAI, and even the interactive impacts of these factors on LAI may offset each other. For instance, rising in $CO_2$ concentration and solar radiation can affect temperature and VPD (Chen et al., 2021a). High VPD leads to plants to close their stomata, resulting in lower intercellular $CO_2$ concentrations in the leaves, which reduces the rate of photosynthesis (Yuan et al., 2019). Additionally, changes in LAI can feed back to the climate through biogeochemical and biogeophysical processes (Li et al., 2023). In short, there is a bidirectional interaction between vegetation and the atmosphere, and the relationship between vegetation dynamics and driving factors is complicated. The current methods used in this study cannot eliminate the complex interactions of the climate factors and elevated $CO_2$ concentration on LAI changes, which may bring some uncertainties to our results."

References:

Chen, S. et al., 2021a. Vegetation structural change and $CO_2$ fertilization more than offset gross primary production decline caused by reduced solar radiation in China. Agricultural and Forest Meteorology, 296: 108207.

Sun et al., 2022. Causes for the increases in both evapotranspiration and water yield over vegetated mainland China during the last two decades. Agricultural and Forest Meteorology, 324, 109118.

Chen, C., et al., 2019. China and India Lead in Greening of the World through Land-Use Management. Nature Sustainability, 2 (2), 122–129.

Zhu, et al., 2016. Greening of the Earth and Its Drivers. Nature Climate Change, 6 (8), 791–795.

Zhu et al., 2017. Attribution of seasonal leaf area index trends in the northern latitudes with "optimally" integrated ecosystem models. Global Change Biology, 23, 4798–4813.

Yuan, W. P., et al., 2019. Increased atmospheric vapor pressure deficit reduces global vegetation growth. Science Advances, 5, eaax1396.

Li, Y., et al., 2023. Biophysical impacts of earth greening can substantially mitigate regional land surface temperature warming. Nature Communications,14, 121.

3. Results: This study compares the contribution of different drivers to GPP in the unit of TgC/year (e.g., Figure 9). It is not introduced in the method section how the total GPP is calculated. If I assume GPP in TgC/year is the sum of GPP from all regions or the sum for each PFT, then it is highly related to the specific regions. Figure 1 shows the study region is mostly occupied by EBF and ENF, there is no wonder GPP is higher in TgC/year in EBF. In addition to that, the title indicates that $CO_2$ and LAI contribute more to GPP than forest cover changes. However, only very few regions are affected by forest cover change (Figure 5c), by contrast, all of the regions are under increasing $CO_2$ in the model experiment. It is unfair to compare the relative impact between these two drivers when looking at the total GPP. Or one has to make it clear in the beginning, that only total GPP in this specific region is considered.

**Response:** Thanks. The total GPP (TgC/year) for the entire forest area or a specific forest area (e.g., EBF, ENF, etc.) was calculated based on the regional mean value of GPP (gC/m$^2$/year) multiplied by the total area (m$^2$) of a certain forest type (1TgC = 1x10$^{12}$ gC). Then, the trends of the total GPP for different forests were computed from 2001 to 2018 based on the linear regression method, and the magnitude of the trends were used to measure the effect of different factors on GPP. For example, when we calculated the impact of factors such as LAI, climate change, and $CO_2$ concentration on the GPP of EBF, the area of EBF remained the same in the corresponding year. The same method was also used when calculating other forest types or the entire forest of the study area. Therefore, the results are

only affected by the magnitude of the mean value of the EBF area, and thus the effects of different factors on the GPP of a given forest type are comparable. In the revised version, we also added the statement of the method for calculating the total GPP for a given forest as follows:

"In this study, the total GPP (TgC/year and $1TgC=1\times10^{12}$ gC) was only applicable to a specific forest region, namely the entire subtropical forest region or the regions of EBF, DBF, ENF, and MXF, and it was calculated based on the regional mean of GPP (gC/m$^2$/year) multiplied by the total area (m$^2$) of a certain forest type."

4. Discussion: I like they discuss the model uncertainties. Most of the model discussion is about the input data, though it is important, the inherent model structure and underlying assumptions and how would these possibly affect the attribution is not so well discussed. For instance, it is not clear how the model simulates plants' response to $CO_2$. It would greatly enhance the understanding of the contribution results if the authors included more discussion on these elements.

**Response**: Thanks. As suggested, we added the following discussion about the BEPS model to the revised manuscript.

"In the BEPS model, the LAI is separated into two parts including the LAI of sunlit and shaded leaves, which are adopted to calculate the photosynthesis at leaf level (sunlit and shaded leaves) based on the FvCB photosynthesis model (Farquhar et al., 1980), and further compute the GPP at canopy level by adding the photosynthesis rates of sunlit and shaded leaves. Moreover, the Ball-Berry equation (Ball et al., 1987) was used in the model to calculate the stomatal conductance of sunlit and shaded leaves, which influenced the intercellular $CO_2$, the photosynthesis rate, and evapotranspiration (ET). Therefore, the LAI directly determined the allocation of energy and water availability and influenced the gross photosynthesis rate of the sunlit and shaded leaves. The accuracy of the LAI may impact its contribution to GPP variations through these processes. The atmospheric $CO_2$ concentration affects the intercellular $CO_2$ through the stomatal conductance, which, together with temperature and maximum carboxylation rate ($V_{cmax}$), determines the Rubisco-limited ($A_c$) and RuBP-limited ($A_j$) gross photosynthesis rate in the model. Over the past few decades, the $CO_2$ concentrations continuously increased and reached the current level of over 400 ppm. Elevated atmospheric $CO_2$ concentration can increase photosynthesis by accelerating the rate of carboxylation, thereby influencing the GPP changes. Additionally, solar radiation variability would directly influence the potential electron transport rate and thus regulate the RuBP-limited ($A_j$) gross photosynthesis rate. The temperature in the model directly impacts the $V_{cmax}$ and the $CO_2$ compensation point without dark respiration ($\Gamma$), thereby determining the gross photosynthesis rate. Moreover, the temperature positively affects the $V_{cmax}$ when it is below the optimal temperature. However, when the temperature exceeds the optimal temperature, $V_{cmax}$ will not continue to increase with the temperature. Therefore, changes in temperature in the model may have a positive or negative impact on GPP."

References:

Farquhar, et al., 1980. A biochemical model of photosynthetic $CO_2$ assimilation in leaves of C3 species. Planta 149, 78–90.

Ball, J.T., et., 1987. A model predicting stomatal conductance and its contribution to the control of photosynthesis under different environmental conditions. J. Biggins (Ed.). Progress in Photosynthesis Research: Volume 4 Proceedings of the VIIth International Congress On Photosynthesis Providence, Rhode Island, USA, August 10–15, 1986. Springer Netherlands, Dordrecht, pp. 221–224.

Specific comments:

1. L16: If you only use LAI to represent vegetation structural change, it might not be necessary to mention "VSC" explicitly.

   **Response:** Thanks for the suggestion. Yes, LAI does not represent all vegetation structure changes. As suggested, we use LAI directly in the revised version instead of VSC.

2. L29: Please be consistent with abbreviations.

   **Response:** Thanks. As suggested, the LAI and FCC were adopted here to be consistent with the abbreviations mentioned above.

3. L30: What do you mean by "overlooked"?

   **Response:** We are sorry for the confusion. Here we are trying to emphasize the importance of LAI. For clear understanding, the "overlooked" was changed to "essential".

4. L32: How might these findings inform climate change mitigation efforts or forest management strategies?

   **Response:** Thank you for the comment. GPP is a crucial indicator for estimating the carbon sequestration capacity of ecosystems (Chen et al., 2021b; Ma et al., 2019). Firstly, estimation of the GPP in the subtropical forests is important for people to understand how much carbon sequestration capacity it offers. For example, in this study, we have estimated the GPP of different forests, thus providing forest managers with basic data on the carbon sequestration potential of different Chinese subtropical forests. Secondly, we investigated the dynamics of GPP and their dominant driving factors in the study area. This information is crucial for decision-makers to adjust and optimize forest management policies promptly, so as to ensure that forests can provide the best ecological services for humans (Fang et al., 2010).

Additionally, China is still one of the world's top emitters of greenhouse gases that directly contribute to global warming (Chen al et., 2021). In September 2020, China announced the plan to achieve carbon neutrality by 2060 (Dong et al., 2021). This target closely aligns with the Intergovernmental Panel on Climate Change (IPCC) Special Report on 1.5 °C (SR15), which states that global $CO_2$ emissions must decline well before 2050 to curb the anticipated 1.5 °C global warming. Vegetation carbon uptake could significantly regulate the inter-annual variability of atmospheric $CO_2$ concentration and mitigate climate change. Developing forest carbon sinks is very important for China to achieve carbon neutrality. The Chinese government implemented several large-scale forestation programs since the 2000s, especially in the subtropical regions. Therefore, quantification of China's subtropical forest GPP and understanding of its driving mechanisms in this study can provide policy makers with a basic reference to answer the question: (1) How has the carbon sequestration potential of subtropical forests changed over the past decades? (2) Does the region have the potential to achieve carbon neutrality and mitigate climate change?

References:

Chen, Y. et al., 2021. Accelerated increase in vegetation carbon sequestration in China after 2010: A turning point resulting from climate and human interaction. Global Change Biology, 27(22), 5848-5864.

Ma, J., 2019. Trends and controls of terrestrial gross primary productivity of China during 2000–2016. Environmental Research Letters, 14, 084032.

Zhao et al., 2023. Toward the carbon neutrality: Forest carbon sinks and its spatial spillover effect in China. Ecological Economics, 209, 107837.

Dong, L., et al., 2021.China's carbon neutrality policy: objectives, impacts and paths. East Asian Policy, 13, 5-18.

Beer, C., et al., 2010. Terrestrial gross carbon dioxide uptake: global distribution and covariation with climate. Science, 329 (5993), 834–838.

Fang, J., et al, 2010. Why are East Asian ecosystems important for carbon cycle research? Science China Life Sciences, 53(7): 753–756.

5. L37: Carbon emissions?

**Response:** Sorry for the confusion. We have reworded the sentence as follows:

"Terrestrial ecosystems can capture carbon dioxide ($CO_2$) from the atmosphere through photosynthesis, which was regarded as a potential solution for slowing down the increase in global $CO_2$ concentration (Keenan et al., 2016) and mitigating global warming (Fang et al., 2018; Shevliakova et al., 2013)."

References:

Keenan, T.F., et al., 2016. Recent pause in the growth rate of atmospheric $CO_2$ due to enhanced terrestrial carbon uptake. Nature Communications, 7, 13428.

Fang, J., Yu, G., Liu, L., Hu, S. and Chapin, F.S., 2018. Climate change, human impacts, and carbon sequestration in China. Proceedings of the National Academy of Sciences, 115(16): 4015-4020.

Shevliakova E., et al., 2013. Historical warming reduced due to enhanced land carbon uptake. Proceedings of the National Academy of Sciences, 110,16730–16735.

6. L66-68: Which regions are they looking at? The major drivers on GPP vary a lot depending on regions and even seasons. Please be precise here.

   **Response:** Thanks. As suggested, we added the specific regions to the revised text as follows:

   "Previous studies showed that temperature is the major factor influencing GPP variations in the Yangtze River Basin located in southern China (Nie et al., 2023), as well as other southern parts of China (Ma et al., 2019). Li et al., (2022) highlighted that precipitation dominated the interannual changes in the GPP of forest ecosystem in Southwest China, while the GPP changes were more affected by solar radiation than by precipitation and temperature in humid region of China (Chen et al., 2021a). Therefore, the dominant factors affecting GPP vary a lot depending on regions and different time scales."

   References:

   Nie, C., et al., 2023. The Spatio-Temporal Variations of GPP and Its Climatic Driving Factors in the Yangtze River Basin during 2000–2018. Forests, 14(9):1898.

   Li, Y., et al., 2022. Interannual variations in GPP in forest ecosystems in Southwest China and regional differences in the climatic contributions. Ecological Informatics, 69: 101591.

   Ma et al., 2019. Trends and controls of terrestrial gross primary productivity of China during 2000–2016. Environmental Research Letters, 14, 8.

   Chen, S. et al., 2021a. Vegetation structural change and $CO_2$ fertilization more than offset gross primary production decline caused by reduced solar radiation in China. Agricultural and Forest Meteorology, 296: 108207.

7. L70-71: The term "$CO_2$ fertilization" has not been introduced. Do you mean the $CO_2$ fertilization effect is stronger in China than in other regions, or the $CO_2$ effect is stronger in forest ecosystems than in other ecosystems?

**Response:** Thank you. As suggested, we have added a brief introduction to $CO_2$ fertilization as follows.

"Elevated $CO_2$ concentrations may enhance the plant productivity, i.e., GPP, at the ecosystem scale (Wenzel et al 2016), which is known as the $CO_2$ fertilization effect."

Here, we mean that the southern region of China is more affected by the carbon dioxide fertilization effect than other regions of China. Revised text to:

"$CO_2$ fertilization was the pivotal driver for enhancing carbon sink in terrestrial vegetation, and the southern region of China also being more affected by the $CO_2$ effect than other regions of China (Chen et al., 2019b; Zhu et al., 2019b)."

References:

Wenzel, S., et al., 2016. Projected land photosynthesis constrained by changes in the seasonal cycle of atmospheric $CO_2$. Nature, 538, 499–501.

Chen et al., 2019b. Vegetation structural change since 1981 significantly enhanced the terrestrial carbon sink. Nature Communications, 10(1): 4259.

Zhu et al., 2016. Greening of the Earth and its drivers. Nature Climate Change, 6, 791–795.

8. L73-74: "…most of the current studies…", really? At least different PFTs are represented in land surface or earth system models.

    **Response:** We agree that different PFTs are represented in land surface or earth system models. Although some studies set different parameters based on PFTs during the simulation (e.g., simulating GPP), they often consider different forests (e.g., EBF, ENF, MXF, etc.) as a single forest type (i.e., forest) when analyzing the results, especially in large-scale research (Chen et al., 2021a). We apologize for the misleading description. We have changed the statement "most of the current studies" to "some of the current studies".

    References:

    Chen, S. et al., 2021a. Vegetation structural change and $CO_2$ fertilization more than offset gross primary production decline caused by reduced solar radiation in China. Agricultural and Forest Meteorology, 296: 108207.

9. L86: How "better-performed" is BEPS? It seems unusual to encounter the conclusion without having reviewed the results, where the performance of the BEPS model has been tested.

    **Response:** Thanks. Yes, the BEPS model has been tested and validated at the regional and global scales. Considering the statement "better-performed" is not necessary, we have removed the confusing sentence from the revised text.

Revised text to: "Recently, the BEPS model has been widely used to simulate carbon fluxes at the regional and global scales (Chen et al., 2019b; Chen et al., 2012; Liu et al., 1997; Luo et al., 2019; Wang et al., 2021a), especially it has been well evaluated and validated in China (Feng et al., 2007; Liu et al., 2018; Peng et al., 2021; Wang et al., 2018)"

References:

Chen, J.M., 2019b. Vegetation structural change since 1981 significantly enhanced the terrestrial carbon sink. Nature Communications, 10(1): 4259.

Chen, J.M. et al., 2012. Effects of foliage clumping on the estimation of global terrestrial gross primary productivity. Global Biogeochemical Cycles, 26(1): GB1019.

Liu, J., et al., 1997. A process-based boreal ecosystem productivity simulator using remote sensing inputs. Remote Sensing of Environment, 62(2): 158-175.

Luo, X., et al., 2019. Improved estimates of global terrestrial photosynthesis using information on leaf chlorophyll content. Global Change Biology, 25(7): 2499-2514.

Wang, M., Wang, S., Zhao, J., Ju, W. and Hao, Z., 2021a. Global positive gross primary productivity extremes and climate contributions during 1982-2016. Science of the Total Environment, 774: 145703.

Feng, X. et al., 2007. Net primary productivity of China's terrestrial ecosystems from a process model driven by remote sensing. Journal of Environmental Management, 85(3): 563-573.

Liu, Y. et al., 2018. Satellite-derived LAI products exhibit large discrepancies and can lead to substantial uncertainty in simulated carbon and water fluxes. Remote Sensing of Environment, 206: 174-188.

Peng, J. et al., 2021. Incorporating water availability into autumn phenological model improved China's terrestrial gross primary productivity (GPP) simulation. Environmental Research Letters, 16(9): 094012.

Wang, M. et al., 2018. Detection of Positive Gross Primary Production Extremes in Terrestrial Ecosystems of China During 1982-2015 and Analysis of Climate Contribution. Journal of Geophysical Research: Biogeosciences, 123(9): 2807-2823.

10. L93: Do you mean different GPP products?

**Response:** Thanks for catching the error in the description. Here are the GPP of different forest types. We reworded the sentence as follows:

"to quantify the spatiotemporal trends in GPP of different forests across the subtropics."

11. L95-96: I find this statement not specific. Also, see my comment before.

**Response:** To make it clearer, we have added the following sentences to the revised manuscript.

"The results of this study can provide forest managers with basic data on the carbon sequestration potential of different Chinese subtropical forests. Moreover, investigating the dynamics of GPP and their dominant driving factors in the study area is crucial for decision-makers to adjust and optimize forest management policies promptly, so as to ensure that forests can provide the best ecological services for humans."

12. L139: What are "the other parameters"?

**Response:** The other important parameters include the clumping index, maximum stomatal conductance, specific leaf area, respiration coefficient for leaf, stem, coarse root, and fine root, as well as the Q10 for leaf, stem, and root. We have added this information to the revised text. Revised text to:

"The other important parameters, including the clumping index, maximum stomatal conductance, specific leaf area, respiration coefficient for leaf, stem, coarse root, and fine root, and Q10 for leaf, stem, and root, used in the BEPS model for each plant functional type can be found in Liu et al. (2018)"

References:

Liu, Y. et al., 2018. Satellite-derived LAI products exhibit large discrepancies and can lead to substantial uncertainty in simulated carbon and water fluxes. Remote Sensing of Environment, 206: 174-188.

13. L147-149: How is the "nighttime flux correction" done? Gap filling and flux partitioning are not data quality control.

**Response**: According to the flux dataset processing standards developed by ChinaFLUX (Zhang et al.), the nighttime flux correction mainly includes removing outliers when there is precipitation, $CO_2$ concentration exceeds the instrument's measurement range, insufficient turbulence (e.g., u* <0.2 m/s), and less than 15,000 valid samples. We have added this information to the revised text. As suggested, we also removed the statement "gap filling and flux partitioning" from the revised text.

References:

Zhang et al., 2019. Carbon and water fluxes observed by the Chinese Flux Observation and Research Network (2003–2005). China Scientific Data, 4(1), DOI: 10.11922/csdata.2018.0028.zh.

14. L150: Which u* is used for each site?

**Response**: Thanks. We have added the specific values of u* for each site, namely the threshold of u* < 0.2 m s$^{-1}$ was used for the QYZ and ALS stations, while the threshold of u* < 0.05 m s$^{-1}$ was used for DHS station.

15. L167: Vague statement. What does "robust enough" mean?

**Response:** We apologize for this error in the description. Revised text to:

"It has been shown that this can effectively reduce the uncertainty in the simulations of the BEPS model."

16. L195: You mean "original vegetation classes"?

**Response:** Yes, the "original classes" changed to "original vegetation classes".

17. L210-213: The sentence is not clear.

**Response:** Sorry for the inappropriate description. To avoid confusion, we have removed the statement from the revised text where there are unnecessary.

18. L244: "reasonably well" is not an accurate phrasing, notably considering that all $R^2$ values are below 0.5. Why is NEP only used for testing model performance? Why is NEP exclusively used for testing the model's performance? There seems to be a lack of additional results or discussion regarding NEP thereafter.

**Response:** Thank you very much for the comment. As suggested, we have removed the "reasonably" from the revised text. Yes, we also used the NEP for testing the model performance, because NEP (i.e., -NEE) is a direct measurement of carbon fluxes between the atmosphere and ecosystems, while the ecosystem GPP cannot be measured directly and is derived from the partitioning of NEE from flux measurements. Therefore, we not only used the observed GPP from the flux sites to validate our model, but also the NEP. We recognized that the validation of model performance based on measured NEP was relatively lower than that of GPP. One reason for this is that the simulation of NEP in the model is affected not only by the accuracy of simulated GPP, but also by the accuracy of simulated heterotrophic respiration ($R_h$) and autotrophic respiration ($R_a$). Therefore, a relatively poor performance of the simulated NEP was observed in this study.

However, the purpose of this is to disentangle how different drivers affect GPP changes in China's subtropical forests. Therefore, we mainly focus on the GPP in our study area. Our findings also showed that the validation of the simulated GPP

at three flux sites performed well. We have added the explanations to the discussion section in the revised manuscript. Revised text to:

"In this study, we also used the NEP for testing the model performance, because NEP (i.e., -NEE (net ecosystem exchange)) is a direct measurement of carbon fluxes between the atmosphere and ecosystems, while the ecosystem GPP cannot be measured directly and is derived from the partitioning of NEE from flux measurements. Therefore, we not only used the observed GPP from the flux sites to validate our model, but also the NEP. The validation of model performance based on measured NEP was relatively lower than that of GPP. One cause is that the simulation of NEP in the model is influenced not only by the accuracy of simulated GPP, but also by the accuracy of simulated heterotrophic respiration ($R_h$) and autotrophic respiration ($R_a$). Therefore, a relatively poor performance of the simulated NEP was observed in this study."

19. What do the green lines and circles represent in Figure 2?

**Response:** Thanks. The green lines represent the simulated GPP, and the dark circles represent the observations. We added the description of the green lines and dark circles in the Figure caption (see below).

"Figure 2 Comparison of simulated GPP with measured GPP from three flux tower stations at daily (a-c) and annual (d-f) scales. The green lines and dark circles represent the simulated GPP and observed GPP, respectively."

20. L254-255: It is not clear how the spatial correlation is calculated.

**Response:** Thanks. Here the spatial correlation is calculated pixel by pixel at the annual scale. For example, we obtained the MODIS GPP from a certain pixel, and our simulated GPP was also derived from the same pixel during the same period. Then, the correlation between the two GPPs was computed. Similarly, we can calculate the correlation coefficients of different pixels to obtain their spatial distribution. We added the following description of the methodology for calculating spatial correlation to Section 2.5 of the revised manuscript. Revised text to:

"spatial correlation is calculated pixel by pixel at the annual scale. First, we should obtain two GPP time series for a certain pixel in the same period, and then calculate the correlation between the two GPPs. By analogy, the correlation coefficients of different pixels can be calculated to obtain the spatial distribution of the correlation coefficients."

21. L261-264: The number does not align within the range of all five GPP products as mentioned. Additionally, the reference to 'another BEPS' requires clarification. How to interpret the difference between "another BEPS" and "this BEPS" in Figure S7d?

**Response:** Thanks. We acknowledge that our simulated GPP is slightly higher than other products. Although our estimated GPP is slightly higher for the entire

study area, our modeled GPP is very close to other GPP products for a specific forest type, such as the DBF and MXF (Fig. S8). In fact, other GPP products (e.g., MODIS GPP, EC-LUE GPP, NIRv GPP, and VPM GPP) also have large differences when they are compared with each other (Fig. S8). These results indicate that there are still significant differences in simulating GPP to date. The possible reasons are:

(1) there are some substantial differences in the simulated GPP from various ecosystem models due to many differences in model structure, parameterization, and driving data (Cai et al., 2014; Lin et al., 2023).

(2) our simulated GPP is compared with other GPP products mainly generated by the LUE model-based and remote sensing-based models. However, previous studies (Zhu et al., 2018; O'Sullivan et al., 2020; Wang et al., 2023) also reported that LUE-based models, remote sensing-based models, and machine-learning-based models may underestimate the GPP at an annual scale. For example, the GPP estimates by the LUE models mainly depend on a few important factors, including solar radiation, air temperature, water availability, and vegetation indexes (e.g., EVI or NDVI). Current LUE-based models do not completely integrate the other key environmental regulations to vegetation productivity, such as the effect of atmospheric $CO_2$ concentration, canopy structure (e.g., LAI), diffuse radiation, etc. on GPP.

(3) the underestimation in other GPP products is possibly due to failure to assess the $CO_2$ fertilizer effects, because almost no apparent response to the rising atmospheric $CO_2$ concentration in the LUE models leads to an underestimated trend (Anav et al., 2015). In our study, the GPP was estimated by a process-based model (i.e., BEPS) that considers the effects of these important factors on GPP, especially the $CO_2$ fertilization effect, which may lead to a higher GPP compared to all the other products.

For what it's worth, the results of our comparisons showed that the interannual trends of our simulated results were in line with other GPP products (Fig. S8). Despite possible overestimation, the purpose of this study mainly focuses on the trends and explains the driving mechanism behind them, thus it may not affect our results and conclusions.

We added the above discussion to the revised version. We also changed the statement "…, which falled in the range of the five GPP products… " to "…, which was closed to the magnitudes of the five GPP products… ".

In order to distinguish it from the GPP we simulated, the reference (BEPS$_g$ GPP) to 'another BEPS' has been added to Table S3. Actually, another BEPS GPP product was also produced by a similar BEPS model. However, this model is driven by the global datasets, and the parameters in the model are also calibrated for the GPP mapping. Therefore, it can be used for comparison with our simulated GPP.

References

Cai, W., et al., 2014. Large differences in terrestrial vegetation production derived from satellite-based light use efficiency models. Remote Sensing, 6(9), 8945–8965.

Lin et al., 2023. Underestimated Interannual Variability of Terrestrial Vegetation Production by Terrestrial Ecosystem Models. Global Biogeochemical Cycles, 34(4), e2023GB007696.

Zhu et al., 2018. Underestimates of Grassland Gross Primary Production in MODIS Standard Products. Remote Sensing, 2018, 10(11), 1771.

Wang et al., 2023. Assessment of Six Machine Learning Methods for Predicting Gross Primary Productivity in Grassland. Remote sensing, 15(14), 3475.

O'Sullivan, M., et al. 2020. Climate-driven variability and trends in plant productivity over recent decades based on three global products. Global Biogeochemical Cycles, 34(12), e2020GB006613.

Anav, A., et al., 2015. Spatiotemporal patterns of terrestrial gross primary production: a review. Reviews of Geophysics, 53(3), 785-818.

22. L268-269: Rather than a simple conclusion that BEPS-GPP aligns well with other GPP products, it would be more informative to delineate areas of agreement and disagreement between the models.

**Response**: Thank you for the constructive suggestion. We added the following sentences to the revised text:

"Overall, our simulated GPP is slightly higher for the entire study area than other products. For specific forest types such as DBF and MXF, our modeled GPP is very close to other GPP products, but has some differences compared to EBF and ENF (Fig. S8). Similarly, these commonly used GPP products also have large differences when compared to each other (Fig. S8). These results indicate that there is still a large discrepancy in modelling GPP to date, due to many differences in model structure, parameterization, and driving data."

23. L277: Please explain what is the "interactive effect".

**Response:** Here the "interactive effect" represents the effect of different drivers together, namely, GPP is simultaneously influenced by different driving factors, such as changes in the climatic factors, vegetation status, etc. It is worth noting that these effects may be non-linear and cannot be simply added together.

24. L281: "…of the forest GPP", do you mean forest areas showed increased and decreased GPP?

**Response:** Yes, we mean that 90.4% of the forest areas in the study area exhibited an increasing trend in GPP, while 9.6% of the forest areas showed a decreasing trend in GPP. Sorry for the confusion. We have updated the sentence as follows:

"Spatially, 90.4% of forest areas in the study area showed an increasing trend in GPP, while 9.6% of forest areas exhibited a decreasing trend in GPP."

25. L297: What is "stable state"? No forest cover change? Or no significant effect of forest cover change?

    **Response:** Thanks for the comment. True, here the stable state indicates no forest cover change.

26. In Figure 5 (b), the time series of GPP in MXF seems to be very symmetric with GPP in ENF, any explanations for that?

    **Response:** Thanks. Such results are mainly due to the inter-annual variability of the area of each forest type and the conversion between them. This is because in this section we only investigated the effect of changes in the area of each forest type on the GPP.

27. L307: Is the increasing trend significant?

    **Response:** Thanks. As shown in Fig. S9, the trends in annual precipitation and temperature of the entire study area showed increasing trends, but are not significant. However, the trends in annual precipitation and temperature varied spatially (Fig. S9b and Fig. S9d), with some areas showing significant increasing trends.

28. L334: "…58.2% of the…", but quite a lot of white spaces are shown up on the map. How is the 58.2% derived? Are you referring to Fig. 6h in this statement?

    **Response:** Thank you for the comment. The 58.2% was computed as the ratio of the pixels with a decreasing trend to the total number of pixels in the study area. This statement refers to Fig. 6f. A lot of white spaces mainly arise from the results of masking non-forested areas. To avoid confusion, we have updated the color of the mask area in the revised manuscript.

29. In Figure 6a, most of the variabilities are from EBF, any explanations?

    **Response:** Thanks. As shown in Fig. S9b and Fig. 1, the predominant forest type in areas with significant variability in precipitation is EBF, especially in some parts of the West. However, precipitation is relatively stable in other forest regions (e.g., ENF, MXF, etc.). Therefore, changes in precipitation have a greater impact on EBF, leading to most of the variabilities being from EBF.

30. L381-383: Where does the conclusion "…EBF…has the highest carbon uptake potential" come from?

**Response:** We are sorry for the confusion. We also reworded the sentence as follows:

"Overall, the GPP of EBF in the subtropical region of China showed the fastest growth rate when compared with other forest types (Fig. 9b)."

31. L423-424: But in Table S6, the majority of the ENF has been observed to transition into MXF (19040 km$^2$).

**Response:** Thank you for the comment. Yes, there were 19,040 km$^2$ of MXF was converted from ENF, however, this is a conversion between different forest types. The conversion between different forest types just leads to changes in the internal structure of the forests, such as increasing or decreasing LAI. Therefore, this conversion was also considered in this study. As shown in Table S6, when ENF converts to non-forests, the ENF mainly converts to cropland (13,100 km$^2$). We are sorry for the confusion. We have reworded the sentence as follows:

"The total area of the ENF was lost obviously during the study period in eastern and southern regions, and the ENF was mainly converted to cropland (13,100 km$^2$) (Table S6), causing large parts of GPP to decrease."

32. L450: Could you explain how climate warming negatively influences GPP in your study?

**Response:** Thanks. Our findings found that temperature induced the GPP decrease and mainly located in large parts of the eastern and the southwest (see Fig. 6d). In these areas, the temperature showed significant increasing trends (see Fig. S9d). The results indicated that increased temperature led to GPP reduction. Climate warming could increase the vapor pressure deficit (VPD), leading to more drought stress on plants (Yuan et al., 2019). Generally, when atmospheric moisture is insufficient, plants tend to inhibit photosynthesis by reducing stomatal conductance, thereby significantly reducing GPP (Yuan et al., 2019; Grossiord et al., 2020). Additionally, when the temperature increases exceed the optimal temperature, the activity of enzymes in plants will decrease, thereby affecting the photosynthesis rate and carbon sequestration. As suggested, we have added these discussions to the revised manuscript.

References:

Yuan, W. P., et al., 2019. Increased atmospheric vapor pressure deficit reduces global vegetation growth. Science Advances, 5, eaax1396.

Grossiord, C., et al., 2020. Plant responses to rising vapor pressure deficit. New Phytologist, 226(6), 1550–1566.

33. L460-462: Why do you observe different behaviors between EBF and ENF? Any hypothesis for that?

**Response:** Thanks for the comment. The cause of the observed different behaviors between EBF and ENF is that different forest types have different geographical distributions and are subject to different influences of climatic factors. As shown in Fig. 1, ENF is mainly distributed in the eastern and western regions of the subtropics. Our results showed that climatic factors (e.g., temperature, precipitation, and solar radiation) in these regions have negative effects on the GPP of ENF (Fig. 6), particularly the solar radiation declined significantly in the eastern region, which led to a decrease in the GPP of ENF in the east. For EBF, it is mainly distributed in the central and some western regions where climate change mainly contributes to the increase of GPP of EBF, especially the precipitation and temperature in the small area of the west (see Fig. 6h) contribute significantly to EBF GPP increase.

34. L486-L488: How much increase in LAI is related to the forest protection projects?

**Response**: Thanks for the comment. The Chinese Natural Forest Protection Project (NFPP) has been implemented around 2000 and completed by the end of 2020. Therefore, we first obtained the natural forest protection region in our study area (see below left figure) from the National Ecosystem Science Data Center (http://www.nesdc.org.cn/). Further, we calculated the annual average LAI for the region to compare the LAI changes over two phases (i.e., 1981-2000 and 2001-2018) (see right figure). Before 2000, the annual mean LAI showed a relatively stable state (slope = 0.0004 $m^2/m^2$/year, p > 0.05), and in the second phase (our study period), the annual mean LAI displayed a significant increasing trend (slope = 0.0101 $m^2/m^2$/year, p < 0.05), indicating that the implementation of NFPP may contribute to the increase in LAI.

[Figure]

[Figure]

35. L495: Chen et al. attribute drivers to GPP in gC/m2/year, which is not comparable with the GPP attribution in this study because of different regions and units as I mentioned in the general comments. The results in Zhan et al. stem from a land surface model instead of eddy covariance records.

**Response:** Thanks for the comment! As suggested, we first removed the reference of Chen et al.,2022a from the revised text. Besides, we also reworded the sentence in the revised version as follows:

"This was also confirmed by the results of a previous study using a terrestrial biosphere model (i.e., the QUINCY model) (Zhan et al., 2022)."

References:

Chen, C., 2022a. CO$_2$ fertilization of terrestrial photosynthesis inferred from site to global scales. Proceedings of the National Academy of Sciences, 119(10): e2115627119.

Zhan, C., et al., 2022. Emergence of the physiological effects of elevated CO$_2$ on land-atmosphere exchange of carbon and water. Global Change Biology, 28(24): 7313-7326.

36. L515-517: "…still in the early developing stage…" Could you specify the limitations of using this V$_{cmax25}$ product? Is the limitation about the theory or data quality?

    **Response:** Thanks. It is possible that the limitation may derive from the data quality and the key parameters in the model. Following your suggestion, we added the following sentences to the revised text to specify the limitations of using this V$_{cmax25}$ product.

    "The V$_{cmax25}$ product was mainly generated by the MODIS surface reflectance, thus the data quality of the surface reflectance may cause the uncertainty in V$_{cmax25}$ product. The uncertainties in MODIS reflectance datasets can arise from sensor calibration issues, cloud contamination, atmospheric correction errors, etc. Changes in the reflectance could result in large changes in the modelled chlorophyll values, thereby affecting the V$_{cmax25}$ product. Additionally, the V$_{cmax25}$ was produced by a semi-mechanistic model (Friend., 1995), and the key parameter $K_{cat}^{25}$ in the model (the Rub turnover rate at 25 °C) would bring uncertainties in modeling V$_{cmax25}$, because current ground-based data are still rarely used for calibration of this parameter and validation of the V$_{cmax25}$ products (Lu et al., 2022; Chen et al., 2022b)."

    References:

    Friend, A., 1995. PGEN: an integrated model of leaf photosynthesis, transpiration, and conductance Ecol. Modell. 77 233–55.

    Lu, X., et al., 2022. Estimating photosynthetic capacity from optimized Rubisco–chlorophyll relationships among vegetation types and under global change. Environmental Research Letters, 17(1): 014028.

    Chen, J.M. et al., 2022b. Global datasets of leaf photosynthetic capacity for ecological and earth system research. Earth System Science Data, 14(9): 4077-4093.

37. Kindly utilize diverging color schemes with the midpoint at 0 for clarity.

**Response:** Thanks for the suggestion. Yes, a diverging color scheme with the midpoint at 0 was adopted in this study. We also revised Fig. S9 using the diverging color scheme.

38. I suggest minimizing the use of abbreviations in the conclusion for better clarity. If necessary, they can be reintroduced.

    **Response:** Thank you for the suggestion. The full name of different abbreviations was added to the conclusion section of the revised manuscript, as suggested.

Technical corrections:

1. L164: "yearly" means "from year to year".

   **Response:** We changed the "yearly" to "annual".

2. L470: "increase" instead of "improve".

   **Response:** Thanks again! The "increase" has been changed to "improve".

---

## Author Comment (AC2)

**RC1**: 'Comment on bg-2023-140', Anonymous Referee #1, 29 Oct 2023

Dear Reviewer and Editor,

We would like to thank the reviewer for his interest in our study, and for the feedback provided. We appreciate these constructive and specific comments, which will help improve the quality of the manuscript. We have carefully inspected all reviewer comments. Below, you will find our responses to the comments (responses in blue). Please find the response to each comment below.

We hope that you will find the result satisfying.

Sincerely,

Tao Chen, Félicien Meunier, Marc Peaucelle, Guoping Tang, Ye Yuan, Hans Verbeeck

**Reviewer #1**

In their study, the authors investigate the influence of different drivers on changes in GPP in subtropical forests in China. The considered drivers were climate change, forest cover change, change in vegetation structure, and changes in $CO_2$ concentrations.

The authors use the BEPS model and run multiple simulations to disentangle the impact of the different drivers and find that atmospheric $CO_2$ and vegetation structure play the most important roles.

This is an interesting and well-conducted study and the manuscript is decently written. In my opinion, this study can be published in Biogeosciences after it went through some major revisions.

I mainly find that there needs to be some more model evaluation. Furthermore, some results need to be explained better. Also, the discussion has some points that need to be made clearer or added (see details below).

I'd further suggest some streamlining of results, figures, and text. There are 10 Figures, often with 6 panels. I believe this could be made more concise.

Further detailed comments follow below.

**Response:** Thank you very much for the valuable comments and suggestions. Below we go through point-by-point our answers to the comments, and our responses are in blue. Moreover, we have also streamlined the results, figures and text as suggested. Especially, Figures 1, 2, 9 and 10 are the most important and remain in the main text. The rest do not necessarily need to be placed in the main text and have been moved to the supplementary.

**Abstract:**

Why call it VSC and not just LAI?

**Response:** Thank you for this comment. LAI is one of the most important parameters representing vegetation structure, which can influence the carbon cycle and is widely used in models (e.g., LUE-based models and process-based models) to simulate carbon and water fluxes (Chen et al., 2019; Zhang, X. et al., 2022). Thus, the VSC was adopted in our study to represent LAI. Indeed, LAI does not represent all vegetation structure changes. As suggested, we use LAI directly in the revised version.

References:

Chen, J.M. et al., 2019. Vegetation structural change since 1981 significantly enhanced the terrestrial carbon sink. Nature Communications, 10(1): 4259.

Zhang, X. et al., 2022. Land cover change instead of solar radiation change dominates the forest GPP increase during the recent phase of the Shelterbelt Program for Pearl River. Ecological Indicators, 136: 108664.

**Introduction**

l. 39: the statement about the 30% is not a result of the cited study and is also not cited there... Please find a better reference

**Response:** Thank you for this suggestion. We added the new reference in the revised version (see below).

"Giovanni Forzieri, et al, 2022. Emerging signals of declining forest resilience under climate change. Nature, 608, 534–539"

l. 55: should be 0.82 billion I guess.

**Response:** Yes. The 8.2 billion has been changed to 0.82 billion.

l. 59: is this compared to global surface temp or temp over land?

**Response:** Thanks. It is compared to the global surface temperature. We have rewritten the sentence as follows.

"the annual mean temperature in the Chinese subtropical monsoon region has increased by more than 1.0 °C over the past 30 years (Fang et al., 2018), which was higher than the global surface temperature increase (Sun et al., 2019)."

References:

Sun, C., et al, 2019. Changes in extreme temperature over China when global warming stabilized at 1.5 °C and 2.0 °C. Scientific Reports, 9:14982.

Fang, J., et., 2018. Climate change, human impacts, and carbon sequestration in China. Proceedings of the National Academy of Sciences, 115(16): 4015-4020.

**Methods:**

l. 103: what about the spread of temperature as you mention for precipitation?

**Response:** Thanks for this comment. We added the following sentence to the revised version for describing the spread of temperature.

"The average annual temperature normally increases from the northwest toward the southeast, and the average annual temperature is about 15.5°C."

l. 115: NEP was not introduced. Generally, a glossary with abbreviations would be helpful.

**Response:** We have added the full name of the NEP (i.e., net ecosystem productivity) in the revised text. As suggested, we also added the following glossary of acronyms in the revised text to show abbreviations for other terms.

List of Abbreviations

| Abbreviation | Definition |
| --- | --- |
| BEPS | The Boreal Ecosystem Productivity Simulator |
| GPP | Gross primary productivity |
| FCC | Forest cover change |
| LAI | Leaf area index |
| CC | Climate change |
| $CO_2$ | Carbon dioxide |
| EBF | Evergreen broadleaved forest |
| ENF | Evergreen needle-leaved forest |
| DBF | Deciduous broadleaved forest |
| MXF | Mixed forest |
| QYZ | Qianyanzhou station |
| DHS | Dinghushan station |
| ALS | Ailaoshan station |
| $V_{cmax}$ | The maximum carboxylation rate |
| NEP | Net ecosystem productivity |
| ER | Ecosystem respiration |

l. 116: some more text on the model is necessary to allow the reader to get a basic understanding of it. It may go into the supplements.

**Response:** Thank you for the constructive suggestion. Following your suggestion, we have added more descriptions (please see below) about the model in the supplementary (see Text S1).

"Text S1 (description of the BEPS model)

The BEPS model was originally developed at the Canada Centre for Remote Sensing to assist in natural resources management (Liu et al., 1997). Compared with 15 prognostic models that participated in the Global Carbon Project (GCP) (Le Quere et al., 2018), BEPS results are mostly better in terms of the Pearson regression coefficient ($R^2$), root mean square error (RMSE), accumulated total sink, and trend against the residual land sink reported by Le Quere et al (2018). The BEPS model was mainly driven by remotely sensed datasets, which can be used for simulating the key carbon (e.g., GPP, NPP and NEP) and water (e.g., ET) fluxes of the terrestrial ecosystems at the yearly, daily and hourly scales. In the BEPS model, there are 8 plant functional types (PFTs), including shrubland, grassland, cropland, and four forest types (the evergreen needleleaf forests (ENF), deciduous needleleaf forests (DNF), deciduous broadleaf forests (DBF), evergreen broadleaf forests (EBF), mixed forests (MXF)).

At the daily scale, the BEPS model was driven by the daily leaf area index (LAI), daily meteorological data, etc. Daily carbon fixation in the BEPS model is calculated by scaling Farquhar's leaf biochemical model (Farquhar et al., 1980) up to canopy-level implemented with a spatial and temporal scaling scheme (Chen et al., 1999). Daily gross primary productivity (GPP) is calculated separately for sunlit and shaded leaves (see Eq. (1-3) and Eq. (S1-S6)). The photosynthesis of sunlit and shaded leaves $A$ (i.e., $A_{sun}$ (unit: $\mu mol\ m^{-2}\ s^{-1}$) and $A_{shade}$ (unit: $\mu mol\ m^{-2}\ s^{-1}$) ) can be calculated as follows:

$$A = min(A_c, A_j) - 0.015 \times V_m \tag{S1}$$

where $A_c$ denotes the Rubisco-limited gross photosynthesis rate ($\mu mol\ m^{-2}\ s^{-1}$) and is computed as Eq. S2; $A_j$ is the RuBP-limited gross photosynthesis rate ($\mu mol\ m^{-2}\ s^{-1}$) and is calculated as Eq. S3.

$$A_c = V_m \frac{C_i - \Gamma}{C_i + K} \tag{S2}$$

$$A_j = J \frac{C_i - \Gamma}{4.5C_i + 10.5\Gamma} \tag{S3}$$

where $C_i$ is the intercellular $CO_2$ (Pa); $K$ is a function of enzyme kinetics (Pa) and is calculated as $K = K_C \times \left(1 + \frac{O_2}{K_O}\right)$; $O_2$ is oxygen concentrations in the atmosphere (Pa); $K_C$ and $K_O$ are the Michaelis-Menten constants for $CO_2$ (Pa) and $O_2$ (Pa), respectively; $\Gamma$ denotes the $CO_2$ compensation point without dark respiration (Pa) and is calculated as $\Gamma = 4.04 \times 1.75^{(T_a - 25)/10}$ ; $V_{cmax}$ is the maximum carboxylation rate ($\mu mol\ m^{-2}\ s^{-1}$) and $J$ represents the electron transport rate ($\mu mol\ m^{-2}\ s^{-1}$). The corresponding formulas for $V_m$ and $J$ are as follows:

$$V_m = V_{cmax25} \times 2.4^{\frac{T_a - 25}{10}} f(T_a) f(N) \tag{S4}$$

$$f(T_a) = \left\{1 + exp\left[\frac{-220000 + 710 \times (T_a + 273)}{8.314 \times (T_a + 273)}\right]\right\}^{-1} \tag{S5}$$

$$J = (29.1 + 1.64V_m) \times PPFD / (PPFD + 2.1 \times (29.1 + 1.64V_m)) \qquad (S6)$$

where $V_{cmax25}$ is the maximum carboxylation rate at 25°C ($\mu molm^{-2}s^{-1}$); Ta is air temperature (°C); $f(N)$ is the function of nitrogen (N) and is usually set to 0.5 in BEPS model (Liu et al., 1999; Zhang et al., 2018), which can adjust the photosynthesis rate for foliage nitrogen (Bonan, 1995). The $PPFD$ is the photosynthesis photon flux density ($\mu mol\ m^{-2}\ s^{-1}$).

When BEPS modelled the dynamics of carbon pools beyond the GPP, it stratified soil carbon stocks into 9 pools (i.e., surface structural litter, surface metabolic litter, soil structural litter, soil metabolic litter, coarse woody litter, surface microbe, soil microbe, slow, and passive carbon pools). These 9 carbon pools were used to calculate heterotrophic respiration ($R_h$) and autotrophic respiration ($R_a$). Eventually, the net ecosystem productivity (NEP) is calculated as the difference between GPP and $R_h$ and $R_a$.

$$NEP = GPP - R_h - R_a \qquad (S7)$$

References:

Liu, J., et al., 1997. A process-based boreal ecosystem productivity simulator using remote sensing inputs. Remote Sensing Environment, 62, 158-175.

Le Quere, C., 2018. Global carbon budget 2017. Earth System Science Data, 10, 405-448.

Farquhar, G.D., et al., 1980. A biochemical-model of photosynthetic $CO_2$ assimilation in leaves of C-3 Species. Planta, 149, 78-90.

Chen, J.M., et al., 1999. Daily canopy photosynthesis model through temporal and spatial scaling for remote sensing applications. Ecological Modelling, 124, 99-119.

Bonan, G.B., 1995. Land-atmosphere $CO_2$ exchange simulated by a land surface process model coupled to an atmospheric general circulation model. Journal of Geophysical Research, 100(D2): 2817-2831.

l. 148: flux partitioning is not quality control.

**Response:** Thanks. We have removed this inappropriate description from the text.

l. 151: ER not introduced

**Response:** The full name of ER has been added in the text, namely ecosystem respiration (ER).

l. 170: I am not an expert on this. Any reason why GOSIF was not used? I thought this would be the state-of-the-art GPP product.

**Response:** Yes, the Sun-induced chlorophyll *a* fluorescence (SIF) retrieved from satellites has shown potential as a remote sensing proxy for gross primary productivity

(GPP), such as GOSIF GPP. Generally, there are two approaches to estimating GPP based on SIF: one is to establish a direct empirical linear model of the two, and the other is based on the models, such as Soil-Canopy-Observation of Photosynthesis and the Energy Balance (SCOPE) model. The GOSIF GPP was not used in this study, mainly considering the following reasons:

(1) Most previous studies have shown that SIF and GPP can be characterized by linear relationships (Smith et al., 2018; Li et al., 2018; Li et al., 2019). However, some studies recently indicated a non-linear relationship between SIF and GPP (Kim et al., 2021; Liu et al., 2022), and the relationship between SIF-GPP varies across different climatic zones and biomes (Chen et al., 2021). All these results suggested that the relationship between SIF and GPP remains highly uncertain across space and time. This is mainly due to an insufficient understanding of the influencing factors of the relationship between SIF-GPP at present. For example, the GPP-SIF relationship is influenced by environmental factors and has a high sensitivity to precipitation. Especially, there will be differences in the trend of changes in SIF and GPP under drought stress conditions, and SIF offers limited potential for quantitatively monitoring GPP during heat waves (Wohlfahrt et al., 2018). However, most of the SIF-based GPP products including the GOSIF GPP were generated by the linear relationships between SIF and GPP to map GPP globally. Therefore, the current GPP products retrieved from SIF may have significant uncertainty and controversy due to insufficient understanding of the mechanism of the relationship between SIF-GPP (Chen et al., 2021; Liao et al., 2023).
(2) Currently, available GPP and SIF products are both known to have large systematic biases, particularly when the resolution is coarse (Frankenberg et al., 2014). Such biases could affect the observed SIF-GPP relationship, which in turn will affect the accuracy and quality of GPP products. Although Li et al., (2019) have produced relatively high-resolution GOSIF GPP products on a global scale, the raw data used for the GOSIF GPP production stems from SIF observed by the Orbiting Carbon Observatory-2 (OCO-2). However, the sparse coverage and coarse spatial resolution (~1°) of OCO-2 may also lead to large uncertainty in GOSIF GPP products.

Actually, we recognize that SIF brings major advancements in measuring terrestrial photosynthesis, especially in estimating GPP. We will consider SIF-based GPP in our future research.

References:

Smith, W.K., et al., 2018. Chlorophyll Fluorescence Better Captures Seasonal and Interannual Gross Primary Productivity Dynamics Across Dryland Ecosystems of Southwestern North America. Geophysical Research Letters, 45, 748–757.

Li, X., et al., 2018. Solar-induced chlorophyll fluorescence is strongly correlated with terrestrial photosynthesis for a wide variety of biomes: First global analysis based on OCO-2 and flux tower observations. Global Change Biology, 24, 3990–4008.

Li, X., Xiao, J., 2019. Mapping photosynthesis solely from solar-induced chlorophyll fluorescence: A global, fine-resolution dataset of gross primary production derived from OCO-2. Remote Sensing, 11(21), 2563.

Kim et al., 2021. Solar-induced chlorophyll fluorescence is non-linearly related to canopy photosynthesis in a temperate evergreen needleleaf forest during the fall transition. Remote Sensing of Environment, 258, 112362.

Liu et al., 2022. Non-linearity between gross primary productivity and far-red solar-induced chlorophyll fluorescence emitted from canopies of major biomes. Remote Sensing of Environment, 271, 112896.

Chen et al., 2021. Moisture availability mediates the relationship between terrestrial gross primary production and solar-induced chlorophyll fluorescence: Insights from global-scale variations. Global Change Biology, 27:1144–1156.

Miao, G., et al, 2018. Sun-induced chlorophyll fluorescence, photosynthesis, and light use efficiency of a soybean field from seasonally continuous measurements. Journal of Geophysical Research, 123, 610-623.

Wohlfahrt, G., et al., 2018. Sun-induced fluorescence and gross primary productivity during a heat wave. Scientific Reports, 8,14169.

Liao, Z., et al., 2023. A critical review of methods, principles and progress for estimating the gross primary productivity of terrestrial ecosystems. Frontiers in Environmental Science,11, 1093095.

Frankenberg, C., et al., 2014. Prospects for chlorophyll fluorescence remote sensing from the Orbiting Carbon Observatory-2. Remote Sensing of Environment, 147, 1–12.

l. 210: this reads strange. In S1 the land cover is fixed. But then you write that "in this scenario, LCC may lead to changes..."

**Response:** Thanks for catching the inappropriate description. We have removed the confusing sentence from the revised text.

l. 212: this is confusing. You talk about the conversion of forest to non-forest, and then about forest cover change. Is that not the same thing?

**Response:** Thanks again for catching the mistake. To avoid confusion, we have also removed the statement from the revised text where there are unnecessary.

Improve Table S3, explain more. What is remote sensing, what is modeled, etc.

**Response:** Thank you for the good suggestion. Following your suggestion, we have modified the Table S3 as follows:

**Table S3** Details of the published GPP products were used for model comparison.

| Dataset | Time Range | Spatial Resolution | Description | Source | References |
|---|---|---|---|---|---|
| MODIS GPP | 2000-2022 | 500 m | MODIS GPP product derived from satellite observations | https://ladsweb.modaps.eosdis.nasa.gov/archive/allData/6/MOD17A2H/ | Running et al. (2015) |
| EC-LUE GPP | 1982–2018 | 0.05° | EC-LUE GPP product derived from the light use efficiency model | https://doi.org/10.6084/m9.figshare.8942336.v3. | Zheng et al. (2020) |
| NIRv GPP | 1982–2018 | 0.05° | NIRv GPP product derived from satellite observations | https://doi.org/10.6084/m9.figshare.12981977.v2. | Wang et al. (2021) |
| VPM GPP | 2000-2016 | 0.05° | VPM GPP product derived from satellite observations and NCEP Reanalysis II climate data | https://figshare.com/articles/dataset/Annual_GPP_at_0_5_degree/5048005 | Zhang et al. (2017) |
| BEPS$_g$ GPP | 1982–2019 | 0.072727° | BEPS$_g$ GPP product derived from the process-based model | http://www.nesdc.org.cn/sdo/detail?id=612f42ee7e28172cbed3d809 | Chen et al. (2019); He et al. (2021) |

**Results:**

The model performance section is very good. But only GPP is evaluated. What about other model outputs?

**Response:** Thank you very much for this positive comment. In this study, we aim to understand how different drivers affect GPP changes. Therefore, we mainly focus on the validation and evaluation of GPP. In order to further validate the simulation results from the BEPS model, we also validated the simulated NEP at the three flux sites. We have listed the validation results of NEP in the supplementary (please see Table S5 and Figure S4-S6).

Also, Fig 3 does not really convince me. Can you discuss why the GPPs are so different?

**Response:** Thank you for this comment. We agree that there are relative differences between these GPP products. This stems mainly from the fact that different products are produced using different methods, data sources, etc, which may lead to differences in the GPPs produced. For example, the MODIS GPP product was mainly generated by the Terra/Aqua satellite observations. The newly released NIRv GPP was produced by near-infrared reflectance (i.e., the AVHRR reflectance from LTDR (Land Long Term Data Record v4) product). Thus, the data sources derived from divergent satellite observations may result in the differences between the two GPPs. Additionally, the EC-LUE GPP, VPM GPP, and BEPS GPP are all based on model outputs, where EC-LUE GPP and VPM GPP are simulated based on different light use efficiency (LUE) models, respectively, and the BEPS GPP is produced based on a process model. So, the parameters, inputs, and model structure of different models are inconsistent, which may also lead to differences in GPP production.

Although these products have differences and were used for comparison in this study, we mainly consider that these GPP products have been widely used in previous studies (NIRv GPP: Zhang et al., 2022; MODIS GPP: Yao et al., 2020; VPM GPP: Zhang et al., 2016; BEPS GPP: Chen et al., 2019; EC-LUE GPP: Wang et al., 2020). Especially, Xing et al., (2023) also adopted the same global GPP products for comparison with the GPP simulated by BEPS over China. Moreover, these products are produced from different data sources and methods, and it would be more reasonable and reliable to use them for comparing the simulated GPP in our study.

To respond to your question, we have moved Figure 3 to the supplementary, mainly because Figure 3 is relatively less important for the understanding of the main text, and on the other hand, it also can reduce the number of figures in the main text.

References:

Zhang et al., 2022. Revisiting the cumulative effects of drought on global gross primary productivity based on new long-term series data (1982–2018). Global Change Biology, 28, 3620–3635.

Yao et al., 2020. Accelerated dryland expansion regulates future variability in dryland gross primary production. Nature Communications, 11, 1665.

Zhang et al., 2016. Consistency between sun-induced chlorophyll fluorescence and gross primary production of vegetation in North America. Remote Sensing of Environment, 183, 154-169.

Chen, J.M. et al., 2019. Vegetation structural change since 1981 significantly enhanced the terrestrial carbon sink. Nature Communications, 10(1): 4259.

Wang et al., 2020. Recent global decline of $CO_2$ fertilization effects on vegetation photosynthesis. Science, 370, 1295-1300.

Xing et al., 2023. Modeling China's terrestrial ecosystem gross primary productivity with BEPS model: Parameter sensitivity analysis and model calibration. Agricultural and Forest Meteorology, 343, 15, 109789.

l. 242: typo: "203-2010"

**Response:** The "203-2010" has been changed to "2003-2010".

l. 240-245: any explanation as to why some of the sites are performing much better? R2 as low as 0.43 in one site, up to 0.85 in another

**Response:** Thanks for this comment. Yes, our validation results show that the performance of the model in simulating the GPP at the three flux sites is different. This may be due to the following reasons:

(1) On the one hand, it may be due to differences in geographic location, topographic features, climate and water variability, complex structure and composition of community, and soil types at different flux sites, leading to inconsistent performance of the model in simulating GPP. Generally, there are a large number of parameters were set as constants in the model, even for the same PFT. Thus not considering the spatial and temporal variability of these parameters, which may cause differences in the accuracy of the simulation results at different sites. For example, the elevations of the three flux sites are 100 m for QYZ, 300 m for DHS, and 2400 m for ALS, respectively. The mean annual temperature and (°C) and annual precipitation (mm) of these sites are also different. Therefore, these factors may result in variability in simulation results.

(2) On the other hand, the quality and accuracy of the observations vary from site to site due to differences in observation equipment (e.g., the eddy covariance technique), topography, data quality controls, etc., which may also affect our validation results. For example, as reported by Wang et al., (2006), the low observed values of $CO_2$ flux are mainly caused by a $CO_2$ leak during the nighttime at the DHS. In addition, the effect of topography also leads to generally low fluxes in the southerly direction at this site (Li et al., 2021).

We also reviewed previous studies and found similar results to our study. For example, Muhammad et al., (2022) simulated the GPP at DHS station based on an improved process model and it had an $R^2$ of only 0.38. He et al., (2013) also reported the $R^2$ between the BEPS-simulated GPP and EC-based GPP for the same site (DHS) was 0.48, but the $R^2$ was 0.78 for the QYZ. Zeng et al., (2020) used the Random forest model to simulate global GPP and showed that there was a relatively low $R^2$ ($< 0.5$) in the DHS site when comparing their simulated results with global flux data sets. These results indicate that there may be relatively low-quality issues with observed flux data from DHS.

References:

Muhammad A., et al., 2021. Reflectance and chlorophyll fluorescence-based retrieval of photosynthetic parameters improves the estimation of subtropical forest productivity. Ecological Indicators, 131, 108133.

He, M., et al., 2013. Development of a two-leaf light use efficiency model for improving the calculation of terrestrial gross primary productivity. Agricultural and Forest Meteorology,173, 28–39.

Zeng et al., 2020. Global terrestrial carbon fluxes of 1999–2019 estimated by upscaling eddy covariance data with a random forest. Scientific Data, 7, 313.

Wang et al., 2006. $CO_2$ flux evaluation over the evergreen coniferous and broad-leaved mixed forest in Dinghushan, China. Science in China Series D: Earth Sciences, 49, 127–138.

Li et al., 2021. An observation dataset of carbon and water fluxes in a mixed coniferous broad-leaved forest at Dinghushan, Southern China (2003 – 2010). China Scientific Data, 6(1), DOI: 10.11922/csdata. 2020. 0046.zh.

Fig 2: do you have any explanation about the small bias in DHS at low observed values? This is also visible in all years in the supplements.

**Response:** Thanks. As mentioned above, the small bias may be caused by the observations of the flux tower itself. As reported by Wang et al., (2006), the low observed values of $CO_2$ flux are mainly caused by a $CO_2$ leak during the nighttime at the site. In addition, the effect of topography also leads to generally low fluxes in the southerly direction (Li et al., 2021). Despite the presence of lower observations at the DHS, the small bias is systematic errors and it may not affect the validation of our model. Besides, at the other two stations (e.g., QYZ and ALS), our validation results confirmed the good performance of the model used in this study.

References

Wang et al., 2006. $CO_2$ flux evaluation over the evergreen coniferous and broad-leaved mixed forest in Dinghushan, China. Science in China Series D: Earth Sciences, 49, 127–138.

Li et al., 2021. An observation dataset of carbon and water fluxes in a mixed coniferous broad-leaved forest at Dinghushan, Southern China (2003 – 2010). China Scientific Data, 6(1), DOI: 10.11922/csdata.2020.0046.zh.

Fig 2: The caption misses that the dots are observations

**Response:** Thanks. The dark circles represent the observations. We added the description of the green lines and dark circles in the Figure caption (see below).

"Figure 2 Comparison of simulated GPP with measured GPP from three flux tower stations at daily (a-c) and annual (d-f) scales. The green lines and dark circles represent the simulated GPP and observed GPP, respectively."

l. 266: This is an issue: obviously the increase in GPP is similar in a study with the same model. The next data product has a much lower increase, 0.017, compared to this study's 0.026.

**Response:** Thank you very much for catching an error in the description. For more clarification, we have removed this sentence from the revised text.

BEPS simulates a higher GPP compared to all the other products, and a higher trend, too. This needs to be discussed further.

**Response:** True. Our simulated GPP is slightly higher than other products. Firstly, there are some uncertainties and substantial differences in the simulated interannual variability in GPP from various ecosystem models due to many differences in model

structure, parameterization and driving data (Cai et al., 2014; Lin et al., 2023). Secondly, in this study, our simulated GPP is mainly compared with other GPP products generated by the LUE model-based and remote sensing-based models. However, previous studies (Zhu et al., 2018; O'Sullivan et al., 2020; Wang et al., 2023) reported that LUE-based model, remote sensing-based models, machine-learning-based model, and some terrestrial ecosystem models may underestimate the GPP at an annual scale. For example, the GPP estimates by the LUE models mainly depend on a few important factors, including solar radiation, air temperature, water availability, and vegetation indexes (e.g., EVI or NDVI). Current LUE-based models do not completely integrate the other key environmental regulations to vegetation productivity, such as the effect of atmospheric $CO_2$ concentration, canopy structure (e.g., LAI), diffuse radiation, etc. on GPP. Therefore, one cause of the underestimation in other GPP products is possibly failure to assess the $CO_2$ fertilizer effects, because almost no apparent response to the rising atmospheric $CO_2$ concentration in the LUE models leads to an underestimated trend (Anav et al., 2015). In our study, the GPP was estimated by a process-based model (i.e., BEPS) that considers the effects of these important factors on GPP, especially the $CO_2$ fertilization effect, which may lead to a higher GPP compared to all the other products.

For what it's worth, the results of our comparisons showed that the interannual trends of our simulated results were in line with other GPP products (Fig. S8). Despite possible overestimation, the purpose of this study mainly focuses on the trends and explains the driving mechanism behind them, thus it may not affect our results and conclusions. The above discussion has been added to the revised version.

References

Cai, W., et al., 2014. Large differences in terrestrial vegetation production derived from satellite-based light use efficiency models. Remote Sensing, 6(9), 8945–8965.

Lin et al., 2023. Underestimated Interannual Variability of Terrestrial Vegetation Production by Terrestrial Ecosystem Models. Global Biogeochemical Cycles, 34(4), e2023GB007696.

Zhu et al., 2018. Underestimates of Grassland Gross Primary Production in MODIS Standard Products. Remote Sensing, 2018, 10(11), 1771.

Wang et al., 2023. Assessment of Six Machine Learning Methods for Predicting Gross Primary Productivity in Grassland. Remote sensing, 15(14), 3475.

O'Sullivan, M., et al. 2020. Climate-driven variability and trends in plant productivity over recent decades based on three global products. Global Biogeochemical Cycles, 34(12), e2020GB006613.

Anav, A., et al., 2015. Spatiotemporal patterns of terrestrial gross primary production: a review. Reviews of Geophysics, 53(3), 785-818.

l. 276: what do you mean by simulated actual GPP?

**Response:** Thanks for the comment. Here, the simulated actual GPP represents the GPP in the actual situation, i.e. under the interactive influence of different drivers (e.g., climate change, vegetation change, etc.), which is different from the GPP under other scenario simulations, such as the climate change-induced GPP.

l. 280: grammar

**Response:** Thanks for catching this error. Revised text to:

"Spatially, 90.4% of forest areas in the study area showed an increasing trend in GPP, while 9.6% of forest areas exhibited a decreasing trend in GPP."

l. 290s: streamline this section to make clear that the change in GPP comes from the increasing/decreasing areas

**Response:** Thanks for the suggestion. We have streamlined this paragraph as follows:

"Based on the ESA CCI land cover data, it showed the area of gains or losses for different forest types between 2001 and 2018 (Fig. 5a). We found that FCC positively affected the entire forest GPP at a rate of 1.35 TgC year$^{-1}$ (p = 0.000) (Fig. 5b), mainly driven by EBF GPP (1.17 TgC year$^{-1}$, p = 0.001) and MXF GPP (2.15 TgC year$^{-1}$, p = 0.000). However, the FCC had a negative effect on the DBF GPP and ENF GPP variations at the rate of -0.05 TgC year$^{-1}$ (p = 0.195) and -1.92 TgC year$^{-1}$ (p = 0.000), respectively. Spatially, 92.2% of the total forest GPP showed a stable state, and only 7.8% of GPP exhibited an increase or decrease under the effect of FCC (Fig. 5c). Among them, 3.9% of the forest GPP increased significantly, mainly located in the western region (e.g., the south slope of the Qinling mountains, the southwest karst region), while 2.6% of the forest GPP was significantly reduced in the eastern regions, which belong to the ENF (Fig. 5)."

l. 305: In section 3.3.2, the point needs to be better explained that although climate change contributes to a 1.11 TgC/year most of the area has a decreasing trend. This increase seems to stem from a small region in the west. What is happening in this region? E.g. Fig 6b

**Response:** Thanks for the comment. The main vegetation types in the small regions (the area you mentioned) located in the south of Tibet are natural broad-leaved evergreen forests (Cheng et al.,2023), and they are in middle age and in range of 40-60 years old, which has a strong carbon sequestration potential (Zhang et al.,2017; Zhang et al.,2014). The magnitude of GPP increase (see the legend in Fig. 6) in the small areas is significantly higher than in other regions because temperature, precipitation, and radiation all contribute to GPP increase in this region (Fig. 6). Although the area of GPP reduction due to climate change is relatively large, its impact magnitude is relatively small, resulting in smaller areas with higher magnitude offsetting the larger area of GPP decrease.

References:

Cheng, K., Chen, Y., Xiang, T., Yang, H., Liu, W., Ren, Y., Guan, H., Hu, T., Ma, Q., and Guo, Q.: 2020 forest age map for China with 30 m resolution, Earth Syst. Sci. Data Discuss. [preprint], https://doi.org/10.5194/essd-2023-385, in review, 2023.

Zhang, Y., et al., 2017. Mapping spatial distribution of forest age in China. Earth and Space Science,4, 108–116.

Zhang, C., et al., 2014. Mapping forest stand age in China using remotely sensed forest height and observation data. Journal of Geophysical Research: Biogeosciences,119,1163–1179.

l. 346: why is LAI increasing at all?

**Response:** Thanks. The LAI indeed shows the increasing trend for the different subtropical forests in our study during 2001-2018. This is in line with many previous studies that reported the greening (using LAI as an indicator) of our Earth due to different driving factors (e.g., climate change, land cover change, etc.) during the past 30 years (Zhu et al., 2016; Tong et al., 2018; Chen et al., 2019; Tong et al., 2020; Chen et al., 2020). Especially in the southern region of China, there is a significant increase in forest LAI, and the main driving factors for the increase in LAI are climate change (Zhu et al., 2016) and ecological engineering projects (e.g., afforestation and reforestation projects) (Tong et al., 2018; Chen et al., 2019; Tong et al., 2020; Chen et al., 2020).

References:

Zhu et al., 2016. Greening of the Earth and its drivers. Nature Climate Change, 6, 791–795.

Tong et al., 2018. Increased vegetation growth and carbon stock in China karst via ecological engineering. Nature Sustainability, 1, 44–50.

Chen et al., 2019. China and India lead in greening of the world through land-use management. Nature Sustainability, 2, 122–129.

Tong et al., 2020. Forest management in southern China generates short term extensive carbon sequestration. Nature Communications, 11, 129.

Chen et al., 2020. Afforestation promotes the enhancement of forest LAI and NPP in China. Forest Ecology and Management, 462, 117990.

l. 349 and in general: The wording "Especially, the positive effect of VSC on EBF" is strange. I mean, the VSC change inside the EBF and that led to a change in GPP in those forests.

**Response:** We are sorry for the confusion. We have reworded the following sentence to make it clear in the revised text.

"Especially, the LAI change significantly promotes EBF GPP increase at a rate of 1.64 TgC year[-1] (p = 0.025)."

Fig S10: There is a rapid increase in trend around 2011. Why is that? Also, how does LAI look in the model pre-2000?

**Response:** As reported by many studies (Zhu et al., 2016; Tong et al., 2018; Chen et al., 2019; Tong et al., 2020; Chen et al., 2020), the LAI showed a significant increase over the past two decades. Especially, the Chinese government has made an enormous investment to implement some key ecological restoration programs since 2000. Lu et al., (2015) indicated that the vegetation had a relatively stable status from 2000 to 2010. After 2010, the vegetation may begin to show significant growth. This may be due to the lagged response of vegetation to these measures. Therefore, there was a rapid increase in trend around 2011. Based on different vegetation indices (e.g., LAI), Chen et al. (2021b) also demonstrated a turning point in vegetation change in China around 2010. They also found that the GPP and LAI increased significantly after 2010 mainly driven by the climatic factors and ecological restoration programs.

Based on your suggestion, we also compared the changes in LAI before and after 2001. It indicates that there is also an upward trend in LAI before 2001 (see figure on the left and right).

[Figure]

Figures showing annual changes of GLASS LAI for entire forest region and different forest types before and after 2001. EBF: evergreen needleleaf forest; DBF: deciduous broadleaf forest; ENF: evergreen needleleaf forest; MF: mixed forest.

References:

Zhu et al., 2016. Greening of the Earth and its drivers. Nature Climate Change, 6, 791–795.

Tong et al., 2018. Increased vegetation growth and carbon stock in China karst via ecological engineering. Nature Sustainability, 1, 44–50.

Chen et al., 2019. China and India lead in greening of the world through land-use management. Nature Sustainability, 2, 122–129.

Tong et al., 2020. Forest management in southern China generates short term extensive carbon sequestration. Nature Communications, 11, 129.

Chen et al., 2020. Afforestation promotes the enhancement of forest LAI and NPP in China. Forest Ecology and Management, 462, 117990.

Lu et al., 2015. Recent ecological transitions in China: greening, browning, and influential factors. Scientific Reports, 5, 8732.

Chen, Y. et al., 2021b. Accelerated increase in vegetation carbon sequestration in China after 2010: A turning point resulting from climate and human interaction. Global Change Biology, 27(22), 5848-5864.

l. 355: You write:

"results showed that most GPP increases in China's subtropical forests due to the increase of LAI, which also offset the negative effects of VSC on GPP, thus allowing VSC to play a key driving factor in promoting GPP increases throughout the forest area."

This is confusing. Did you mean FSC maybe instead of VSC at the first mention? LAI is the same as VSC, right? So how does the effect of change in LAI on GPP offset the effect of change in LAI on GPP? They are the same thing? Or do you mean, there is more positive change that heavily offsets the negative changes?

**Response:** We are sorry for the confusion. FSC in this study represents forest cover changes, while VSC indicates vegetation structure changes. As you suggested above, we have changed VSC to LAI to avoid confusion. Yes, there is the more positive change that heavily offsets the negative changes. Because not all pixels show the same changes in space (see Figure 7), some pixels have a positive GPP trend due to the influence of LAI, while others may have a negative trend. Meanwhile, the magnitude of the pixels with positive trends is larger than that of the pixels with negative trends, which results in these pixels cancelling each other out.

For clear understanding, revised text to:

"Overall, there are more positive changes in GPP due to the effect of LAI that heavily offsets the negative changes, ultimately resulting LAI to be the major factor to GPP increases throughout China's subtropical forests."

l. 361: verb is missing

**Response:** As suggested, we have revised this sentence as follows:

"The annual mean $CO_2$ concentration increased from 371.3 ppm to 408.7 ppm during 2001-2018."

Fig. 9: This is a nice figure that shows the main results.

**Response:** Thank you very much for this encouraging comment.

Fig 9: I am puzzled that, e.g., in b) CC-ALL is nowhere near the sum of the three. I understand that there will be interactions, but I find it quite strange that the interactions are quite strongly positive but each of the components is almost 0. Maybe these cancel each other out over the entire region.

**Response:** We are sorry for the confusion. We agree with your comment that the impact of each climatic factor is almost 0, while the interactions seem to be strong. However, the overall effects of different factors may not be able to be simply added together. Although their individual effects may be small, their interactive effects may become relatively large, because their interactions are not simply linear relationships in our model. This is where we differ from previous studies (Zhang et al., 2014; Zhang et al., 2022) that used relatively simple models to detect single or interactive effects of different factors. In this study, the effects of these factors are analysed using a process-based model in which the different factors are nonlinearly related to each other. Actually, this is also the purpose of this study, which is to try to unravel the possible effects of different drivers on GPP changes individually and interactively based on a process model.

References:

Zhang et al., 2014. Effects of land use/land cover and climate changes on terrestrial net primary productivity in the Yangtze River Basin, China, from 2001 to 2010. Journal of Geophysical Research: Biogeosciences, 119, 1092–1109.

Zhang, X. et al., 2022. Land cover change instead of solar radiation change dominates the forest GPP increase during the recent phase of the Shelterbelt Program for Pearl River. Ecological Indicators, 136: 108664.

Results: when you describe the changes for each of the forest types, I believe the results stem solely from the changing areas. It would be better to show the changes on a per-area basis or in the simulations even keep the forest cover stable...

**Response:** Thank you for the comment! Yes, when we investigated the impact of forest cover changes on GPP variations, the results stem solely from the changing areas of different forest types. In other simulations, we do keep different forest cover unchanged, and it would not be influenced by forest cover change.

**Discussion:**

l. 416: "which is mainly converted from cropland". You need to elaborate here. Croplands can be highly productive. A few models even indicate that in some regions in China, cropland could potentially be more productive than forests in terms of GPP (Fig. 3 in https://doi.org/10.1038/s41598-022-23120-0). To back your claim, can you provide some numbers here on GPP values of the crops that have been reforested?

**Response:** Thank you very much for the suggestion. Yes, Krause et al., (2022) suggested that cropland could potentially be more productive than forests in terms of GPP, while the suitable area was mostly in Central Africa, Indonesia and northern

Australia, western North America, and parts of the Amazon. Indeed, the findings derived from 3 models of Krause et al. (2022) (Figure 3) indicated that some regions of China have higher productivity of cropland. However, the results derived from the other 4 models also showed that the forests were the most productive land cover when compared with grasslands and croplands in the subtropical region of China. Therefore, there may be some uncertainties in their study.

As suggested, we have provided some numbers here on the GPP values of the crops that have been reforested. Here, we take the conversion of cropland to MXF as an example, we counted the changes in GPP resulting from the conversion of cropland to MXF. We found that the GPP value in the changed area was 7.48 TgC in 2001 and increased to 7.64 TgC in 2018 due to the conversion of cropland to MXF. Revised text to:

"For example, after the conversion of cropland to MXF, GPP in this changed area increased by 0.16 Tg C between 2001 and 2018."

l. 419: what do you mean by the negative effect of a specific forest type on forest GPP variations? That the planting of a certain forest type may result in a lower GPP than the previous land cover? Or something else?

Generally in this section, you need to be careful with the wording as you refer to "forest GPP" most of the time, but sometimes you mean the GPP of the entire area.

**Response:** Thank you very much for the comment and suggestion. Here, we mean that the changes in the area of a specific forest type may lead to changes in the total GPP of a certain forest type. For example, an increase in the EBF area leads to an increase in its GPP in our study, while a decrease in the ENF area results in a decrease in its GPP. We are sorry for the confusion. We have reworded the statement to revised text: "which may ignore the different effects of a specific forest type on forest GPP variations.". As you mentioned in a previous study (Krause et al., 2022), the planting of a certain forest type may result in a lower GPP than the previous land cover at the global scale. However, Krause et al., 2022 indicated that croplands are most productive in 21% of the suitable area, mostly in Central Africa, Indonesia and northern Australia, western North America, and parts of the Amazon. In our study, we also found that a decrease in ENF area led to a decrease in GPP, while an increase in EBF and MXF area led to an increase in GPP. This implies that forests have higher productivity, which is consistent with the findings of Krause et al., (2022), who also showed that forests are more productive in the subtropical region of China. Moreover, we have harmonized the description of forest GPP, and removed the statement "the GPP of the entire area" from the revised text.

References:

Krause et al., 2022. Quantifying the impacts of land cover change on gross primary productivity globally. Scientific Reports, 12, 18398.

l. 447ff: citations for the claim? Also, drought relates more to precipitation, maybe you can instead mention increased VPD as a result of a high temp increase.

**Response:** Thank you again for the suggestion. Following your suggestions, we have added some citations in the revised text (see below). We also reworded this sentence to mention increased VPD as a result of a high temperature increase.

"Many studies suggested that an increment in temperature can benefit the vegetation productivity (Myneni, et al., 1997; Nemani, et al., 2003; Song et al., 2022), or could reduce the vegetation productivity due to increased VPD as a result of a high temperature increase (Yuan et al., 2019; Lopez et al., 2021)."

References:

Myneni, R. B., et al., 1997. Increased plant growth in the northern high latitudes from 1981 to 1991. Nature, 386, 698–702.

Nemani, R. R., et al., 2003. Climate-driven increases in global terrestrial net primary production from 1982 to 1999. Science, 300, 1560–1563.

Song, Y., et al., 2022. Increased global vegetation productivity despite rising atmospheric dryness over the last two decades. Earth's Future, 10, e2021EF002634.

Yuan, W. P., et al., 2019. Increased atmospheric vapor pressure deficit reduces global vegetation growth. Science Advances, 5, eaax1396.

Lopez, J., et al., 2021. Systemic effects of rising atmospheric vapor pressure deficit on plant physiology and productivity. Global Change Biology, 27, 1704–1720.

l. 450 mention again the magnitudes. They should explain that the smaller area of increase outweighs the larger area of decrease

**Response:** Thank you very much for the suggestion. As the same responses mentioned above. We also added the following explanation to the revised text:

"The main vegetation types in the small regions are natural broad-leaved evergreen forests (Cheng et al.,2023), and they are in middle age and in the range of 40-60 years old, which has a strong carbon sequestration potential (Zhang et al.,2017; Zhang et al.,2014). The magnitude of GPP increase (see the legend in Fig. 6) in the small areas is significantly higher than in other regions because temperature, precipitation and radiation all contribute to GPP increase in this region (see Fig. 6). Although the area of GPP reduction due to climate change is relatively large, its impact magnitude is relatively small, resulting in smaller areas with higher magnitude offsetting the larger area of GPP decrease."

References:

Cheng, K., Chen, Y., Xiang, T., Yang, H., Liu, W., Ren, Y., Guan, H., Hu, T., Ma, Q., and Guo, Q.: 2020 forest age map for China with 30 m resolution, Earth Syst. Sci. Data Discuss. [preprint], https://doi.org/10.5194/essd-2023-385, in review, 2023.

Zhang, Y., et al., 2017. Mapping spatial distribution of forest age in China. Earth and Space Science,4, 108–116.

Zhang, C., et al., 2014. Mapping forest stand age in China using remotely sensed forest height and observation data. Journal of Geophysical Research: Biogeosciences,119,1163–1179.

l. 461: why is that?

**Response:** Thanks for the comment. As shown in Fig. 1, ENF is mainly distributed in the eastern and western regions of the subtropics. Moreover, our results also showed that climatic factors (e.g., temperature, precipitation, and solar radiation) in these regions have negative effects on the GPP of ENF (Fig. 6), and particularly the solar radiation declined significantly in the eastern region, which led to a decrease in the GPP of ENF in the east. For EBF, it is mainly distributed in the central and western regions where climate change mainly contributes to the increase of GPP of EBF.

For clear understanding, revised text to:

"climate change has a positive effect on the GPP of EBF, but a negative effect on the GPP of ENF. The main reason is that ENF is predominantly located in the eastern and western parts of the subtropics (Fig. 1). In these areas, individual climatic factors (e.g., temperature, precipitation, and solar radiation) or their interactions caused the GPP of ENF decrease (Fig. 6), and particularly the solar radiation declined significantly in the eastern region, which led to a decrease in the GPP of ENF in the east. The EBF is mainly distributed in the central and western regions (Fig. 1) where climate change mainly contributes to the increase of EBF GPP (Fig. 6)."

l. 488: Forest protection has greater carbon uptake potential than what? This also relates to my comment on l. 416. Also, you only refer to GPP. Can you make any claims on NPP?

**Response:** Thanks again for your comment and suggestion. We are sorry for the confusion. Considering this sentence is not necessary, we have removed the confusing sentence from the revised text. As suggested, we also added some claims on NPP to the revised text as follows.

"Consistent with our study period (2001–2018), Chen et al. (2021b) reported an increase in vegetation carbon sequestration in China based on the two indicators of GPP and NPP, especially with an accelerated increase in carbon sequestration potential after 2010. They showed that GPP and NPP in China increased obviously at the rate of 49.1–53.1 TgC/yr$^2$ and 22.4–24.9 TgC/yr$^2$, respectively, and the significant increase of GPP and NPP was highly attributed to human activities (e.g., ecological

restoration projects) in southern and eastern China, especially the human-induced NPP gains can offset the climate-induced NPP losses in southern China. Based on the Carnegie-Ames-Stanford (CASA) model, Li et al. (2021) stated that NPP in China showed an increasing trend of 15.2 TgC/yr, and that the humid region (i.e., most of southern China) dominated the interannual variation of the NPP. Using NPP as an indicator, many previous studies have also reported similar results that the carbon sequestration potential of the tropical and subtropical forests has increased over the past few decades (Wang et al., 2008; Shang et al., 2023)."

References:

Chen, Y. et al., 2021b. Accelerated increase in vegetation carbon sequestration in China after 2010: A turning point resulting from climate and human interaction. Global Change Biology, 27(22), 5848-5864.

Li et al., 2021. Regional contributions to interannual variability of net primary production and climatic attributions. Agricultural and Forest Meteorology, 303, 108384.

Wang et al., 2008. Spatiotemporal dynamics of forest net primary production in China over the past two decades. Global and Planetary Change, 61(3–4), 267-274.

Shang et al., 2023. China's current forest age structure will lead to weakened carbon sinks in the near future. The Innovation, 4 (6), 100515.

section 4.1.4: here I also find that some discussion on the relation of GPP to carbon sequestration is missing.

**Response:** Following your suggestion, we have added the following discussion to the revised text:

"The carbon sequestered by vegetation through photosynthesis in a given unit of space and time, i.e., gross primary productivity (GPP), forms the fundamental part of the carbon cycle (Monteith 1972). GPP is a crucial indicator for estimating the carbon sequestration capacity of ecosystems (Chen et al., 2021b; Ma et al., 2019), which reflects the largest carbon sequestered by photosynthesis in carbon budget (Christian et al., 2010; Xu et al., 2019). Moreover, GPP drives land carbon sequestration and partly offsets anthropogenic $CO_2$ emission, which significantly affects global carbon balance and climate change (Lan et al., 2021; Running et al., 2008). However, the distribution and dynamics of terrestrial GPP are significantly affected by global environmental changes (Piao et al., 2015; Chen et al., 2021b). Even minor changes in GPP may have a significant effect on regional and global carbon balance (Yao et al., 2018). Investigating the variations of GPP and their drivers at the spatial-temporal scale is crucial for human beings to understand the changes in carbon sequestration in terrestrial ecosystems and are conducive to making appropriate ecological and environmental management decisions (Andersson et al., 2009)."

References:

Monteith, J. L., 1972. Solar-radiation and productivity in tropical ecosystems. Journal of Applied Ecology, 9(3), 747-766.

Chen, Y. et al., 2021b. Accelerated increase in vegetation carbon sequestration in China after 2010: A turning point resulting from climate and human interaction. Global Change Biology, 27(22), 5848-5864.

Ma, J., 2019. Trends and controls of terrestrial gross primary productivity of China during 2000–2016. Environmental Research Letters, 14, 084032.

Christian, B., et al., 2010. Terrestrial gross carbon dioxide uptake: Global distribution and covariation with climate. Science, 329 (5993), 834–838.

Xu, C. et al., 2019. Increasing impacts of extreme droughts on vegetation productivity under climate change. Nature Climate Change, 9, 948–953.

Running, S.T., 2008. Ecosystem Disturbance, Carbon, and Climate. Science, 321 (5889), 652-653.

Piao et al., 2015. Detection and attribution of vegetation greening trend in China over the last 30 years. Global Change Biology, 21 (4), 1601-1609.

Yao, Y., et al. 2018. Spatiotemporal pattern of gross primary productivity and its covariation with climate in China over the last thirty years. Global Change Biology, 24, 184–196.

Andersson, K., 2009. National forest carbon inventories: policy needs and assessment capacity. Climatic Change, 93, 69–101.

**Conclusion**

l. 560ff: I am not sure about this last concluding statement. You basically show that changes in the vegetation structure have a strong impact on GPP. You don't show anything about NPP or NEE. I would doubt that the growth of an entire new forest would have a lower impact on the carbon balance than improving the current ones. At least this claim cannot be made based on your work.

**Response:** Thank you again for the suggestion. We agree that there is some confusion in this statement. To avoid confusion, we have removed the statement where there are unnecessary.

---

## Author Response (AR1)

**RC1**: 'Comment on bg-2023-140', Anonymous Referee #1, 29 Oct 2023

Dear Reviewer and Editor,

We would like to thank the reviewer and you for your interest in our study, and for the feedback provided. We appreciate these constructive and specific comments, which will help improve the quality of the manuscript. We have carefully inspected all reviewer comments. We also streamlined the results, figures and text as suggested by reviewers. Below, you will find our responses to the comments (responses in blue).

We hope that you will find the result satisfying.

Sincerely,

Tao Chen, Félicien Meunier, Marc Peaucelle, Guoping Tang, Ye Yuan, Hans Verbeeck

**Reviewer #1**

In their study, the authors investigate the influence of different drivers on changes in GPP in subtropical forests in China. The considered drivers were climate change, forest cover change, change in vegetation structure, and changes in $CO_2$ concentrations.

The authors use the BEPS model and run multiple simulations to disentangle the impact of the different drivers and find that atmospheric $CO_2$ and vegetation structure play the most important roles.

This is an interesting and well-conducted study and the manuscript is decently written. In my opinion, this study can be published in Biogeosciences after it went through some major revisions.

I mainly find that there needs to be some more model evaluation. Furthermore, some results need to be explained better. Also, the discussion has some points that need to be made clearer or added (see details below).

I'd further suggest some streamlining of results, figures, and text. There are 10 Figures, often with 6 panels. I believe this could be made more concise.

Further detailed comments follow below.

**Response:** Thank you very much for the valuable comments and suggestions. Below we go through point-by-point our answers to the comments, and our responses are in blue. Moreover, we have also streamlined the results, figures and text as suggested. Especially, six most important Figures remain in the main text. The rest do not necessarily need to be placed in the main text and have been moved to the supplementary.

**Abstract:**

Why call it VSC and not just LAI?

**Response:** Thank you for this comment. LAI is one of the most important parameters representing vegetation structure, which can influence the carbon cycle and is widely used in models (e.g., LUE-based models and process-based models) to simulate carbon and water fluxes (Chen et al., 2019; Zhang, X. et al., 2022). Thus, the VSC was adopted in our study to represent LAI. Indeed, LAI does not represent all vegetation structure changes. As suggested, we use LAI directly in the revised version.

References:

Chen, J.M. et al., 2019. Vegetation structural change since 1981 significantly enhanced the terrestrial carbon sink. Nature Communications, 10(1): 4259.

Zhang, X. et al., 2022. Land cover change instead of solar radiation change dominates the forest GPP increase during the recent phase of the Shelterbelt Program for Pearl River. Ecological Indicators, 136: 108664.

**Introduction**

l. 39: the statement about the 30% is not a result of the cited study and is also not cited there... Please find a better reference

**Response:** Thank you for this suggestion. We added the new reference in the revised version (see below).

"Giovanni Forzieri, et al, 2022. Emerging signals of declining forest resilience under climate change. Nature, 608, 534–539."

l. 55: should be 0.82 billion I guess.

**Response:** Agree! The 8.2 billion has been changed to 0.82 billion.

l. 59: is this compared to global surface temp or temp over land?

**Response:** Thanks. It is compared to the global surface temperature. We have rewritten the sentence as follows (see Page 3, Lines 87-89).

"the annual mean temperature in the Chinese subtropical monsoon region has increased by more than 1.0 °C over the past 30 years (Fang et al., 2018), which was higher than the global surface temperature increase (Sun et al., 2019)."

References:

Sun, C., et al, 2019. Changes in extreme temperature over China when global warming stabilized at 1.5 °C and 2.0 °C. Scientific Reports, 9:14982.

Fang, J., et., 2018. Climate change, human impacts, and carbon sequestration in China. Proceedings of the National Academy of Sciences, 115(16): 4015-4020.

**Methods:**

l. 103: what about the spread of temperature as you mention for precipitation?

**Response:** Thanks for this comment. We added the following sentence to the revised version to describe the spread of temperature (see Page 5, Lines 152-153).

"The mean annual temperature of the study area is about 15.5°C, and it normally increases from the northwest toward the southeast."

l. 115: NEP was not introduced. Generally, a glossary with abbreviations would be helpful.

**Response:** We have added the full name of the NEP (i.e., net ecosystem productivity) in the revised text. As suggested, we also added the following glossary of acronyms in the revised text to show abbreviations for other terms (see Page 2, Lines 39-43).

List of Abbreviations

| Abbreviation | Definition |
| --- | --- |
| BEPS | The Boreal Ecosystem Productivity Simulator |
| GPP | Gross primary productivity |
| FCC | Forest cover change |
| LAI | Leaf area index |
| CC | Climate change |
| $CO_2$ | Carbon dioxide |
| EBF | Evergreen broadleaved forest |
| ENF | Evergreen needle-leaved forest |
| DBF | Deciduous broadleaved forest |
| MXF | Mixed forest |
| QYZ | Qianyanzhou station |
| DHS | Dinghushan station |
| ALS | Ailaoshan station |
| $V_{cmax}$ | The maximum carboxylation rate |
| NEP | Net ecosystem productivity |
| ER | Ecosystem respiration |

l. 116: some more text on the model is necessary to allow the reader to get a basic understanding of it. It may go into the supplements.

**Response:** Thank you for the constructive suggestion. Following your suggestion, we have added more descriptions (please see below) about the model in the supplementary (see also Text S1).

"Text S1 (description of the BEPS model)

    The BEPS model was originally developed at the Canada Centre for Remote Sensing to assist in natural resources management (Liu et al., 1997). Compared with 15 prognostic models that participated in the Global Carbon Project (GCP) (Le Quere et al., 2018), BEPS results are mostly better in terms of the Pearson regression coefficient ($R^2$), root mean square error (RMSE), accumulated total sink, and trend against the residual land sink reported by Le Quere et al (2018). The BEPS model was mainly driven by remotely sensed datasets, which can be used for simulating the key carbon (e.g., GPP, NPP and NEP) and water (e.g., ET) fluxes of the terrestrial ecosystems at the yearly, daily and hourly scales. In the BEPS model, there are 8 plant functional types (PFTs), including shrubland, grassland, cropland, and four forest types (the evergreen needleleaf forests (ENF), deciduous needleleaf forests (DNF), deciduous broadleaf forests (DBF), evergreen broadleaf forests (EBF), mixed forests (MXF)).

    At the daily scale, the BEPS model was driven by the daily leaf area index (LAI), daily meteorological data, etc. Daily carbon fixation in the BEPS model is calculated by scaling Farquhar's leaf biochemical model (Farquhar et al., 1980) up to canopy-level implemented with a spatial and temporal scaling scheme (Chen et al., 1999). Daily gross primary productivity (GPP) is calculated separately for sunlit and shaded leaves (see Eq. (1-3) and Eq. (S1-S6)). The photosynthesis of sunlit and shaded leaves $A$ (i.e., $A_{sun}$ (unit: $\mu mol\ m^{-2}\ s^{-1}$) and $A_{shade}$ (unit: $\mu mol\ m^{-2}\ s^{-1}$) ) can be calculated as follows:

$$A = min(A_c, A_j) - 0.015 \times V_m \tag{S1}$$

where $A_c$ denotes the Rubisco-limited gross photosynthesis rate ($\mu mol\ m^{-2}\ s^{-1}$) and is computed as Eq. S2; $A_j$ is the RuBP-limited gross photosynthesis rate ($\mu mol\ m^{-2}\ s^{-1}$) and is calculated as Eq. S3.

$$A_c = V_m \frac{C_i - \Gamma}{C_i + K} \tag{S2}$$

$$A_j = J \frac{C_i - \Gamma}{4.5 C_i + 10.5\Gamma} \tag{S3}$$

where $C_i$ is the intercellular $CO_2$ (Pa); $K$ is a function of enzyme kinetics (Pa) and is calculated as $K = K_C \times \left(1 + \frac{O_2}{K_O}\right)$; $O_2$ is oxygen concentrations in the atmosphere (Pa); $K_C$ and $K_O$ are the Michaelis-Menten constants for $CO_2$ (Pa) and $O_2$ (Pa), respectively; $\Gamma$ denotes the $CO_2$ compensation point without dark respiration (Pa) and is calculated as $\Gamma = 4.04 \times 1.75^{(T_a - 25)/10}$; $V_{cmax}$ is the maximum carboxylation rate ($\mu mol\ m^{-2}\ s^{-1}$) and $J$ represents the electron transport rate ($\mu mol\ m^{-2}\ s^{-1}$). The corresponding formulas for $V_m$ and $J$ are as follows:

$$V_m = V_{cmax25} \times 2.4^{\frac{T_a - 25}{10}} f(T_a) f(N) \tag{S4}$$

$$f(T_a) = \left\{1 + exp\left[\frac{-220000 + 710 \times (T_a + 273)}{8.314 \times (T_a + 273)}\right]\right\}^{-1} \tag{S5}$$

$$J = (29.1 + 1.64V_m) \times PPFD / (PPFD + 2.1 \times (29.1 + 1.64V_m)) \qquad \text{(S6)}$$

where $V_{cmax25}$ is the maximum carboxylation rate at 25°C ($\mu molm^{-2}s^{-1}$); Ta is air temperature (°C); $f(N)$ is the function of nitrogen (N) and is usually set to 0.5 in BEPS model (Liu et al., 1999; Zhang et al., 2018), which can adjust the photosynthesis rate for foliage nitrogen (Bonan, 1995). The $PPFD$ is the photosynthesis photon flux density ($\mu mol\ m^{-2}\ s^{-1}$).

When BEPS modelled the dynamics of carbon pools beyond the GPP, it stratified soil carbon stocks into 9 pools (i.e., surface structural litter, surface metabolic litter, soil structural litter, soil metabolic litter, coarse woody litter, surface microbe, soil microbe, slow, and passive carbon pools). These 9 carbon pools were used to calculate heterotrophic respiration ($R_h$) and autotrophic respiration ($R_a$). Eventually, the net ecosystem productivity (NEP) is calculated as the difference between GPP and $R_h$ and $R_a$.

$$NEP = GPP - R_h - R_a \qquad \text{(S7)}$$

References:

Liu, J., et al., 1997. A process-based boreal ecosystem productivity simulator using remote sensing inputs. Remote Sensing Environment, 62, 158-175.

Le Quere, C., 2018. Global carbon budget 2017. Earth System Science Data, 10, 405-448.

Farquhar, G.D., et al., 1980. A biochemical-model of photosynthetic $CO_2$ assimilation in leaves of C-3 Species. Planta, 149, 78-90.

Chen, J.M., et al., 1999. Daily canopy photosynthesis model through temporal and spatial scaling for remote sensing applications. Ecological Modelling, 124, 99-119.

Bonan, G.B., 1995. Land-atmosphere $CO_2$ exchange simulated by a land surface process model coupled to an atmospheric general circulation model. Journal of Geophysical Research, 100(D2): 2817-2831.

l. 148: flux partitioning is not quality control.

**Response:** Thanks. We have removed this inappropriate description from the text.

l. 151: ER not introduced

**Response:** The full name of ER has been added in the text (see list of abbreviations), namely ecosystem respiration (ER).

l. 170: I am not an expert on this. Any reason why GOSIF was not used? I thought this would be the state-of-the-art GPP product.

**Response:** Yes, the Sun-induced chlorophyll *a* fluorescence (SIF) retrieved from satellites has shown potential as a remote sensing proxy for gross primary productivity

(GPP), such as GOSIF GPP. Generally, there are two approaches to estimating GPP based on SIF: one is to establish a direct empirical linear model of the two, and the other is based on the models, such as Soil-Canopy-Observation of Photosynthesis and the Energy Balance (SCOPE) model. The GOSIF GPP was not used in this study, mainly considering the following reasons:

(1) Most previous studies have shown that SIF and GPP can be characterized by linear relationships (Smith et al., 2018; Li et al., 2018; Li et al., 2019). However, some studies recently indicated a non-linear relationship between SIF and GPP (Kim et al., 2021; Liu et al., 2022), and the relationship between SIF-GPP varies across different climatic zones and biomes (Chen et al., 2021). All these results suggested that the relationship between SIF and GPP remains highly uncertain across space and time. This is mainly due to an insufficient understanding of the influencing mechanism of the relationship between SIF-GPP at present. For example, the GPP-SIF relationship is strongly influenced by environmental factors and has a high sensitivity to precipitation. Especially, there will be differences in the trend of changes in SIF and GPP under drought stress conditions, and SIF offers limited potential for quantitatively monitoring GPP during heat waves (Wohlfahrt et al., 2018). However, most of the SIF-based GPP products including the GOSIF GPP were generated by the linear relationships between SIF and GPP to map GPP globally. Therefore, the current GPP products retrieved from SIF may have significant uncertainty and controversy due to insufficient understanding of the mechanism of the relationship between SIF-GPP (Chen et al., 2021; Liao et al., 2023).

(2) The SIF signal of vegetation is very weak, and it is only 1% of the incident radiation. However, current satellites for SIF detection typically have coarse spatial resolution, which could result in a large systematic bias in both the available SIF and SIF-based GPP products, particularly when the resolution is coarse (Frankenberg et al., 2014). Indeed, Li et al., (2019) have produced relatively high-resolution GOSIF GPP products on a global scale. The raw data used for the GOSIF GPP production stems from SIF observed by the Orbiting Carbon Observatory-2 (OCO-2). The sparse coverage and coarse spatial resolution (~1°) of OCO-2 may also lead to large uncertainty in the production of GOSIF GPP. Additionally, the GOSIF is not fully independent from MODIS greenness indices, since its derivation relies on both solar-induced fluorescence measurements from OCO-2 and MODIS reflectance measurements.

Actually, we recognize that SIF brings major advancements in measuring terrestrial photosynthesis, especially in estimating GPP. We will consider SIF-based GPP in our future research.

References:

Smith, W.K., et al., 2018. Chlorophyll Fluorescence Better Captures Seasonal and Interannual Gross Primary Productivity Dynamics Across Dryland Ecosystems of Southwestern North America. Geophysical Research Letters, 45, 748–757.

Li, X., et al., 2018. Solar-induced chlorophyll fluorescence is strongly correlated with terrestrial photosynthesis for a wide variety of biomes: First global analysis based on OCO-2 and flux tower observations. Global Change Biology, 24, 3990–4008.

Li, X., Xiao, J., 2019. Mapping photosynthesis solely from solar-induced chlorophyll fluorescence: A global, fine-resolution dataset of gross primary production derived from OCO-2. Remote Sensing, 11(21), 2563.

Kim et al., 2021. Solar-induced chlorophyll fluorescence is non-linearly related to canopy photosynthesis in a temperate evergreen needleleaf forest during the fall transition. Remote Sensing of Environment, 258, 112362.

Liu et al., 2022. Non-linearity between gross primary productivity and far-red solar-induced chlorophyll fluorescence emitted from canopies of major biomes. Remote Sensing of Environment, 271, 112896.

Chen et al., 2021. Moisture availability mediates the relationship between terrestrial gross primary production and solar-induced chlorophyll fluorescence: Insights from global-scale variations. Global Change Biology, 27:1144–1156.

Miao, G., et al, 2018. Sun-induced chlorophyll fluorescence, photosynthesis, and light use efficiency of a soybean field from seasonally continuous measurements. Journal of Geophysical Research, 123, 610-623.

Wohlfahrt, G., et al., 2018. Sun-induced fluorescence and gross primary productivity during a heat wave. Scientific Reports, 8,14169.

Liao, Z., et al., 2023. A critical review of methods, principles and progress for estimating the gross primary productivity of terrestrial ecosystems. Frontiers in Environmental Science,11, 1093095.

Frankenberg, C., et al., 2014. Prospects for chlorophyll fluorescence remote sensing from the Orbiting Carbon Observatory-2. Remote Sensing of Environment, 147, 1–12.

l. 210: this reads strange. In S1 the land cover is fixed. But then you write that "in this scenario, LCC may lead to changes..."

**Response:** Thanks for catching the inappropriate description. We have removed the unnecessary and confusing sentences from the revised text.

l. 212: this is confusing. You talk about the conversion of forest to non-forest, and then about forest cover change. Is that not the same thing?

**Response:** Thanks again for pointing out the inappropriate description. To avoid confusion, we have also removed the statement from the revised text where there are unnecessary.

Improve Table S3, explain more. What is remote sensing, what is modeled, etc.

**Response:** Following your suggestion, we have modified the Table S3 as follows:

**Table S3** Details of the published GPP products were used for model comparison.

| Dataset | Time Range | Spatial Resolution | Description | Source | References |
|---|---|---|---|---|---|
| MODIS GPP | 2000-2022 | 500 m | MODIS GPP product derived from satellite observations | https://ladsweb.modaps.eosdis.nasa.gov/archive/allData/6/MOD17A2H/ | Running et al. (2015) |
| EC-LUE GPP | 1982–2018 | 0.05° | EC-LUE GPP product derived from the light use efficiency model | https://doi.org/10.6084/m9.figshare.8942336.v3. | Zheng et al. (2020) |
| NIRv GPP | 1982–2018 | 0.05° | NIRv GPP product derived from satellite observations | https://doi.org/10.6084/m9.figshare.12981977.v2. | Wang et al. (2021) |
| VPM GPP | 2000-2016 | 0.05° | VPM GPP product derived from satellite observations and NCEP Reanalysis II climate data | https://figshare.com/articles/dataset/Annual_GPP_at_0_5_degree/5048005 | Zhang et al. (2017) |
| BEPS$_g$ GPP | 1982–2019 | 0.072727° | BEPS$_g$ GPP product derived from the process-based model | http://www.nesdc.org.cn/sdo/detail?id=612f42ee7e28172cbed3d809 | Chen et al. (2019); He et al. (2021) |

**Results:**

The model performance section is very good. But only GPP is evaluated. What about other model outputs?

**Response:** Thank you very much for this positive comment. In this study, we aim to understand how different drivers affect GPP changes. Therefore, we mainly focus on the validation and evaluation of GPP. In order to further validate the simulation results from the BEPS model, we also validated the simulated NEP at the three flux sites. The validation results of NEP have listed in the supplementary (please see Table S5 and Figure S4-S6).

Also, Fig 3 does not really convince me. Can you discuss why the GPPs are so different?

**Response:** Thank you for this comment. We agree that there are relative differences between these GPP products. This stems mainly from the fact that different products are generated by different methods, data sources, etc, which may lead to differences in the GPPs produced. For example, the MODIS GPP product was mainly generated by the Terra/Aqua satellite observations. The newly released NIRv GPP was produced by near-infrared reflectance (i.e., the AVHRR reflectance from LTDR (Land Long Term Data Record v4) product). Thus, the data sources derived from divergent satellite observations may result in the differences between the two GPPs. Additionally, the EC-LUE GPP, VPM GPP, and the published BEPS GPP are all model outputs, where

EC-LUE GPP and VPM GPP are simulated based on different light use efficiency (LUE) models, respectively, and the BEPS GPP is produced based on a process model. Current LUE-based models do not completely integrate other key environmental regulations to vegetation productivity, such as the effect of atmospheric $CO_2$ concentration. Thus, the underestimation in other GPP products is possibly due to the failure to assess the $CO_2$ fertilizer effects, because almost no apparent response to the rising atmospheric $CO_2$ concentration in the LUE models leads to an underestimated trend (Anav et al., 2015). In our study, the GPP was estimated by a process-based model (i.e., BEPS) that considered the effects of these important factors on GPP, especially the $CO_2$ fertilization effect, which may lead to a higher GPP when compared to other GPP products. Overall, the parameters, inputs, and model structure of different models are inconsistent, which may also lead to differences in GPP production.

Although these products have differences and were used for comparison in this study, we mainly consider that these GPP products have been widely used in previous studies (NIRv GPP: Zhang et al., 2022; MODIS GPP: Yao et al., 2020; VPM GPP: Zhang et al., 2016; BEPS GPP: Chen et al., 2019; EC-LUE GPP: Wang et al., 2020). Especially, Xing et al., (2023) also adopted the same global GPP products for comparison with the GPP simulated by BEPS over China. Moreover, these products are produced from different data sources and methods, and it would be more reasonable and reliable to use them for comparing the simulated GPP in our study.

To respond to your question, we added these discussion in the revised manuscript (see Pages 11, Lines 347-361). We also moved Figure 3 to the supplementary, mainly because Figure 3 is relatively less important for the understanding of the main text, and on the other hand, it also can reduce the number of figures in the main text.


"Figure 2 Comparison of simulated GPP with measured GPP from three flux tower stations at daily (a-c) and annual (d-f) scales. The green lines and dark circles represent the simulated GPP and observed GPP, respectively."

l. 266: This is an issue: obviously the increase in GPP is similar in a study with the same model. The next data product has a much lower increase, 0.017, compared to this study's 0.026.

**Response:** Thank you very much for pointing out the inappropriate description. To avoid confusion, we have removed this sentence from the revised text.

BEPS simulates a higher GPP compared to all the other products, and a higher trend, too. This needs to be discussed further.

**Response:** Agree. Our simulated GPP is slightly higher than other products. Firstly, there are some uncertainties and substantial differences in the simulated interannual variability in GPP from various ecosystem models due to many differences in model structure, parameterization and driving data (Cai et al., 2014; Lin et al., 2023). Secondly, the other GPP products used in this study were mainly generated by the LUE model-based and remote sensing-based models. However, previous studies (Zhu et al., 2018; O'Sullivan et al., 2020; Wang et al., 2023) reported that LUE-based models, remote sensing-based models, machine-learning-based models, etc., may underestimate the GPP at an annual scale. For example, the GPP estimates by the LUE models mainly depend on a few important factors, including solar radiation, air temperature, water availability, and vegetation indexes (e.g., EVI or NDVI). Current LUE-based models do not completely integrate other key environmental regulations to vegetation productivity, such as the effect of atmospheric $CO_2$ concentration on GPP.

Therefore, one cause of the underestimation in other GPP products is possibly failure to assess the $CO_2$ fertilizer effects, because almost no apparent response to the rising atmospheric $CO_2$ concentration in the LUE models leads to an underestimated trend (Anav et al., 2015). In our study, the GPP was estimated by a process-based model (i.e., BEPS) that considers the effects of these important factors on GPP, especially the $CO_2$ fertilization effect, which may lead to a higher GPP compared to other GPP products.

For what it's worth, the results of our comparisons showed that the interannual trends of our simulated results were in line with other GPP products (Fig. S9). Despite possible overestimation, the purpose of this study mainly focuses on the trends and explains the driving mechanism behind them, thus it may not affect our results and conclusions. The above discussion has been added to the revised version (see Page 11, Lines 344-361).


l. 355: You write:

"results showed that most GPP increases in China's subtropical forests due to the increase of LAI, which also offset the negative effects of VSC on GPP, thus allowing VSC to play a key driving factor in promoting GPP increases throughout the forest area."

This is confusing. Did you mean FSC maybe instead of VSC at the first mention? LAI is the same as VSC, right? So how does the effect of change in LAI on GPP offset the effect of change in LAI on GPP? They are the same thing? Or do you mean, there is more positive change that heavily offsets the negative changes?

**Response:** We are sorry for the confusion. FSC in this study represents forest cover changes, while VSC indicates vegetation structure changes. In this study, LAI is the same as VSC. As you suggested above, we have changed VSC to LAI to avoid confusion. Yes, there is the more positive change that heavily offsets the negative changes. Because not all pixels show the same changes in space (see previous Figure 7 and the updated Figure 4f), some pixels have a positive trend in GPP due to the influence of LAI, while others may have a negative trend. Meanwhile, the magnitude of the pixels with positive trends is larger than that of the pixels with negative trends, which results in these pixels cancelling each other out.

For clear understanding, revised text to (see Page 17, Lines 465-467):

"Overall, there are more positive changes in GPP due to the effect of LAI that heavily offsets the negative changes in GPP, ultimately making LAI the main factor in GPP increases throughout China's subtropical forests."

l. 361: verb is missing

**Response:** As suggested, we have revised this sentence as follows (see Page 19, Line 483):

"The annual mean $CO_2$ concentration increased from 371.3 ppm to 408.7 ppm during 2001-2018."

Fig. 9: This is a nice figure that shows the main results.

**Response:** Thank you very much for this encouraging comment.

Fig 9: I am puzzled that, e.g., in b) CC-ALL is nowhere near the sum of the three. I understand that there will be interactions, but I find it quite strange that the interactions are quite strongly positive but each of the components is almost 0. Maybe these cancel each other out over the entire region.

**Response:** We are sorry for the confusion. We agree with your comment that the impact of each climatic factor is almost 0, while the interactions seem to be strong. However, the overall effects of different factors may not be able to be simply added together. Although their individual effects may be small, their interactive effects may become relatively large, because their interactions are not simply linear relationships in our model. This is where we differ from previous studies (Zhang et al., 2014; Zhang et al., 2022) that used relatively simple models to detect single or interactive effects of different factors. In this study, the effects of these factors are analysed using a process-based model in which the different factors are nonlinearly related to each other. Actually, this is also the purpose of this study, which is to try to unravel the possible effects of different drivers on GPP changes individually and interactively based on a process model.


"The magnitude of GPP increase in the small areas is significantly higher than in other regions because temperature, precipitation and radiation all contribute to GPP increase in these areas (Fig. S12). Although the area of GPP reduction due to climate change is relatively large, the magnitude of its impact is relatively small, resulting in smaller areas with higher magnitude offsetting the larger area of GPP decrease."

l. 461: why is that?

**Response:** Thanks for the comment. As shown in Fig. 1, ENF is mainly distributed in the eastern and western regions of the subtropics. Our results also showed that climatic factors (e.g., temperature, precipitation, and solar radiation) in these regions have negative effects on the GPP of ENF (Fig. 6), and particularly the solar radiation declined significantly in the eastern region, which led to a decrease in the GPP of ENF

in the east. For EBF, it is mainly distributed in the central and western regions where climate change mainly contributes to the increase of GPP of EBF.

For clear understanding, revised text to (Page 25, Lines 606-614):

"climate change has a positive effect on the GPP of EBF, but a negative effect on the GPP of ENF. The main reason is that ENF is predominantly located in the eastern and western parts of the subtropics (Fig. 1). In these areas, individual climatic factors (e.g., temperature, precipitation, and solar radiation) or their interactions caused the GPP of ENF to decrease (Fig. 4c-4d), and particularly the solar radiation declined significantly in the eastern region, which led to a decrease in the GPP of ENF in the east. The EBF is mainly distributed in the central and western regions (Fig. 1) where climate change mainly contributes to the increase of EBF GPP (Fig. 4c-4d)."

l. 488: Forest protection has greater carbon uptake potential than what? This also relates to my comment on l. 416. Also, you only refer to GPP. Can you make any claims on NPP?

**Response:** Thanks again for your comment and suggestion. We are sorry for the confusion. Considering this sentence is not necessary, we have removed the confusing sentence from the revised text. As suggested, we also added some claims on NPP to the revised text as follows (Page 26, Lines 642-648).

"Consistent with our study period (2001–2018), Chen et al. (2021b) also reported an increase in vegetation carbon sequestration in China based on the two indicators of GPP and NPP, especially with an accelerated increase in carbon sequestration potential after 2010. They showed that GPP and NPP in China increased obviously at the rate of 49.1–53.1 TgC/yr$^2$ and 22.4–24.9 TgC/yr$^2$, respectively. The significant increase of subtropical forest GPP and NPP was highly attributed to human activities (e.g., ecological restoration projects) in southern and eastern China, especially the human-induced NPP gains can offset the climate-induced NPP losses in southern China."


**Reviewer #2**

Review of "Elevated atmospheric $CO_2$ and vegetation structural changes contributed to GPP increase more than climate and forest cover changes in subtropical forests of China" by Chen et al.

The manuscript by Chen et al. investigates drivers of subtropical forest GPP trends in China using a process-based model that runs to provide causal attribution. The study concludes that the primary drivers of GPP change are the $CO_2$ fertilization effect and increased LAI. While the study conducts comprehensive model experiments and maintains a well-organized structure, it lacks a convincing theoretical framework for designing the experiments and conducting the analysis, which is essential for consideration in publication. Additionally, the manuscript requires careful revision for the English language and logical syntax. Please refer to my comments for further details.

**Response:** Thank you very much for your valuable and thoughtful comments and suggestions. Below we go through point-by-point our answers to the comments, and our responses are in blue. Based on your suggestion, in the introduction, we added relevant theories and statements explaining why we designed the experiments and conducted our analysis. We also streamlined the results, figures and text as suggested. Moreover, we have carefully checked and improved the English writing in the revised manuscript.

General comments:

1. Introduction: In the second paragraph, several relevant drivers are listed, followed by the research question "the relative contributions of these factors…not clear" in the next paragraph. It does not adequately explain to the reader why these factors are crucial to GPP or provide mechanistic expectations. For instance, in Line 60, rather than stating the increased temperature "has also influenced the forest carbon uptake", it would be beneficial to summarize the specific mechanisms and reasons behind this influence. Is the influence positive or negative? Some clarifications would be helpful.

**Response:** Thank you very much for the valuable suggestions. To make the possible mechanisms behind the GPP changes clearer, we have added the following sentences to the revised text (see Pages 2-3, Lines 71-114).

[revised manuscript text omitted]

2. Experiment design: I have two main concerns concerning the experiment design in Table 1. A) When assessing the effect of climate variables on GPP, one of the climate variables (e.g., precipitation) is fixed as the value in 2001 in the forcing for the S2 scenario. As I understand it, that means in the S2 scenario there is no climatological cycle at all. The difference in GPP between S2 and the control run should include the effect of both the long-term trend and short-term variabilities of climate. This means, by design, the trend of GPP driven by climate is overshadowed by the shorter-term variabilities (Figure 6). However, when designing the $CO_2$ and LAI scenarios, the difference of $CO_2$ or LAI forcings are less variable (Figure S10, S11), thus a "clear" trend of GPP can be observed in both Figure 7 (a) and 8 (a). There is no surprise when the authors find that $CO_2$ and LAI are the most prominent drivers, when they are comparing the effect of "trend" (e.g., $CO_2$) and "trend + variabilities" (e.g., precipitation). One may need to test to which extent the way of prescribing climate forcings influences the conclusion, e.g., by removing the trend of climate variables but keeping variabilities. B) Is the GLASS LAI also sensitive to climate change and increasing $CO_2$? With an increased carbon uptake due to increasing $CO_2$, more carbon can be allocated to leaf growth. I wonder if the authors have some thoughts about the causal link when discussing the effect of LAI on GPP.

**Response:** Thank you very much for the comment and suggestion. As suggested, we used the mean value of each climate variable, including the precipitation,

temperature, and solar radiation rather than the initial (2001) value for the different variables to redo the simulation. Here, taking the precipitation as an example, we compared the simulated results based on the mean value of the precipitation over the study periods (see below right figure) with the simulations in the first year (2001) (see below left figure), and found that there are no significant differences between them. Although there are relative differences in the magnitude of the slopes (i.e., compared the present simulations with previous simulations) under the effect of precipitation changes on GPP, they are not significant and show a similar effect, suggesting that the effect of precipitation on GPP changes in different forests is less influenced by trends. As shown in Fig. S11, our results also indicated that the annual variations of the climate variables have insignificant trends from 2001 to 2018. From Fig. S11 and Fig. S12, the results also suggested that the year 2001 was not an extreme year for any of those variables. Therefore, the initial year (2001) used in this study may be reasonable. The same experiment designs were also adopted in previous studies (Chen et al., 2021a; Sun et al., 2022). Overall, considering that there are no obvious trends in these climate variables and the low effects of these variables when compared to the $CO_2$ and LAI, whatever the experimental design it wouldn't change our findings.

[Figure]

For question B, we acknowledge that LAI may be affected by climatic factors and $CO_2$ fertilization. We added the following discussion to the revised manuscript as suggested.

Revised text to (see Page 27, Lines 697-710):

"It should be noted that changes in LAI could be influenced by both climatic factors and elevated atmospheric $CO_2$ concentration (Chen et al., 2019; Chen et al., 2021a; Sun et al., 2022). Previous studies reported that the elevated atmospheric $CO_2$ concentration was the dominant driver of global LAI increase, and there are also regional differences in the impact mechanism of climate factors on LAI changes (Zhu et al., 2016; Zhu et al., 2017), thereby influencing the GPP dynamics. Moreover, the interactions between these driving factors can also influence the LAI, and even the interactive impacts of these factors on LAI may offset each other. For instance, rising $CO_2$ concentration and solar radiation can affect temperature and VPD (Chen et al., 2021a). High VPD leads to plants to close their stomata, resulting in lower intercellular $CO_2$ concentrations in the leaves, which reduces the rate of photosynthesis (Yuan et al., 2019). Additionally, changes in LAI can feed back to the climate through biogeochemical and biogeophysical processes (Li et al., 2023).

There is a bidirectional interaction between vegetation and the atmosphere, and the relationship between vegetation dynamics and driving factors is complicated. The current methods used in this study cannot elucidate the complex interactions of the climate factors and elevated $CO_2$ concentration on LAI changes, which may bring some uncertainties to our results."


    **Response:** Thank you very much for the comment. As suggested, we have removed the "reasonably" from the revised text. Yes, we also used the NEP for testing the model performance, because NEP (i.e., -NEE) is a direct measurement of carbon fluxes between the atmosphere and ecosystems, while the ecosystem GPP cannot be measured directly and is derived from the partitioning of NEE from flux measurements. Therefore, we not only used the observed GPP from the flux sites to validate our model, but also the NEP. We recognized that the validation of model performance based on measured NEP was relatively lower than that of GPP. One reason for this is that the simulation of NEP in the model is affected not only by the accuracy of simulated GPP, but also by the accuracy of simulated heterotrophic respiration (R$_h$) and autotrophic respiration (R$_a$). Therefore, a relatively poor performance of the simulated NEP was observed in this study.

    However, the purpose of this is to disentangle how different drivers affect GPP changes in China's subtropical forests. Therefore, we mainly focus on the GPP in our study area. Our findings also showed that the validation of the simulated GPP at three flux sites performed well. We have added the explanations to the revised manuscript. Revised text to (see Pages 10, Lines 317-323):

    "In this study, we used the NEP for testing the model performance, because NEP (i.e., -NEE (net ecosystem exchange)) is a direct measurement of carbon fluxes between the atmosphere and ecosystems. Therefore, we not only used the

observed GPP from the flux sites to validate our model, but also the NEP. The validation of model performance based on measured NEP was relatively lower than that of GPP. One cause is that the simulation of NEP in the model is influenced not only by the accuracy of simulated GPP, but also by the accuracy of simulated heterotrophic respiration ($R_h$) and autotrophic respiration ($R_a$)."

19. What do the green lines and circles represent in Figure 2?

**Response:** Thanks. The green lines represent the simulated GPP, and the dark circles represent the observations. We added the description of the green lines and dark circles in the Figure caption (see below and Page 10, Lines 326-327).

"Figure 2 Comparison of simulated GPP with measured GPP from three flux tower stations at daily (a-c) and annual (d-f) scales. The green lines and dark circles represent the simulated GPP and observed GPP, respectively."

20. L254-255: It is not clear how the spatial correlation is calculated.

**Response:** Thanks. Here the spatial correlation is calculated pixel by pixel at the annual scale. For example, we obtained the MODIS GPP from a certain pixel, and our simulated GPP was also derived from the same pixel during the same period. Then, the correlation between the two GPPs was computed. Similarly, we can calculate the correlation coefficients of different pixels to obtain their spatial distribution. We added the following description of the methodology for calculating spatial correlation in the revised manuscript. Revised text to (see Page 9, Lines 296-300):

"Moreover, the spatial correlation was adopted in this study to compare the spatial consistency of our simulated GPP with other GPP products. The spatial correlation was calculated pixel by pixel at the annual scale. First, two GPP time series for a certain pixel were obtained in the same period, and then the correlation between the two GPPs was calculated. By analogy, the spatial distribution of the correlation coefficients can be achieved."

21. L261-264: The number does not align within the range of all five GPP products as mentioned. Additionally, the reference to 'another BEPS' requires clarification. How to interpret the difference between "another BEPS" and "this BEPS" in Figure S7d?

**Response:** Thanks. We acknowledge that our simulated GPP is slightly higher than other products. Although our estimated GPP is slightly higher for the entire subtropical forests, our modeled GPP is very close to other GPP products for a specific forest type, such as the DBF and MXF (Fig. S9). In fact, other GPP products (e.g., MODIS GPP, EC-LUE GPP, NIRv GPP, and VPM GPP) also have significant differences when compared to each other (Fig. S9). The results indicate there are still significant differences in simulating GPP to date. The possible reasons are:

(1) there are some substantial differences in the simulated GPP from various ecosystem models due to many differences in model structure, parameterization, and driving data (Cai et al., 2014; Lin et al., 2023).

(2) other GPP products used in this study were mainly generated by the LUE model-based and remote sensing-based models. However, previous studies (Zhu et al., 2018; O'Sullivan et al., 2020; Wang et al., 2023) also reported that LUE-based models, remote sensing-based models, and machine-learning-based models may underestimate the GPP at an annual scale. For example, the GPP estimates by the LUE models mainly depend on a few important factors, such as solar radiation, air temperature, water availability, and vegetation indexes (e.g., EVI or NDVI). Current LUE-based models do not completely integrate other key environmental regulations to vegetation productivity, such as the effect of atmospheric $CO_2$ concentration. Thus, the underestimation in other GPP products is possibly due to the failure to assess the $CO_2$ fertilizer effects, because almost no apparent response to the rising atmospheric $CO_2$ concentration in the LUE models leads to an underestimated trend (Anav et al., 2015). In our study, the GPP was estimated by a process-based model (i.e., BEPS) that considers the effects of these important factors on GPP, especially the $CO_2$ fertilization effect, which may lead to a higher GPP compared to all the other products.

For what it's worth, the results of our comparisons showed that the interannual trends of our simulated results were in line with other GPP products (Fig. S9). Despite possible overestimation, the purpose of this study mainly focuses on the trends and explains the driving mechanism behind them, thus it may not affect our results and conclusions.

We added the discussion to the revised version (see Page 11, Lines 344-361). We also reworded the statement "…, which falled in the range of the five GPP products… " to "…, closing to the magnitudes of the three GPP products… ".

In order to distinguish it from the GPP we simulated, the reference (i.e., $BEPS_g$ GPP) to 'another BEPS' has been added to the revised text (see Page 7, Line 239) and Table S3. Actually, the $BEPS_g$ GPP product was also produced by a similar BEPS model. However, this model is driven by the global datasets, and the parameters in the model are also calibrated for the global GPP mapping. Therefore, It is different from our simulated GPP and can be used for comparison with our simulated GPP.

   **Response:** Thanks again! The "increase" has been changed to the revised manuscript.

---

## Referee Report (RR1)

The authors have improved the manuscript quite a lot. The discussion now contains much more explanations to bring the results into context.
I still have some more remarks, and would say that "minor revisions" are still necessary. Furthermore, I would recommend to have a native speaker of English read over the paper with the authors. Clearer language could improve the paper even lot more I think.

Concrete remarks (line numbers refer to the revised version):

**Methods:**

l. 134 (new version of manuscript): Ok, but a range of mean temp values would be nice, just like for precipitation.

Description of other models: Can still be made clearer. You discuss now more the reasons for the discrepancies, but those should be easy to check from the table, for instance, in the table it only says that about VPM: "from satellite observations and NCEP Reanalysis II climate data" but you could mention that this is based on LUE.

Model explanation: Ok, thank you for adding more details in the supplements. Please also provide a reference for the performance in comparison to GCP. Also mention that it is a process-based model. In your response you also mention Xing et al. (2023), maybe you can also cite them in your S1 Text because they have a nice figure depicting BEPS. For a reader who does not know BEPS, having a graphic like their Fig 1 makes it so much easier to understand the scope of the model quickly and be able to interpret your study.

**Results:**

Model validation: Ok, yes, you also showed NEP. Even if you're only interested in GPP, I find it important to also check the performance of other variables, too, to make sure the model isn't right for the wrong reasons. But I think I am just used to models with more outputs, I would have wanted to see graphs for biomass and so on, but it seems that the model does not put these out. So I think it is ok what you did. I just believe that as a modeling community, we have to really pay attention to model validation.

Again regarding the bias in the low values in DHS and the generally lower performance of the model at DHS: You explained it nicely in your response but didn't change anything in the manuscript it seems. This is important information to convey to the reader and to strengthen the confidence in the model. For instance, you mentioned in your response:
*For example, as reported by Wang et al., (2006), the low observed values of CO2 flux are mainly caused by a CO2 leak during the nighttime at the DHS station. In addition, the effect of topography also led to generally low fluxes in the southerly direction at DHS site (Li et al., 2021).*
But this information did not make it in the manuscript. Just a short remark in the caption of Fig 2 or in the main text would be helpful.

Performance of GPP: Thanks, you provide me with some answers that make sense. Especially that the LUE products will have lower GPP due to missing CO2 fertilization makes a lot of sense.
I think you should condense the new text. For instance, of course two different satellite products will lead to different GPP estimates. Also regarding the lines 321-329. It suffices that you mention that a likely reason for the higher estimation compared to VPM and EC-LUE is the missing CO2-fertilization in the light use efficiency based models. No need for 8 lines.

I would however be interested: NIRv is sometimes really much lower than the rest. Can you find a possible explanation for this? It can't be missing CO2-fertilization, since it is satellites. For ENF for instance, the NIRv value is half that of BEPS. I agree that you are interested in trends, but still, it is really important to thoroughly address model/data differences.

l. 332: Should be "The *simulated* forest GPP". Please also mention that you used the S_baseline for this.

l. 366-368: unclear.

l. 377: But Fig 5 shows that CO2 is the main factor?

l. 413 grammar, verb is missing?

**Discussion:**

l. 427: "have the highest carbon sequestration rate under the background of global change". Needs to be clarified. I don't know what you mean here.

l. 439: Again the point about cropland being potentially more productive. The added phrase does not really back your claim. What was the GPP per area in that cropland before? What is it now? The "0.16TgC" increase does not really help me. I want to understand what the GPP was in that area, so I can interpret that value, whether this cropland was simply low in production before it was converted.

l. 446: interesting, so here a change from ENF to MXF and cropland leads to a decrease in GPP. But again, I want to know why that could be? Why are MXF less productive than ENF in those regions where the FCC happened?

Section 4.1.2: This section is very nice now, discussing why the different CC effects can have positive and negative effects and relating it to your findings. Well done.

l. 502: LAI the dominant contributor? Second-dominant, no?

l. 515: I appreciate the connection of GPP, NPP and carbon uptake. This is important to understand the implications of your study in terms of carbon uptake.

Section 4.1.4: This is very introduction-y, and not what I meant in my previous review. I mean, yes, CO2 fertilization enhances GPP. But in your first version you linked that to C sequestration. That's ok but then you need to discuss also what happens to respiration in the meantime, what happens to tree mortality, and tree longevity. There are numerous uncertainties between CO2 fertilization effect and the carbon sink. That's what was missing. Not re-iterating the relevance of GPP.

I apologize if my review comment was not clear here.
Also, I would say that the statement "The carbon sequestered by vegetation through photosynthesis in a given unit of space and time, i.e., GPP" is not correct, because it ignores respiration.

Finally, I think it makes much more sense to measure everything per m2 as you do now. The only problem: The total impacts are now not in the paper anymore. I think you should conclude the discussion with a short section on the total impact in Tg/year, and discuss the briefly discuss the total changes in areas, LAI and so forth.

---

## Author Response (AR2)

**Response to Referee1's Comments**

Dear Editor and Reviewer,

Thank you and the reviewer for the additional feedback on our manuscript. The reviewer lists some good points for clarification, and we have tried to address them in our revised revision. Reviewer comments are presented in black font; our responses are in blue font. Thank you again for your consideration. Please see below our replies, which hopefully will address the reviewer's comments in a satisfactory manner.

Sincerely,

Tao Chen, Félicien Meunier, Marc Peaucelle, Guoping Tang, Ye Yuan, Hans Verbeeck

The authors have improved the manuscript quite a lot. The discussion now contains much more explanations to bring the results into context.
I still have some more remarks, and would say that "minor revisions" are still necessary. Furthermore, I would recommend to have a native speaker of English read over the paper with the authors. Clearer language could improve the paper even lot more I think.

**Response:** Thank you very much for providing constructive and valuable criticism. Below we go through point-by-point our answers to the comments. We hope that you will find the result satisfying. Based on your suggestion, we also polished the English throughout the revised manuscript using a language editing service (https://www.papertrue.com/ordering/academic-editing-proofreading-servicess). Please see the certificate below.

[Figure]

PAPER**TRUE**

**Certificate of Editing**

1st March 2024

Title: Elevated atmospheric CO2 concentration and vegetation structural changes contributed to GPP increase more than climate and forest cover changes in subtropical forests of China.

To whoever it may concern,

This is to certify that PaperTrue Editing and Proofreading Services (www.papertrue.com) have edited and proofread the following document for **Tao Chen** and duly delivered the edited document on **1st March 2024**. The document was edited by our native English-speaking editors.

Manuscript.docx (9689 words)

The PaperTrue Team
www.papertrue.com

PAPERTRUE PTE. LTD.

100 Peck Seah Street, #08-14, PS100, 079333, Singapore

Concrete remarks (line numbers refer to the revised version):

**Methods:**

l. 134 (new version of manuscript): Ok, but a range of mean temp values would be nice, just like for precipitation.

**Response**: Thanks for the suggestion. We added a range of mean temp values to the revised text (please see below and Page 4, Lines 138-140).

*"The mean annual temperature was between 10.8 °C and 22.9 °C normally increasing from the northwest toward the southeast."*

Description of other models: Can still be made clearer. You discuss now more the reasons for the discrepancies, but those should be easy to check from the table, for instance, in the table it only says that about VPM: "from satellite observations and NCEP Reanalysis II climate data" but you could mention that this is based on LUE.

**Response**: Thanks. According to your suggestions, we have added more detailed information to Table S3 to describe the published GPP products (see below).

**Table S3** Details of the published GPP products were used for model comparison.

| Dataset | Time Range | Spatial Resolution | Description | Source | References |
|---|---|---|---|---|---|
| MODIS GPP | 2000-2022 | 500 m | MODIS GPP products are generated by the MOD17 algorithm and Biome-Property-Look-Up-Table by integrating the Terra/Aqua satellite observations (i.e., MODIS surface reflectances, MOD09) and meteorological data | https://ladsweb.modaps.eosdis.nasa.gov/archive/allData/6/MOD17A2H/ | Running et al. (2015) |
| EC-LUE GPP | 1982–2018 | 0.05° | EC-LUE GPP data are derived from the Eddy Covariance-Light Use Efficiency model by integrating several major long-term environmental variables (e.g., air temperature, leaf area index, and atmospheric water vapor pressure) | https://doi.org/10.6084/m9.figshare.8942336.v3. | Zheng et al. (2020) |
| NIRv GPP | 1982–2018 | 0.05° | NIRv GPP data are generated by combining the long-term satellite observations of AVHRR reflectance from LTDR (Land Long Term Data Record v4) product and global flux sites with the machine-learning algorithm | https://doi.org/10.6084/m9.figshare.12981977.v2. | Wang et al. (2021) |
| VPM GPP | 2000-2016 | 0.05° | VPM GPP products are based on an improved light use efficiency model and are driven by satellite data from MODIS (e.g., MCD12Q1, MYD11A2 and MOD09A1) and climate data from NCEP Reanalysis II | https://figshare.com/articles/dataset/Annual_GPP_at_0_5_degree/5048005 | Zhang et al. (2017) |
| BEPS$_g$ GPP | 1982–2019 | 0.072727° | BEPS$_g$ GPP products are generated by the process-based Boreal Ecosystem Productivity Simulator model with global calibrated parameters and are driven by remotely sensed LAI, meteorological data (e.g., CRUNCEP V8.0 dataset), soil data, etc. | http://www.nesdc.org.cn/sdo/detail?id=612f42ee7e28172cbed3d809 | Chen et al. (2019); He et al. (2021) |

Model explanation: Ok, thank you for adding more details in the supplements. Please also provide a reference for the performance in comparison to GCP. Also mention that it is a process-based model. In your response you also mention Xing et al. (2023), maybe you can also cite them in your S1 Text because they have a nice figure depicting BEPS. For a reader who does not know BEPS, having a graphic like their Fig 1 makes it so much easier to understand the scope of the model quickly and be able to interpret your study.

**Response**: Thanks for the good suggestion. As suggested, we have provided a reference for the performance in comparison to Global Carbon Project (GCP). The annual net ecosystem productivity (NEP) can be used to characterize terrestrial $CO_2$ sinks (positive values represent a flux from the atmosphere to the land or vice versa). Therefore, we obtained the annual terrestrial sink from the Global Carbon Budget (GCB) 2023 provided by the Global Carbon Project (Friedlingstein et al., 2023) and used it for comparison. The annual terrestrial sink is computed as the sum of emissions from fossil fuel consumption, cement production, and land-use change minus the sum of $CO_2$ accumulated each year in the atmosphere (i.e., the annual global residual terrestrial sink). Considering that GCB only provides annual global terrestrial $CO_2$ sink data, we also re-simulated global NEP based on the BEPS model to make it comparable. Compared to the GCB, the modeled NEP during 2001-2018 showed a good performance in terms of the interannual changing trends (0.07 Pg C/year vs. 0.09 Pg C/year) (Fig. S7a) and Pearson's coefficient ($R^2 = 0.46$, $p < 0.05$) (Fig. S7b). The results further confirmed that the BEPS have a good performance in the simulation of the global or regional carbon fluxes (e.g., GPP, NEP, etc.).

We added the results of the comparison to the supplementary and mentioned it in the main text. We also mentioned that GCP is based on the outputs of the process-based model. Additionally, following your suggestion, we added the reference (i.e., Xing et al., 2023) to Text S1.

[Figure]

**Fig S7.** Comparison of the simulated annual terrestrial sink (NEP) by the BEPS model and the residual terrestrial sink estimated by the Global Carbon Project (a). The insert figure represents the correlation between the simulated annual terrestrial sink (NEP) by BEPS and the annual residual terrestrial sink estimated by the Global Carbon Project (b).

References
Friedlingstein, P., et al., 2023. Global Carbon Budget 2023. Earth System Science Data, 15, 5301–5369.

Xing et al., 2023. Modeling China's terrestrial ecosystem gross primary productivity with BEPS model: Parameter sensitivity analysis and model calibration. Agricultural and Forest Meteorology, 343, 15, 109789.

**Results:**

Model validation: Ok, yes, you also showed NEP. Even if you're only interested in GPP, I find it important to also check the performance of other variables, too, to make sure the model isn't right for the wrong reasons. But I think I am just used to models with more outputs, I would have wanted to see graphs for biomass and so on, but it seems that the model does not put these out. So I think it is ok what you did. I just believe that as a modeling community, we have to really pay attention to model validation.

**Response**: Following your suggestion, we also simulated the net primary productivity (NPP) based on our model, and we obtained 33 measured subtropical forest NPP values from the published literature to validate our simulated NPP (see below Table S6 and Fig. S8). The results show that our model performs well in simulating NPP ($R^2 = 0.62$, $p < 0.001$) (Fig. S8). We also added these results to the Supplementary.

[Figure]

**Fig S8**. Validation of modeled forest NPP using measured forest NPP in the Chinese subtropics

**Table S6** Sites information of the measured net primary productivity (NPP) data used in this study

| ID | Longitude | Latitude | Measured NPP (g C/m²/year) | References |
|----|-----------|----------|----------------------------|------------|
| 1 | 112.53 | 23.17 | 395.95 | Yang et al., 2017 |
| 2 | 101.02 | 24.53 | 976.15 | Tan et al., 2011 |
| 3 | 115.05 | 26.73 | 487.51 | Yang et al., 2017 |
| 4 | 109.75 | 26.83 | 313.40 | Zhang, 2010 |
| 5 | 112.86 | 29.53 | 515.65 | Han, 2008 |
| 6 | 116.99 | 30.47 | 506.10 | Han, 2008 |
| 7 | 113.91 | 33.35 | 343.40 | Geng, 2011 |
| 8 | 109.445 | 28.405 | 640.15 | Fan et al., 2011, |
| 9 | 109.445 | 28.405 | 591.25 | Fang et al., 2003 |
| 10 | 109.445 | 28.405 | 742.39 | Fang et al., 2002 |
| 11 | 110.515 | 27.505 | 484.55 | Lan et al., 2004 |
| 12 | 106.985 | 26.455 | 626.81 | Li et al., 2007 |
| 13 | 106.985 | 26.455 | 471.22 | Li et al., 2008 |
| 14 | 106.985 | 26.455 | 493.45 | Liang et al., 2007 |
| 15 | 106.985 | 26.455 | 626.81 | Liu et al., 2007 |
| 16 | 106.985 | 26.455 | 529.01 | Liu et al., 2007 |
| 17 | 109.675 | 23.755 | 382.31 | Luo et al., 2011 |
| 18 | 109.675 | 23.755 | 426.76 | Luo et al., 2011 |

| 19 | 109.785 | 26.915 | 222.27 | Luo et al., 2011 |
|----|---------|--------|--------|------------------|
| 20 | 108.355 | 22.975 | 448.99 | Qi et al., 2007 |
| 21 | 109.835 | 22.625 | 1138.04 | Qin et al., 2011 |
| 22 | 100.855 | 23.205 | 1200.27 | Xia et al., 2010 |
| 23 | 100.855 | 23.205 | 1066.90 | Xia et al., 2010 |
| 24 | 99.455 | 24.335 | 1089.14 | Xia et al., 2010 |
| 25 | 99.455 | 24.335 | 817.96 | Xiong et al., 2006 |
| 26 | 107.965 | 25.305 | 635.70 | Yang et al., 2008 |
| 27 | 107.955 | 25.305 | 569.02 | Yang et al., 2001 |
| 28 | 111.885 | 23.455 | 764.62 | Yang et al., 2003 |
| 29 | 111.885 | 23.455 | 831.30 | Yang et al., 2003 |
| 30 | 108.355 | 22.975 | 422.32 | Ye et al., 2010 |
| 31 | 108.355 | 22.975 | 711.27 | Ye et al., 2010 |
| 32 | 108.355 | 22.975 | 733.50 | Ye et al., 2010 |
| 33 | 112.535 | 23.175 | 915.76 | Yin et al., 2010 |

References

Yang, J. et al., 2017. Nonlinear Variations of Net Primary Productivity and Its Relationship with Climate and Vegetation Phenology, China. Forests, 8, 361.

Tan ZH, et al., 201. An old-growth subtropical Asian evergreen forest as a large carbon sink. Atmospheric Environment, 45, 1548-1554.

Zhang LP, 2010. Characteristics of $CO_2$ flux in a Chinese Fir Plantations Eeosystem in Huitong County,Hunan Province. Unpublished Master Central South University of Forestry and Technology,Changsha, P.R.China (in Chinese with English abstract), 61 pp.

Han, S., 2008. Productivity estimation of the poplar plantations on the beaches in middle and low reaches of Yangtze river using eddy covariance measurement. Unpublished Master Chinese Academy of Forestry, Beijing, P.R. China (in Chinese with English abstract), 75 pp.

Geng, S., 2011. Study on the carbon flux observation over poplar plantation ecosystem of XiPing city in Henan Province of China. Unpublished Master Beijing Forestry University, Beijing, P.R.China (in Chinese with English abstract), 91 pp.

Fan, J.Y. et al., 2011. Research on the tree biomass and productivity of Cryptomeria fortunei Hooibrenk plantation. Journal of Fujian Forestry Science and Technology, 38, 1–5.

Fang, X. et al., 2003. Productivity and carbon dynamics of Masson Pine plantation. Journal of Central South Forestry University, 23, 11–15.

Fang, Y.T. and Mo, J.M. 2002. Study on carbon distribution and storage of a pine forest ecosystem in Dinghushan Biosphere Reserve. Guihaia, 22, 305–310.

Lan, Z.J. et al., 2004. Biomass distribution of major plant communities in Jiuzhaigou valley, Sichuan. Chinese Journal of Appplied Environmental Biology, 10, 299–306.

Li, W.B., et al., 2007. Biomass compositions of Pinus tabulaeformis plantation and their relationships in the Dagou valley of the upper Minjiang River. Journal of Mountain Science, 25, 236–244.

Li, Z.H. et al., 2008. Effects of stand density upon the biomass and productivity of Eucalyptus urograndis. Journal of Central South Forestry University, 28, 49–54.

Liang, N. et al., 2007. A study on biomass in sapling stage of pure Betula alnoides forest and Betula alnoides and Cinnamomum cassia mixed forest. Journal of West China Forestry Science, 36, 44–49.

Liu, X.C., et al., 2007. Biomass study of the plantation of Alnus cremastogyne Burkill at different stages of age. Journal of Central South Forestry University, 27, 83–86.

Luo, J. et al., 2011. Study on biomass of different types for protection forest system area around Dongting Lake. Hunan Forestry Science and Technology, 38, 27–29.

Qi, L.H. et al., 2007. Species diversity and biomass allocation of vegetation restoration communities on degraded lands. Chinese Journal of Ecology, 26, 1697–1702.

Qin, W.M., et al., 2011. Study on the biomass and growth law of Paramichelia baillonii plantation. Journal of Fujian College of Forestry, 31, 110–114.

Xia, H.B. 2010. Biomass and net primary production in different successional stages of karst vegetation in Maolan, SW China. Guizhou Forestry Science and Technology, 38, 1–7.

Xiong, D.G. et al., 2006. A study of annual growth and biomass of Alnus formosana in Yuanba District of Guangyuan City. J. Sichuan Forestry Science and Technology, 27, 55–58..

Yang, L.L. et al., 2008. Comparison between biomass and productivity of young Alnus cremastogyne Burkill plantation under different site conditions. Journal of Central South University of Forestry & Technology, 28, 122–126.

Yang, Q.P. et al., 2001. Studies on the dynamic succession of Pinus massoniana community in Heishiding Natural Reserve. Guihaia, 21, 295–300.

Ye, S.M.; Wen, Y.G.; Yang, M.; Liang, H.W.; Lan, J.X. Correlation analysis on productivity and plant diversity of Eucalyptus plantations under successive rotation. Acta Bot. Boreal.-Occident. Sin. 2010, 30, 1458–1467.

Yin, G.Q. et al., 2010. Studies on biomass of different young forests converted from farm land in Huitong, Hunan Province. Journal of Central South University of Forestry & Technology, 30, 9–14.

Again regarding the bias in the low values in DHS and the generally lower performance of the model at DHS: You explained it nicely in your response but didn't change anything in the manuscript it seems. This is important information to convey to the reader and to strengthen the confidence in the model. For instance, you mentioned in your response:

*For example, as reported by Wang et al., (2006), the low observed values of CO2 flux are mainly caused by a CO2 leak during the nighttime at the DHS station. In addition, the effect of topography also led to generally low fluxes in the southerly direction at DHS site (Li et al., 2021).*

But this information did not make it in the manuscript. Just a short remark in the caption of Fig 2 or in the main text would be helpful.

**Response**: We appreciate this insightful suggestion. We have added the explanation to the caption of Fig. 2 in the revised version as suggested (see below and Page 10, Line 320-324).

"*There may be relatively low-quality issues with observed flux data from DHS, which may affect our validation results. For example, as reported by Wang et al., (2006), the low observed values of $CO_2$ flux are mainly caused by a $CO_2$ leak during the nighttime at the DHS station. In addition, the effect of topography also led to generally low fluxes in the southerly direction at the DHS site (Li et al., 2021).*"

References:
Wang et al., 2006. $CO_2$ flux evaluation over the evergreen coniferous and broad-leaved mixed forest in Dinghushan, China. Science in China Series D: Earth Sciences, 49, 127–138.

Li et al., 2021. An observation dataset of carbon and water fluxes in a mixed coniferous broad-leaved forest at Dinghushan, Southern China (2003 – 2010). China Scientific Data, 6(1), DOI: 10.11922/csdata. 2020. 0046.zh.

Performance of GPP: Thanks, you provide me with some answers that make sense. Especially that the LUE products will have lower GPP due to missing CO2 fertilization makes a lot of sense.
I think you should condense the new text. For instance, of course two different satellite products will lead to different GPP estimates. Also regarding the lines 321-329. It suffices that you mention that a

likely reason for the higher estimation compared to VPM and EC-LUE is the missing CO2-fertilization in the light use efficiency based models. No need for 8 lines.

**Response**: Thank you for this suggestion. We have condensed the text in this Section as follows (Page 11, Lines 344-349).

*"For example, the MODIS GPP, EC-LUE GPP and VPM GPP were simulated by different light use efficiency (LUE) models. However, most current LUE-based models do not completely integrate some key environmental regulations into vegetation productivity, such as the effect of atmospheric $CO_2$ concentration, which may result in underestimation. In this study, GPP was simulated by a process-based model (i.e., BEPS) that considered the $CO_2$ fertilisation effect, which may lead to a higher GPP compared to other GPP products."*

I would however be interested: NIRv is sometimes really much lower than the rest. Can you find a possible explanation for this? It can't be missing CO2-fertilization, since it is satellites. For ENF for instance, the NIRv value is half that of BEPS. I agree that you are interested in trends, but still, it is really important to thoroughly address model/data differences.

**Response**: Yes, we agree that NIRv GPP can be influenced by $CO_2$ fertilization. Actually, we found the NIRv GPP was lower than other GPP products and our simulated GPP. A previous study also reported that NIRv GPP products underestimated in situ observations (Bai et al., 2023). The possible explanations for this are as follows:

The NIRv GPP data were generated by combining the long-term satellite observations of NIRv data and global flux sites with the machine-learning algorithm. However, the generation of this product is based on the relationship between NIRv and GPP for the 104 flux sites (i.e., FLUXNET data, which can be available at https://fluxnet.fluxdata.org/) selected in the model, and then upscaled these NIRv-GPP relationships from the site level to the global scale. (1) Currently, the distribution of FLUXNET data is uneven, and it is mainly distributed in Europe and North America, as well as in mid- and high-latitude regions, whereas it is very rare in subtropical regions of China, especially with fewer forest flux sites in this region. However, the accuracy of machine learning-based GPP depends mainly on the number of flux sites. Considering the limited tower observations of the Chinese subtropical forest region, this may affect the NIRv GPP estimation and thus there is a high degree of uncertainty about this product in our study area. (2) The NIRv is calculated by NDVI and near-infrared band (Badgley et al., 2017). However, the NDVI would tend to saturate in areas with high vegetation coverage such as the subtropical and tropical regions. Although NIRv can partially eliminate this problem by adding additional information in the near-infrared band, it still has an impact on NIRv GPP estimation due to the impact of NDVI saturation, eventually leading to underestimation of NIRv GPP (Bai et al., 2023). (3) Many previous studies also reported the underestimation of GPP based on machine learning method (Anav et al., 2015; Jung et al., 2020; Zheng et al., 2020). For example, there is an underestimation of the commonly used FLUXCOM GPP product, both in terms of trend and magnitude. This may also happen with the NIRv GPP as it is also produced based on a machine learning algorithm. Besides, Wang et al. (2021) also pointed out that NIRv GPP was usually lower than GPP based on process model simulation (i.e., the TRENDY model), in terms of trend and magnitude. Therefore, our simulated GPP by the process-based BEPS model may be higher than NIRv GPP.

References:

Bai et al., 2023. Different Satellite Products Revealing Variable Trends in Global Gross Primary Production. Journal of Geophysical Research: Biogeosciences, 128, e2022JG006918.

Badgley et al., 2017. Canopy near-infrared reflectance and terrestrial photosynthesis. Science Advances, 3(3), e1602244.

Anav et al., 2015. Spatiotemporal patterns of terrestrial gross primary production: a review. Reviews of Geophysics, 53(3), 785-818.

Jung et al., 2020. Scaling carbon fluxes from eddy covariance sites to globe: synthesis and evaluation of the FLUXCOM approach. Biogeosciences, 17, 1343–1365.

Zheng et al., 2020. Improved estimate of global gross primary production for reproducing its long-term variation, 1982–2017. Earth System Science Data, 12, 2725–2746.

l. 332: Should be "The *simulated* forest GPP". Please also mention that you used the S_baseline for this.

**Response**: Thanks for the suggestion. As suggested, we have modified the sentence as follows (see Page 11, Line 364).

"*Based on the scenario $S_{baseline}$ (Table 1), the simulated forest GPP showed a significant increasing trend (20.67 gC/m$^2$/year, p = 0.000) during 2001-2018 ...*"

l. 366-368: unclear.

**Response**: We are sorry for the confusion. We have removed the unnecessary and confusing sentence from the revised text.

l. 377: But Fig 5 shows that CO2 is the main factor?

**Response**: Thanks for catching the inappropriate description. We have rewritten the sentence as follows (see Page 13, Lines 411-412).

"*...ultimately making LAI the second dominant factor in GPP increases throughout China's subtropical forests.*"

l. 413 grammar, verb is missing?

**Response**: Thanks. Done.

**Discussion:**

l. 427: "have the highest carbon sequestration rate under the background of global change". Needs to be clarified. I don't know what you mean here.

**Response**: We apologize for the inappropriate description. We have scrutinised the sentence and found it confusing and unnecessary. To avoid misunderstandings for readers, we have removed it from the revised text.

l. 439: Again the point about cropland being potentially more productive. The added phrase does not really back your claim. What was the GPP per area in that cropland before? What is it now? The

"0.16TgC" increase does not really help me. I want to understand what the GPP was in that area, so I can interpret that value, whether this cropland was simply low in production before it was converted.

**Response**: Thank you very much for your further attention to this point. Following your suggestion, we have carefully recounted the GPP in areas where cropland was converted to ENF. Before the conversion, the regional average GPP for cropland was 1466.37 g $C/m^2$/year (2001). After the conversion of cropland to ENF, the regional average GPP for this region was 1851.86 g $C/m^2$/year (2018). Therefore, after the conversion of cropland to ENF, the GPP in the converted area increased by 385.49 g $C/m^2$/year. We also recounted the GPP in areas where ENF was converted to cropland. Before the conversion, the regional average GPP for ENF was 2120.51 g $C/m^2$/year (2001). After the conversion of ENF to cropland, the regional average GPP for this region was 1317.99 g $C/m^2$/year (2018), a reduction of 802.82 g $C/m^2$ /year in the converted area.

For the conversion of cropland to EBF, the regional average GPP of cropland in 2001 was 1732.40 g $C/m^2$/year. After cropland converting to EBF, the regional average GPP was 1935.69 g $C/m^2$/year (2018), an increase of 203.29 g $C/m^2$/year. All the results indicated that cropland might produce lower GPP than ENF and EBF during their conversion. The previous study also reported that forests exhibit higher GPP than cropland in southern China (Ye et al., 2021; Li et al., 2022).

We are very grateful to the reviewer for raising such valuable scientific questions, which we have not considered before. Considering that there are still some uncertainties regarding the impact of the conversion between cropland and forests on GPP in China's subtropical regions (Krause et al., 2022), in future research, we will carefully explore them in depth based on your suggestions.

As suggested, the sentences in the previous version:

"*This is due to the increase in the total area of EBF and MXF (Fig. 4a), which are mainly converted from cropland (Table S6). For example, after the conversion of cropland to MXF in the study area, GPP in the converted area increased by 0.16 Tg C between 2001 and 2018.*"

have been updated to (Page 17, Lines 480-486):

"*This is due to the increase in the total area of EBF and MXF (Fig. 4a), which are mainly converted from cropland (Table S7). For example, our statistics showed that, before conversion, the regional average GPP of cropland in 2001 was 1732.40 g $C/m^2$/year, whereas after the cropland was converted to EBF, the regional average GPP was 1935.69 g $C/m^2$/year in 2018, an increase of 203.29 g $C/m^2$/year.*"

References:
Krause et al., 2022. Quantifying the impacts of land cover change on gross primary productivity globally. Scientific Reports, 12, 18398.
Ye et al., 2021. Spatio-temporal variations of land vegetation gross primary production in the Yangtze River Basin and correlation with meteorological factors. Acta Ecologica Sinica, 41(17): 6949-6959.
Li et al., 2022.Temporal Changes in Land Use, Vegetation, and Productivity in Southwest China. Land, 11, 1331.

l. 446: interesting, so here a change from ENF to MXF and cropland leads to a decrease in GPP. But

again, I want to know why that could be? Why are MXF less productive than ENF in those regions where the FCC happened?

**Response**: Thank you. We are very sorry for the misunderstanding caused by our statement due to problems with our non-native English speakers. What we mean here is just that the decrease in ENF area is mainly due to its conversion into MXF and cropland (see Table S7). However, we do not want to express that the conversion of ENF to MXF directly leads to a decrease in GPP. In fact, what we mean is that the decrease in ENF GPP (not GPP) is due to the conversion between ENF and MXF and cropland. Yes, you are right. The conversion of ENF to MXF can cause an increase in GPP in our study. Same as above, we further counted the GPP changes caused by the conversion between ENF and MXF. The regional average GPP for ENF was 1436.65 g C/m$^2$/year in 2001, and the regional average GPP was 1840.49 g C/m$^2$/year (2018) after the conversion of ENF to MXF, with an increment of 403.84 g C/m$^2$/year in the converted area. On the contrary, the regional average GPP of MXF in 2001 was 1695.59 g C/m$^2$/year. After MXF converted to ENF, the regional average GPP was 1577.89 g C/m$^2$/year (2018), a decrease of 117.70 g C/m$^2$/year. The results confirmed that MXF might produce higher GPP than ENF.

However, the decrease in ENF GPP of 268.65 g C/m$^2$/year due to the conversion between ENF and cropland (i.e., ENF = 2120.51 g C/m$^2$/year in 2001; ENF = 1851.86 g C/m$^2$/year in 2018, see above) is greater than the increase in ENF GPP of 141.24 g C/m$^2$/year due to the conversion between ENF and MXF (i.e., ENF = 1436.65 g C/m$^2$/year in 2001; ENF = 1577.89 g C/m$^2$/year in 2018, see above), ultimately resulting in a slight decrease in ENF GPP.

The statements in the last version:

 "*The total area of the ENF was reduced obviously during the study period in eastern and southern regions, and most of the ENF was converted to MXF (19,040 km$^2$) and cropland (13,100 km$^2$) (Table S6), causing large parts of ENF GPP to decrease (Fig. 4a).*"

have been changed to (see Page 17, Lines 491-498):

 "*The total area of the ENF was reduced obviously during the study period in eastern and southern regions, and most of the ENF was converted to MXF (19,040 km$^2$) and cropland (13,100 km$^2$) (Table S7). Here, we further counted the changes in GPP caused by conversion between ENF and MXF and cropland, and found that the decrease in ENF GPP of 268.65 g C/m$^2$/year due to the conversion between ENF and cropland (i.e., ENF = 2120.51 g C/m$^2$/year in 2001; ENF = 1851.86 g C/m$^2$/year in 2018) was greater than the increase in the ENF GPP of 141.24 g C/m$^2$/year due to the conversion between ENF and MXF (i.e., ENF = 1436.65 g C/m$^2$/year in 2001; ENF = 1577.89 g C/m$^2$/year in 2018), ultimately resulting in a slight decrease in ENF GPP (Fig. 4a).*"

Section 4.1.2: This section is very nice now, discussing why the different CC effects can have positive and negative effects and relating it to your findings. Well done.
**Response**: Thanks very much for the positive feedback.

l. 502: LAI the dominant contributor? Second-dominant, no?
**Response**: Yes, the second dominant contributor. We have changed "*…LAI being the dominant contributor…*" to "*…LAI is the second dominant contributor…*".

l. 515: I appreciate the connection of GPP, NPP and carbon uptake. This is important to understand the implications of your study in terms of carbon uptake.

**Response**: Thank you very much for the positive comments.

Section 4.1.4: This is very introduction-y, and not what I meant in my previous review. I mean, yes, CO2 fertilization enhances GPP. But in your first version you linked that to C sequestration. That's ok but then you need to discuss also what happens to respiration in the meantime, what happens to tree mortality, and tree longevity. There are numerous uncertainties between CO2 fertilization effect and the carbon sink. That's what was missing. Not re-iterating the relevance of GPP.

**Response**: Thanks for the good suggestion. We have removed the introduction of GPP in this Section. We acknowledge that the carbon sink is not only determined by GPP, but also by processes like respiration, mortality, longevity, etc. We also acknowledge that this is a limitation of our study. We also added a discussion of the changes in respiration, tree mortality and tree longevity that related to the period of occurrence of carbon dioxide fertilization effect and carbon sink.

Revised text (Page 20, Lines 595-617):

*"Moreover, how much the net terrestrial carbon uptake increases in response to rising in atmospheric $CO_2$ is not just dependent on GPP but also on the processes like respiration, mortality, longevity, etc. For example, the increase in forest GPP due to $CO_2$ fertilisation leads to increased tree growth, and the final decomposition of the increased plant matter improves litter and soil organic matter pools, thereby enhancing heterotrophic respiration (Rh) (Quetin et al., 2023). Therefore, the $CO_2$ fertilisation effect can be counteracted by respiration. To date, there is no consensus on the response of photosynthesis and respiration to long-term increases in $CO_2$, due to the magnitude of such an impact and associated mechanisms still remaining uncertain (Sun et al., 2023). While several studies found the simultaneous reduction of respiration at elevated $CO_2$ (Sun et al., 2023.; Hamilton et al., 2001). The opposite conclusion has also been reported (Chen, Y et al., 2022; Crous et al., 2012). Additionally, the effect of elevated atmospheric $CO_2$ on GPP is also related to tree mortality. For example, elevated atmospheric $CO_2$ concentrations can lead to faster tree growth and decreasing the carbon turnover time. Consequently, the acceleration of the tree's life cycle and death will reduce carbon sequestration (Needham et al., 2020). Besides, the $CO_2$ fertilisation effect on forest carbon sinks can be limited by longevity. For example, Jiang et al., (2020) examined the responses of mature forests to atmospheric $CO_2$ enrichment. They found that elevated $CO_2$ led to a 12% increase in carbon uptake through GPP, but the carbon sequestration had not increased, and most of the carbon was returned to the atmosphere through respiration (Jiang et al., 2020). Currently, the forests in China are characterized by relatively young stand age (< 40 years old) due to a large number of new plantations, and thus China's forest carbon sequestration potential may continue to increase in the near future due to rising $CO_2$ concentration (Yao et al., 2018a). However, as the trend of increasing atmospheric $CO_2$ concentration may slow down, the carbon sink potential of China's forests may be further reduced in the future due to the weakening of the $CO_2$ fertilisation effect."*

References:

Quetin et al., 2023. Attributing Past Carbon Fluxes to $CO_2$ and Climate Change: Respiration Response to $CO_2$ Fertilization Shifts Regional Distribution of the Carbon Sink. Global Biogeochemical Cycles, 37(2), e2022GB007478.

Chen, Y. et al., 2022. The stimulatory effect of elevated $CO_2$ on soil respiration is unaffected by N addition. Science of The Total Environment, 813, 151907.

Crous et al., 2012. Light inhibition of leaf respiration in field-grown Eucalyptus saligna in whole-tree chambers under elevated atmospheric $CO_2$ and summer drought. Plant, Cell and Environment 35: 966-981.

Sun et al., 2023. Short- and long-term responses of leaf day respiration to elevated atmospheric $CO_2$. Plant Physiology, 191(4), 2204–2217.

Hamilton JG, et al., 2001. Direct and indirect effects of elevated $CO_2$ on leaf respiration in a forest ecosystem. Plant Cell Environment, 24, 975–982.

Needham et al., 2020. Forest responses to simulated elevated $CO_2$ under alternate hypotheses of size- and age-dependent mortality. Global Change Biology, 26, 5734–5753.

Jiang et al., 2020. The fate of carbon in a mature forest under carbon dioxide enrichment. Nature, 580, 227–231.

Yao, Y., Piao, S. and Wang, T., 2018a. Future biomass carbon sequestration capacity of Chinese forests. Science Bulletin, 63(17): 1108-1117.

I apologize if my review comment was not clear here.

Also, I would say that the statement "The carbon sequestered by vegetation through photosynthesis in a given unit of space and time, i.e., GPP" is not correct, because it ignores respiration.

**Response**: We thank the reviewer for pointing out the error. We have removed it from the revised text.

Finally, I think it makes much more sense to measure everything per m2 as you do now. The only problem: The total impacts are now not in the paper anymore. I think you should conclude the discussion with a short section on the total impact in Tg/year, and discuss the briefly discuss the total changes in areas, LAI and so forth.

**Response**: Thank you again. We also added a short section to discuss the total impact in TgC/year. Revised text (see also Page 16, Lines 465-471):

"*We also calculated the contributions of different factors to the total GPP of the study area, and also found that the $CO_2$ fertilisation effect (8.23 TgC/year, p<0.001) and LAI (4.55 TgC/year, p=0.005) contributed more to the increase in the total GPP of subtropical forests than that of FCC (1.35 TgC/year, p<0.001) and CC (1.11 TgC/year, p=0.08).*"

**Response to Referee2's Comments**

Dear Editor and Reviewer,

Thank you and the reviewer for the additional feedback on our manuscript. The reviewer lists some good points for clarification, and we have tried to address them in our revised revision. Reviewer comments are presented in black font; our responses are in blue font. Thank you again for your consideration. Please see below our replies, which hopefully will address the reviewer's comments in a satisfactory manner.

Sincerely yours,

Tao Chen, Félicien Meunier, Marc Peaucelle, Guoping Tang, Ye Yuan, Hans Verbeeck

I appreciate the authors' time and efforts in addressing specific concerns. The quality of the manuscript has been improved. I only have a few comments:

**Response:** Thank you very much for providing valuable suggestions and comments. Below we go through point-by-point our answers to the comments. We hope that you will find the result satisfying. We also polished the English throughout the revised manuscript using a language editing service (https://www.papertrue.com/ordering/academic-editing-proofreading-servicess). Please see the certificate below.

**PAPER TRUE**

**Certificate of Editing**

**1st March 2024**

**Title:** Elevated atmospheric CO2 concentration and vegetation structural changes contributed to GPP increase more than climate and forest cover changes in subtropical forests of China.

To whoever it may concern,

This is to certify that PaperTrue Editing and Proofreading Services (www.papertrue.com) have edited and proofread the following document for **Tao Chen** and duly delivered the edited document on **1st March 2024**. The document was edited by our native English-speaking editors.

Manuscript.docx (9689 words)

The PaperTrue Team
www.papertrue.com

PAPERTRUE PTE. LTD.

100 Peck Seah Street, #08-14, PS100, 079333, Singapore

1. In the authors' response to my general comment 2, they re-run the analysis with the long-term mean of climate variable instead of the value taken from the initial year in the previous version for attribution. They claim that the results show minor differences in which value to take when running the experiments. However, this is not my intention in that comment. I will reformulate it in another way. The trends of climate forcings are not significant during the studied period (Figure S9). Correspondingly, the temporal attribution of climate to GPP mostly originates from the variabilities of climate (Figure 6). But when $CO_2$ is attributed to GPP variations in the same way, the contribution of $CO_2$ mostly originates from the long-term trend of $CO_2$. Due to these inherent differences in the forcings, it is kind of expected that $CO_2$ turns out to be the most important factor. At least this has to be mentioned in the discussion.

**Response:** We appreciate this insightful comment. We are very grateful to the reviewer for raising such valuable scientific questions, which deepened our understanding of this aspect. Indeed, the trends of these climatic factors are not significant during the study period. We also acknowledge that the contribution of $CO_2$ mostly originates from the long-term trend of $CO_2$ due to its inherent characteristics. In this study, all driving data includes variabilities and long-term trends, and is used to drive the model. Therefore, they all included these two aspects when driving the model. However, it cannot be denied that some driving data is mainly contributed by variabilities during the study period, while others are driven by the contribution of long-term trends. The research period we are concerned about (i.e. 2001-2018) may be one of the reasons for these differences in driving data. Moreover, due to the structure and requirements of the model itself, we cannot change the inherent characteristics (e.g., de-trending of driver data) of the driving data when running the model. Actually, the previous studies (Chen et al., 2021; Chen et al., 2019; Sun et al., 2022; Li et al., 2023) also adopted the same way to study the impact of different driving factors (e.g., climatic factors, LAI, $CO_2$ fertilization effect, etc.) on carbon (GPP, NEP) and water (ET) fluxes changes.

Following your suggestion, we have mentioned this point in the revised version. Revised text to (also see Page 20, Lines 591-594):

"*Due to the inherent differences in the driving factors, it should be noted that the contribution of the $CO_2$ fertilisation effect to subtropical forest GPP changes mostly originates from the long-term trend of $CO_2$. However, the trend of climatic factors during the study period is not significant (Figure S9). The temporal attribution of climate to GPP is mainly due to its variability.*"

References:

Chen, S. et al., 2021. Vegetation structural change and $CO_2$ fertilization more than offset gross primary production decline caused by reduced solar radiation in China. Agricultural and Forest Meteorology, 296: 108207.

Chen et al., 2019b. Vegetation structural change since 1981 significantly enhanced the terrestrial carbon sink. Nature Communications, 10(1): 4259.

Sun et al., 2022. Causes for the increases in both evapotranspiration and water yield over vegetated mainland China during the last two decades. Agricultural and Forest Meteorology, 324: 109118.

Li et al., 2023. Vegetation growth due to $CO_2$ fertilization is threatened by increasing vapor pressure deficit. Journal of Hydrology, 619, 129292.

2. In the authors' response to my specific comment 14, the application of the u* threshold is confusing. Low u* values indicate weak turbulence and stable atmospheric conditions when fluxes are usually underestimated. Thus, the data below a specific u* is often considered unreliable and are often rejected.

**Response:** Thanks. As suggested, we have rewritten the statements as follows (also see Page 6, Lines 191-197):

*"For instance, the nighttime $CO_2$ flux correction mainly included removing outliers when there was precipitation, $CO_2$ concentration exceeded the instrument's measurement range, and there were fewer than 15,000 valid samples. Additionally, the u\* threshold was also used to judge low flux values. For the QYZ and ALS stations, when the threshold of u\* was below 0.2 m s$^{-1}$, the flux data was considered unreliable and was removed. However, the threshold of u\* =0.05 m s$^{-1}$ was used for DHS station, and when u\* was below 0.05 m s$^{-1}$, the flux data was rejected and removed."*

3. Line 22: Isn't the unit TgCyear-2 if it is from the slope of Figure 8 (a)?

**Response:** Thank you again. According to your general comment 3 (last round of review), we have changed the unit TgC/year to gC/m$^2$/year in the R1 version to make the results comparable. Yes, the unit TgCyear$^{-1}$ was derived from the slope of Figure 8 (a), but it was from the first submitted version and was not considered in the current study.